# Adapting to Unknown Low-Dimensional Structures in Score-Based Diffusion Models

**Gen Li**[*]
The Chinese University of Hong Kong
genli@cuhk.edu.hk

**Yuling Yan**[*]
University of Wisconsin-Madison
yuling.yan@wisc.edu

## Abstract

This paper investigates score-based diffusion models when the underlying target distribution is concentrated on or near low-dimensional manifolds within the higher-dimensional space in which they formally reside, a common characteristic of natural image distributions. Despite previous efforts to understand the data generation process of diffusion models, existing theoretical support remains highly suboptimal in the presence of low-dimensional structure, which we strengthen in this paper. For the popular Denoising Diffusion Probabilistic Model (DDPM), we find that the dependency of the error incurred within each denoising step on the ambient dimension $d$ is in general unavoidable. We further identify a unique design of coefficients that yields a converges rate at the order of $O(k^2/\sqrt{T})$ (up to log factors), where $k$ is the intrinsic dimension of the target distribution and $T$ is the number of steps. This represents the first theoretical demonstration that the DDPM sampler can adapt to unknown low-dimensional structures in the target distribution, highlighting the critical importance of coefficient design. All of this is achieved by a novel set of analysis tools that characterize the algorithmic dynamics in a more deterministic manner.

## 1 Introduction

Score-based diffusion models are a class of generative models that have gained prominence in the field of machine learning and artificial intelligence for their ability to generate high-quality new data instances from complex distributions, such as images, audio, and text [23, 11, 25, 24, 9]. These models operate by gradually transforming noise into samples from the target distribution through a denoising process guided by pre-trained neural networks that approximate the score functions. In practice, score-based diffusion models have demonstrated remarkable performance in generating realistic and diverse content across various domains [19, 20, 21, 7], achieving state-of-the-art performance in generative AI.

### 1.1 Diffusion models

The development of score-based diffusion models is deeply rooted in the theory of stochastic processes. At a high level, we consider a forward process:

$$X_0 \overset{\text{add noise}}{\longrightarrow} X_1 \overset{\text{add noise}}{\longrightarrow} \cdots \overset{\text{add noise}}{\longrightarrow} X_T, \tag{1.1}$$

which draws a sample from the target data distribution (i.e., $X_0 \sim p_{\mathsf{data}}$), then progressively diffuses it to Gaussian noise over time. The key aspect of the diffusion model is to construct a reverse process:

$$Y_T \overset{\text{denoise}}{\longrightarrow} Y_{T-1} \overset{\text{denoise}}{\longrightarrow} \cdots \overset{\text{denoise}}{\longrightarrow} Y_0 \tag{1.2}$$

---

[*]Equal contribution. Corresponding author: Yuling Yan (Email: yuling.yan@wisc.edu).

38th Conference on Neural Information Processing Systems (NeurIPS 2024).

satisfying $Y_t \overset{\mathrm{d}}{\approx} X_t$ for all $t$, which starts with pure Gaussian noise (i.e., $Y_T \sim \mathcal{N}(0, I_d)$) and gradually converts it back to a new sample $Y_0$ sharing a similar distribution to $p_{\mathsf{data}}$.

The classical results on time-reversal of SDEs [1, 10] provide the theoretical foundation for the above task. Consider a continuous time diffusion process:

$$\mathrm{d}X_t = -\frac{1}{2}\beta(t)X_t\mathrm{d}t + \sqrt{\beta(t)}\mathrm{d}W_t \quad (0 \le t \le T), \qquad X_0 \sim p_{\mathsf{data}} \qquad (1.3)$$

for some function $\beta : [0, T] \to \mathbb{R}^+$, where $(W_t)_{0 \le t \le T}$ is a standard Brownian motion. For a wide range of functions $\beta$, this process converges exponentially fast to a Gaussian distribution. Let $p_{X_t}(\cdot)$ be the density of $X_t$. One can construct a reverse-time SDE:

$$\mathrm{d}\widetilde{Y}_t = -\frac{1}{2}\beta(t)\big(\widetilde{Y}_t + 2\nabla \log p_{X_{T-t}}(\widetilde{Y}_t)\big) + \sqrt{\beta(t)}\mathrm{d}Z_t \quad (0 \le t \le T), \qquad \widetilde{Y}_0 \sim p_{X_T}, \quad (1.4)$$

where $(Z_t)_{0 \le t \le T}$ is another standard Brownian motion. Define $Y_t = \widetilde{Y}_{T-t}$. It is well-known that $X_t \overset{\mathrm{d}}{=} Y_t$ for all $0 \le t \le T$. Here, $\nabla \log p_{X_t}$ is called the score function for the law of $X_t$, which is not explicitly known.

The above result motivates the following paradigm: we can construct the forward process (1.1) by time-discretizing the diffusion process (1.3), and construct the reverse process (1.2) by discretizing the reverse-time SDE (1.4) and learning the score functions from the data. This approach leads to the popular DDPM sampler [11, 16]. Although the idea of the DDPM sampler is rooted in the theory of SDEs, the algorithm and analysis presented in this paper do not require any prior knowledge of SDEs.

This paper examines the accuracy of the DDPM sampler by establishing the proximity between the output distribution of the reverse process and the target data distribution. Since these two distributions are identical in the continuous time limit with perfect score estimation, the performance of the DDPM sampler is influenced by two sources of error: discretization error (due to a finite number of steps) and score estimation error (due to imperfect estimation of the scores). This paper views the score estimation step as a black box (often addressed by training a large neural network) and focuses on understanding how time discretization and imperfect score estimation affect the accuracy of the DDPM sampler.

## 1.2 Inadequacy of existing results

The past few years have witnessed a significant interest in studying the convergence guarantees for the DDPM sampler [4, 6, 3, 14, 13]. To facilitate discussion, we consider an ideal setting with perfect score estimation. In this context, existing results can be interpreted as follows: to achieve $\varepsilon$-accuracy (i.e., the total variation distance between the target and the output distribution is smaller than $\varepsilon$), it suffices to take a number of steps exceeding the order of $\mathsf{poly}(d)/\varepsilon^2$ (ignoring logarithm factors), where $d$ is the problem dimension. Among these results, the state-of-the-art is given by [3], which achieved linear dependency on the dimension $d$.

However, there seems to be a significant gap between the practical performance of the DDPM sampler and the existing theory. For example, for two widely used image datasets, CIFAR-10 (dimension $d = 32 \times 32 \times 3$) and ImageNet (dimension $d \ge 64 \times 64 \times 3$), it is known that 50 and 250 steps (also known as NFE, the number of function evaluations) are sufficient to generate good samples [16, 9]. This is in stark contrast with the existing theoretical guarantees discussed above, which suggest that the number of steps $T$ should exceed the order of the dimension $d$ to achieve good performance.

Empirical evidence suggests that the distributions of natural images are concentrated on or near low-dimensional manifolds within the higher-dimensional space in which they formally reside [22, 18]. In view of this, a reasonable conjecture is that the convergence rate of the DDPM sampler actually depends on the intrinsic dimension rather than the ambient dimension. However, the theoretical understanding of diffusion models when the support of the target data distribution has a low-dimensional structure remains vastly under-explored. As some recent attempts, [8] established the first convergence guarantee under the Wasserstein-1 metric. However, their error bound has linear dependence on the ambient dimension $d$ and exponential dependence on the diameter of the low-dimensional manifold. Another line of works [5, 26, 17] focused mainly on score estimation with properly chosen neural networks that exploit the low-dimensional structure, which is also different from our main focus.

## 1.3 Our contributions

In light of the large theory-practice gap and the insufficiency of prior results, this paper takes a step towards understanding the performance of the DDPM sampler when the target data distribution has low-dimensional structure. Our main contributions can be summarized as follows:

- We show that, with a particular coefficient design, the error of the DDPM sampler, evaluated by the total variation distance between the laws of $X_1$ and $Y_1$, is upper bounded by

$$\frac{k^2}{\sqrt{T}} + \sqrt{\frac{1}{T} \sum_{t=1}^{T} \mathbb{E}\big[ \|s_t (X_t) - s_t^\star (X_t)\|_2^2 \big]},$$

up to some logarithmic factors, where $k$ is the intrinsic dimension of the target data distribution (which will be rigorously defined later), and $s_t^\star$ (resp. $s_t$) is the true (resp. learned) score function at each step. The first term represents the discretization error (which vanishes as the number of steps $T$ goes to infinity), while the second term should be interpreted as the score matching error. This bound is nearly dimension-free — the ambient dimension $d$ only appears in logarithmic terms.

- We also show that our choice of the coefficients is, in some sense, the unique schedule that does not incur discretization error proportional to the ambient dimension $d$ at each step. This is in sharp contrast with the general setting without a low-dimensional structure, where a fairly wide range of coefficient designs can lead to convergence rates with polynomial dependence on $d$. Additionally, this confirms the observation that the performance of the DDPM sampler can be improved through carefully designing coefficients [2, 16].

As far as we know, this paper provides the first theory demonstrating the capability of the DDPM sampler in adapting to unknown low-dimensional structures.

## 2 Problem set-up

In this section, we introduce some preliminaries and key ingredients for the diffusion model and the DDPM sampler.

**Forward process.** We consider the forward process (1.1) of the form

$$X_t = \sqrt{1 - \beta_t} X_{t-1} + \sqrt{\beta_t} W_t \quad (t = 1, \ldots, T), \qquad X_0 \sim p_{\mathsf{data}}, \tag{2.1}$$

where $W_1, \ldots, W_T \overset{\text{i.i.d.}}{\sim} \mathcal{N}(0, I_d)$, and the learning rates $\beta_t \in (0, 1)$ will be specified later. For each $t \geq 1$, $X_t$ has a probability density function (PDF) supported on $\mathbb{R}^d$, and we will use $q_t$ to denote the law or PDF of $X_t$. Let $\alpha_t := 1 - \beta_t$ and $\overline{\alpha}_t := \prod_{i=1}^{t} \alpha_i$. It is straightforward to check that

$$X_t = \sqrt{\overline{\alpha}_t} X_0 + \sqrt{1 - \overline{\alpha}_t}\, \overline{W}_t \qquad \text{where} \qquad \overline{W}_t \sim \mathcal{N}(0, I_d). \tag{2.2}$$

We will choose the learning rates $\beta_t$ to ensure that $\overline{\alpha}_T$ becomes vanishingly small, such that $q_T \approx \mathcal{N}(0, I_d)$.

**Score functions.** The key ingredients for constructing the reverse process with the DDPM sampler are the score functions $s_t^\star : \mathbb{R}^d \to \mathbb{R}^d$ associated with each $q_t$, defined as

$$s_t^\star(x) := \nabla \log q_t(x) \quad (t = 1, \ldots, T).$$

These score functions are not explicitly known. Here we assume access to an estimate $s_t(\cdot)$ for each $s_t^\star(\cdot)$, and we define the averaged $\ell_2$ score estimation error as

$$\varepsilon_{\mathsf{score}}^2 := \frac{1}{T} \sum_{t=1}^{T} \mathbb{E}_{X \sim q_t} \left[ \|s_t(X) - s_t^\star(X)\|_2^2 \right].$$

This quantity captures the effect of imperfect score estimation in our theory.

**The DDPM sampler.**   To construct the reverse process (1.2), we use the DDPM sampler

$$Y_{t-1} = \frac{1}{\sqrt{\alpha_t}}\big(Y_t + \eta_t s_t\left(Y_t\right) + \sigma_t Z_t\big) \quad (t = T, \ldots, 1), \qquad Y_T \sim \mathcal{N}(0, I_d) \tag{2.3}$$

where $Z_1, \ldots, Z_T \overset{\text{i.i.d.}}{\sim} \mathcal{N}(0, I_d)$. Here $\eta_t, \sigma_t > 0$ are the hyperparameters that play an important role in the performance of the DDPM sampler, especially when the target data distribution has low-dimensional structure. As we will see, our theory suggests the following choice

$$\eta_t^\star = 1 - \alpha_t \qquad \text{and} \qquad \sigma_t^{\star 2} = \frac{(1 - \alpha_t)\left(\alpha_t - \overline{\alpha}_t\right)}{1 - \overline{\alpha}_t}. \tag{2.4}$$

For each $1 \leq t \leq T$, we will use $p_t$ to denote the law or PDF of $Y_t$.

**Target data distribution.**   Let $\mathcal{X} \subseteq \mathbb{R}^d$ be the support set of the target data distribution $p_{\mathsf{data}}$, i.e., the smallest closed set $C \subseteq \mathbb{R}^d$ such that $p_{\mathsf{data}}(C) = 1$. To allow for the greatest generality, we use the notion of $\varepsilon$-net and covering number (see e.g., [27]) to characterize the intrinsic dimension of $\mathcal{X}$. For any $\varepsilon > 0$, a set $\mathcal{N}_\varepsilon \subseteq \mathcal{X}$ is said to be an $\varepsilon$-net of $\mathcal{X}$ if for any $x \in \mathcal{X}$, there exists some $x'$ in $\mathcal{N}_\varepsilon$ such that $\|x - x'\|_2 \leq \varepsilon$. The covering number $N_\varepsilon(\mathcal{X})$ is defined as the smallest possible cardinality of an $\varepsilon$-net of $\mathcal{X}$.

- (**Low-dimensionality**) Fix $\varepsilon = T^{-c_\varepsilon}$, where $c_\varepsilon > 0$ is some sufficiently large universal constant. We define the intrinsic dimension of $\mathcal{X}$ to be some quantity $k > 0$ such that

$$\log N_\varepsilon(\mathcal{X}) \leq C_{\mathsf{cover}} k \log T$$

  for some constant $C_{\mathsf{cover}} > 0$.

- (**Bounded support**) Suppose that there exists a universal constant $c_R > 0$ such that

$$\sup_{x \in \mathcal{X}} \|x\|_2 \leq R \qquad \text{where} \qquad R := T^{c_R}.$$

  Namely we allow polynomial growth of the diameter of $\mathcal{X}$ in the number of steps $T$.

Our setting allows $\mathcal{X}$ to be concentrated on or near low-dimensional manifolds, which is less stringent than assuming an exact low-dimensional structure. In fact, our definition of the intrinsic dimension $k$ is the metric entropy of $\mathcal{X}$ (see e.g., [28]), which is widely used in statistics and learning theory to characterize the complexity of a set or a class. The low-dimensionality is also a concept of complexity, therefore it is natural to use covering number, or metric entropy to characterize the intrinsic dimension. As a sanity check, when $\mathcal{X}$ resides in an $r$-dimensional subspace of $\mathbb{R}^d$, a standard volume argument (see e.g., [27, Section 4.2.1]) gives $\log N_\varepsilon(\mathcal{X}) \asymp r \log(R/\varepsilon) \asymp r \log T$, suggesting that the intrinsic dimension $k$ is of order $r$ in this case. In addition, in applications like image generation, the data is naturally bounded, as pixel values are typically normalized within the range $[-1, 1]$. For example, the $\ell_2$ norm of an image from the CIFAR dataset is typically below 60.

**Learning rate schedule.**   Following [14], we adopt the following learning rate schedule

$$\beta_1 = \frac{1}{T^{c_0}}, \qquad \beta_{t+1} = \frac{c_1 \log T}{T} \min\left\{\beta_1\left(1 + \frac{c_1 \log T}{T}\right)^t, 1\right\} \quad (t = 1, \ldots, T-1) \tag{2.5}$$

for some sufficiently large constants $c_0, c_1 > 0$. This schedule is not unique – any other schedule of $\beta_t$ satisfying the properties in Lemma 8 can lead to the same result in this paper.

# 3   Main results

We are now positioned to present our main theoretical guarantees for the DDPM sampler.

## 3.1   Convergence analysis

We first present the convergence theory for the DDPM sampler. The proof can be found in Section 4.

**Theorem 1.** *Suppose that we take the coefficients for the DDPM sampler (2.3) to be $\eta_t = \eta_t^\star$ and $\sigma_t = \sigma_t^\star$ (cf. (2.4)), then there exists some universal constant $C > 0$ such that*

$$\mathsf{TV}\left(q_1, p_1\right) \leq C \frac{(k + \log d)^2 \log^3 T}{\sqrt{T}} + C \varepsilon_{\mathsf{score}} \log T. \tag{3.1}$$

Several implications of Theorem 1 follow immediately. The two terms in (3.1) correspond to discretization error and score matching error, respectively. Assuming perfect score estimation (i.e., $\varepsilon_{\mathsf{score}} = 0$) for the moment, our error bound (3.1) suggests an iteration complexity of order $k^4/\varepsilon^2$ (ignoring logarithmic factors) for achieving $\varepsilon$-accuracy, for any nontrivial target accuracy level $\varepsilon < 1$. In the absence of low-dimensional structure (i.e., $k \asymp d$), our result also recovers the iteration complexity in [4, 6, 3, 14] of order $\mathsf{poly}(d)/\varepsilon^2$.[2] This suggests that our choice of coefficients (2.4) allows the DDPM sampler to adapt to any potential (unknown) low-dimensional structure in the target data distribution, and remains a valid criterion in the most general settings. The score matching error in (3.1) scales proportionally with $\varepsilon_{\mathsf{score}}$, suggesting that the DDPM sampler is stable to imperfect score estimation.

## 3.2   Uniqueness of coefficient design

In this section, we examine the importance of the coefficient design in the adaptivity of the DDPM sampler to intrinsic low-dimensional structure. Our goal is to show that, unless the coefficients $\eta_t, \sigma_t$ of the DDPM sampler (2.3) are chosen according to (2.4), discretization errors proportional to the ambient dimension $d$ will emerge in each denoising step.

In this paper, as well as in most previous DDPM literature, the analysis on the error $\mathsf{TV}(q_1, p_1)$ usually starts with the following decomposition

$$\mathsf{TV}^2(q_1, p_1) \overset{(i)}{\leq} \frac{1}{2} \mathsf{KL}\left(p_{X_1} \| p_{Y_1}\right) \overset{(ii)}{\leq} \frac{1}{2} \mathsf{KL}\left(p_{X_1,\ldots,X_T} \| p_{Y_1,\ldots,Y_T}\right) \tag{3.2}$$

$$\overset{(iii)}{=} \frac{1}{2} \underbrace{\mathsf{KL}\left(p_{X_T} \| p_{Y_T}\right)}_{\text{initialization error}} + \frac{1}{2} \sum_{t=2}^{T} \underbrace{\mathbb{E}_{x_t \sim q_t}\left[\mathsf{KL}\left(p_{X_{t-1}|X_t}\left(\cdot \,|\, x_t\right) \| p_{Y_{t-1}|Y_t}\left(\cdot \,|\, x_t\right)\right)\right]}_{\text{error incurred in the } (T+1-t)\text{-th denoising step}}.$$

Here step (i) follows from Pinsker's inequality, step (ii) utilizes from the data-processing inequality, while step (iii) uses the chain rule of KL divergence. We may interpret each term in the above decomposition as the error incurred in each denoising step. In fact, this decomposition is also closely related to the variational bound on the negative log-likelihood of the reverse process, which is the optimization target for training DDPM [11, 2, 16].

We consider a target distribution $p_{\mathsf{data}} = \mathcal{N}(0, I_k)$, where $I_k \in \mathbb{R}^{d \times d}$ is a diagonal matrix with $I_{i,i} = 1$ for $1 \leq i \leq k$ and $I_{i,i} = 0$ for $k + 1 \leq i \leq d$. This is a simple distribution over $\mathbb{R}^d$ that is supported on a $k$-dimensional subspace.[3] Our second theoretical result provides a lower bound for the error incurred in each denoising step for this target distribution. The proof can be found in Appendix B.

**Theorem 2.** *Consider the target distribution $p_{\mathsf{data}} = \mathcal{N}(0, I_k)$ and assume that $k \leq d/2$. For the DDPM sampler (2.3) with perfect score estimation (i.e., $s_t(\cdot) = s_t^\star(\cdot)$ for all $t$) and arbitrary coefficients $\eta_t, \sigma_t > 0$, we have*

$$\mathbb{E}_{x_t \sim q_t}\left[\mathsf{KL}\left(p_{X_{t-1}|X_t}\left(\cdot \,|\, x_t\right) \| p_{Y_{t-1}|Y_t}\left(\cdot \,|\, x_t\right)\right)\right] \geq \frac{d}{4}\left(\eta_t - \eta_t^\star\right)^2 + \frac{d}{40}\left(\frac{\sigma_t^{\star 2}}{\sigma_t^2} - 1\right)^2$$

*for each $2 \leq t \leq T$. See (2.4) for the definitions of $\eta_t^\star$ and $\sigma_t^\star$.*

---

[2] Our result exhibits a quartic dimension dependency, which is worse than the linear dependency in [3]. This is mainly because we use a completely different analysis. It is not clear whether their analysis, which utilizes the SDE and stochastic localization toolbox, can tackle the problem with low-dimensional structure.

[3] Although this is not a bounded distribution, similar results can be established if we truncate $\mathcal{N}(0, I_k)$ at the radius $R = T^{c_R}$. However this is not essential and will make the result unnecessarily complicated, hence is omitted for clarity.

Theorem 2 shows that, unless we choose $\eta_t$ and $\sigma_t^2$ to be identical (or exceedingly close) to $\eta_t^\star$ and $\sigma_t^{\star 2}$, the corresponding denoising step will incur an undesired error that is linear in the ambient dimension $d$. This highlights the critical importance of coefficient design for the DDPM sampler, especially when the target distribution exhibits a low-dimensional structure.

Finally, we would like to make note that the above argument only demonstrates the impact of coefficient design on an *upper bound* (3.2) of the error $\mathsf{TV}(q_1, p_1)$, rather than the error itself. It might be possible that a broader range of coefficients can lead to dimension-independent error bound like (3.1), while the upper bound (3.2) remains dimension-dependent. This calls for new analysis tools (since we cannot use the loose upper bound (3.1) in the analysis), which we leave for future works.

## 4    Analysis for the DDPM sampler (Proof of Theorem 1)

This section is devoted to establishing Theorem 1. The idea is to bound the error incurred in each denoising step as characterized in the decomposition (3.2), namely for each $2 \leq t \leq T$, we need to bound

$$\mathbb{E}_{x_t \sim q_t} \left[ \mathsf{KL} \left( p_{X_{t-1}|X_t} \left( \cdot \, | \, x_t \right) \, \| \, p_{Y_{t-1}|Y_t} \left( \cdot \, | \, x_t \right) \right) \right].$$

This requires connecting the two conditional distributions $p_{X_{t-1}|X_t}$ and $p_{Y_{t-1}|Y_t}$. It would be convenient to decouple the errors from time discretization and imperfect score estimation by introducing auxiliary random variables

$$Y_{t-1}^\star := \frac{1}{\sqrt{\alpha_t}} \left( Y_t + \eta_t^\star s_t^\star \left( Y_t \right) + \sigma_t^\star Z_t \right) \qquad (2 \leq t \leq T). \tag{4.1}$$

On a high level, for each $2 \leq t \leq T$, our proof consists of the following steps:

1. Identify a typical set $\mathcal{A}_t \subseteq \mathbb{R}^d \times \mathbb{R}^d$ such that $(X_t, X_{t-1}) \in \mathcal{A}_t$ with high probability.

2. Establish point-wise proximity $p_{X_{t-1}|X_t}(x_{t-1} \, | \, x_t) \approx p_{Y_{t-1}^\star|Y_t}(x_{t-1} \, | \, x_t)$ for $(x_t, x_{t-1}) \in \mathcal{A}_t$.

3. Characterize the deviation of $p_{Y_{t-1}^\star|Y_t}$ from $p_{Y_{t-1}|Y_t}$ caused by imperfect score estimation.

### 4.1    Step 1: identifying high-probability sets

For simplicity of presentation, we assume without loss of generality that $k \geq \log d$ throughout the proof.[4] Let $\{x_i^\star\}_{1 \leq i \leq N_\varepsilon}$ be an $\varepsilon$-net of $\mathcal{X}$, and let $\{\mathcal{B}_i\}_{1 \leq i \leq N_\varepsilon}$ be a disjoint $\varepsilon$-cover for $\mathcal{X}$ such that $x_i^\star \in \mathcal{B}_i$. Let

$$\mathcal{I} := \left\{ 1 \leq i \leq N_\varepsilon : \mathbb{P}(X_0 \in \mathcal{B}_i) \geq \exp(-C_1 k \log T) \right\},$$

$$\mathcal{G} := \left\{ \omega \in \mathbb{R}^d : \|\omega\|_2 \leq 2\sqrt{d} + \sqrt{C_1 k \log T}, \quad \text{and} \right.$$

$$\left. |(x_i^\star - x_j^\star)^\top \omega| \leq \sqrt{C_1 k \log T} \|x_i^\star - x_j^\star\|_2 \quad \text{for all} \quad 1 \leq i, j \leq N_\varepsilon \right\},$$

where $C_1 > 0$ is some sufficiently large universal constants. Then $\cup_{i \in \mathcal{I}} \mathcal{B}_i$ and $\mathcal{G}$ can be interpreted as high probability sets for the variable $X_0$ and a standard Gaussian random variable in $\mathbb{R}^d$. For each $t = 1, \ldots T$, we define a typical set for each $X_t$ as follows

$$\mathcal{T}_t := \left\{ \sqrt{\overline{\alpha}_t} x_0 + \sqrt{1 - \overline{\alpha}_t} \omega : x_0 \in \cup_{i \in \mathcal{I}} \mathcal{B}_i, \omega \in \mathcal{G} \right\},$$

and a typical set for $(X_t, X_{t-1})$ jointly as follows

$$\mathcal{A}_t := \left\{ (x_t, x_{t-1}) : x_t \in \mathcal{T}_t, \frac{x_t - \sqrt{\alpha_t} x_{t-1}}{\sqrt{1 - \alpha_t}} \in \mathcal{G} \right\}.$$

The following lemma shows that $\mathcal{A}_t$ is indeed a high-probability set for $(X_t, X_{t-1})$.

**Lemma 1.**  *Suppose that $C_1 \gg C_{\mathsf{cover}}$. Then for each $1 \leq t \leq T$ we have*

$$\mathbb{P} \left( (X_t, X_{t-1}) \notin \mathcal{A}_t \right) \leq \exp \left( -\frac{C_1}{4} k \log T \right).$$

*Proof.*  See Appendix A.3. □

---

[4]If $k < \log d$, we may redefine $k := \log d$, which does not change the desired bound (3.1).

## 4.2 Step 2: connecting conditional densities $p_{X_{t-1}|X_t}$ and $p_{Y_{t-1}^\star|Y_t}$

Given the definition of $Y_{t-1}^\star$ in (4.1), we can write down the conditional density $p_{Y_{t-1}^\star|Y_t}$ as follows

$$p_{Y_{t-1}^\star|Y_t}\left(x_{t-1}\,|\,x_t\right) = \left(\frac{\alpha_t}{2\pi\sigma_t^{\star 2}}\right)^{d/2}\exp\left(-\frac{\|\sqrt{\alpha_t}x_{t-1} - x_t - \eta_t^\star s_t^\star\left(x_t\right)\|_2^2}{2\sigma_t^{\star 2}}\right). \tag{4.2}$$

Next, we will investigate the conditional density $p_{X_{t-1}|X_t}$ for the forward process. For each $x_0 \in \mathcal{X}$, we define the shorthand notation

$$\widehat{x}_0 := \mathbb{E}\left[X_0\,|\,X_t = x_t\right] = \int_{x_0} x_0 p_{X_0|X_t}\left(x_0\,|\,x_t\right)\mathrm{d}x_0, \tag{4.3}$$

and define a function $\Delta_{x_t,x_{t-1}} : \mathcal{X} \to \mathbb{R}$ as follows

$$\Delta_{x_t,x_{t-1}}\left(x_0\right) := -\frac{\sqrt{\alpha_t}}{\alpha_t - \overline{\alpha}_t}\left(\sqrt{\alpha_t}x_{t-1} - x_t\right)^\top\left(\widehat{x}_0 - x_0\right) - \frac{\left(1 - \alpha_t\right)\overline{\alpha}_t}{2\left(\alpha_t - \overline{\alpha}_t\right)\left(1 - \overline{\alpha}_t\right)}\|\widehat{x}_0 - x_0\|_2^2$$

$$- \frac{\left(1 - \alpha_t\right)\sqrt{\alpha_t}}{\left(\alpha_t - \overline{\alpha}_t\right)\left(1 - \overline{\alpha}_t\right)}\left(x_t - \sqrt{\alpha_t}\widehat{x}_0\right)^\top\left(\widehat{x}_0 - x_0\right). \tag{4.4}$$

The next lemma provides a characterization for $p_{X_{t-1}|X_t}$ that shows an explicit connection with $p_{Y_{t-1}^\star|Y_t}$.

**Lemma 2.** *For any pair $(x_t, x_{t-1}) \in \mathbb{R}^d \times \mathbb{R}^d$, we have*

$$p_{X_{t-1}|X_t}\left(x_{t-1}\,|\,x_t\right) = \left(\frac{\alpha_t}{2\pi\sigma_t^{\star 2}}\right)^{d/2}\exp\left(-\frac{\|\sqrt{\alpha_t}x_{t-1} - x_t - \eta_t^\star s_t^\star\left(x_t\right)\|_2^2}{2\sigma_t^{\star 2}}\right)$$

$$\cdot \int_{\mathcal{X}}\exp\left(\Delta_{x_t,x_{t-1}}\left(x_0\right)\right)p_{X_0|X_t}\left(x_0\,|\,x_t\right)\mathrm{d}x_0.$$

*Proof.* See Appendix A.4. □

Taking Lemma 2 and (4.2) collectively yields

$$\frac{p_{X_{t-1}|X_t}\left(x_{t-1}\,|\,x_t\right)}{p_{Y_{t-1}^\star|Y_t}\left(x_{t-1}\,|\,x_t\right)} = \int_{\mathcal{X}}\exp\left(\Delta_{x_t,x_{t-1}}\left(x_0\right)\right)p_{X_0|X_t}\left(x_0\,|\,x_t\right)\mathrm{d}x_0,$$

which allows us to control the density ratio by the magnitude of $\Delta_{x_t,x_{t-1}}$. By a careful analysis of the above integral for all $(x_t, x_{t-1}) \in \mathcal{A}_t$, we show in the next lemma that the density ratio is uniformly close to 1 within the typical set $\mathcal{A}_t$.

**Lemma 3.** *Suppose that $T \gg k^2\log^3 T$. Then there exists some universal constant $C_5 > 0$ such that, for any $2 \leq t \leq T$ and any $(x_t, x_{t-1}) \in \mathcal{A}_t$, we have*

$$\left|\frac{p_{X_{t-1}|X_t}\left(x_{t-1}\,|\,x_t\right)}{p_{Y_{t-1}^\star|Y_t}\left(x_{t-1}\,|\,x_t\right)} - 1\right| \leq C_5\frac{k^2\log^3 T}{T} \leq \frac{1}{2}.$$

*Proof.* See Appendix A.5. □

For $(x_t, x_{t-1})$ outside the typical set $\mathcal{A}_t$, the following lemma gives a coarse uniform bound for the density ratio, which is already sufficient for our later analysis.

**Lemma 4.** *Suppose that $T \gg 1$. Then for any $2 \leq t \leq T$ and any pair $(x_t, x_{t-1}) \in \mathbb{R}^d \times \mathbb{R}^d$, we have*

$$\left|\log\frac{p_{X_{t-1}|X_t}\left(x_{t-1}\,|\,x_t\right)}{p_{Y_{t-1}^\star|Y_t}\left(x_{t-1}\,|\,x_t\right)}\right| \leq T^{c_0 + 2c_R}\left(\|\sqrt{\alpha_t}x_{t-1} - x_t\|_2 + \|x_t\|_2 + 1\right).$$

*Proof.* See Appendix A.6. □

Armed with Lemmas 3 and 4, we are ready to bound the expected KL divergence between the two conditional distributions $p_{X_{t-1}|X_t}$ and $p_{Y_{t-1}^\star|Y_t}$.

### 4.3 Step 3: bounding the KL divergence between $p_{X_{t-1}|X_t}$ and $p_{Y_{t-1}^\star|Y_t}$

We first decompose the expected KL divergence between $p_{X_{t-1}|X_t}$ and $p_{Y_{t-1}^\star|Y_t}$ into

$$
\mathbb{E}_{x_t \sim q_t} \left[ \mathsf{KL} \left( p_{X_{t-1}|X_t} \left( \cdot \,|\, x_t \right) \,\|\, p_{Y_{t-1}^\star|Y_t} \left( \cdot \,|\, x_t \right) \right) \right]
$$
$$
= \left( \int_{\mathcal{A}_t} + \int_{\mathcal{A}_t^c} \right) p_{X_{t-1}|X_t} \left( x_{t-1} \,|\, x_t \right) \log \left( \frac{p_{X_{t-1}|X_t} \left( x_{t-1} \,|\, x_t \right)}{p_{Y_{t-1}^\star|Y_t} \left( x_{t-1} \,|\, x_t \right)} \right) p_{X_t} \left( x_t \right) \mathrm{d}x_{t-1} \mathrm{d}x_t
$$
$$
=: \Delta_{t,1} + \Delta_{t,2},
$$

where $\Delta_{t,1}$ and $\Delta_{t,2}$ are the integrals over $\mathcal{A}_t$ and $\mathcal{A}_t^c$. It boils down to bounding these two terms.

By a direct application of Lemma 3 together with the first-order Taylor expansion of $\log(x)$ around $x = 1$, one can easily show that $|\Delta_{t,1}| \lesssim k^2 \log^3(T)/T$. However this naive bound will lead to a vacuous final bound on $\mathsf{TV}(q_1, p_1)$, which depends on the sum of $\Delta_{t,1}$ over all $2 \le t \le T$ according to (3.2). By a more careful analysis, we achieve a better bound for $\Delta_{t,1}$, as shown in the following lemma.

**Lemma 5.** *Suppose that $T \gg k^2 \log^3 T$. Then for each $2 \le t \le T$, we have*

$$
|\Delta_{t,1}| \le 2C_5^2 \frac{k^4 \log^6 T}{T^2}.
$$

*Proof.* See Appendix A.7. □

For $\Delta_{t,2}$, we can employ the course bound in Lemma 4 to show that it is exponentially small.

**Lemma 6.** *Suppose that $T \gg 1$. Then for each $2 \le t \le T$, we have*

$$
|\Delta_{t,2}| \le \exp \left( -\frac{C_1}{16} k \log T \right).
$$

*Proof.* See Appendix A.8. □

By putting together Lemma 5 and Lemma 6, we achieve

$$
\mathbb{E}_{x_t \sim q_t} \left[ \mathsf{KL} \left( p_{X_{t-1}|X_t} \left( \cdot \,|\, x_t \right) \,\|\, p_{Y_{t-1}^\star|Y_t} \left( \cdot \,|\, x_t \right) \right) \right] = \Delta_{t,1} + \Delta_{t,2} \le 3C_5^2 \frac{k^4 \log^6 T}{T^2} \tag{4.5}
$$

provided that $T$ is sufficiently large.

### 4.4 Step 4: bounding the KL divergence between $p_{X_{t-1}|X_t}$ and $p_{Y_{t-1}|Y_t}$

Since our goal is to bound the expected KL divergence between $p_{X_{t-1}|X_t}$ and $p_{Y_{t-1}|Y_t}$, we also need to upper bound the following difference

$$
\mathbb{E}_{x_t \sim q_t} \left[ \mathsf{KL} \left( p_{X_{t-1}|X_t} \left( \cdot \,|\, x_t \right) \,\|\, p_{Y_{t-1}|Y_t} \left( \cdot \,|\, x_t \right) \right) \right] - \mathbb{E}_{x_t \sim q_t} \left[ \mathsf{KL} \left( p_{X_{t-1}|X_t} \left( \cdot \,|\, x_t \right) \,\|\, p_{Y_{t-1}^\star|Y_t} \left( \cdot \,|\, x_t \right) \right) \right]
$$
$$
= \int \left[ \int p_{X_{t-1}|X_t} \left( x_{t-1} \,|\, x_t \right) \log \frac{p_{Y_{t-1}^\star|Y_t} \left( x_{t-1} \,|\, x_t \right)}{p_{Y_{t-1}|Y_t} \left( x_{t-1} \,|\, x_t \right)} \mathrm{d}x_{t-1} \right] q_t \left( x_t \right) \mathrm{d}x_t \tag{4.6}
$$
$$
= \int p_{X_{t-1}, X_t} \left( x_{t-1}, x_t \right) \left( -\frac{\alpha_t \| x_{t-1} - \mu_t^\star \left( x_t \right) \|_2^2}{2\sigma_t^{\star 2}} + \frac{\alpha_t \| x_{t-1} - \mu_t \left( x_t \right) \|_2^2}{2\sigma_t^{\star 2}} \right) \mathrm{d}x_{t-1} \mathrm{d}x_t
$$
$$
= \frac{\eta_t^{\star 2}}{2\sigma_t^{\star 2}} \mathbb{E}_{x_t \sim q_t} \left[ \| \varepsilon_t \left( x_t \right) \|_2^2 \right] + \frac{\eta_t^\star \sqrt{\alpha_t}}{\sigma_t^{\star 2}} \underbrace{\int p_{X_{t-1}, X_t} \left( x_{t-1}, x_t \right) \left( x_{t-1} - \mu_t^\star \left( x_t \right) \right)^\top \varepsilon_t \left( x_t \right) \mathrm{d}x_{t-1} \mathrm{d}x_t}_{=:K_t},
$$

where we define

$$
\varepsilon_t \left( x_t \right) := s_t^\star \left( x_t \right) - s_t \left( x_t \right), \quad \mu_t^\star \left( x_t \right) := \frac{x_t + \eta_t^\star s_t^\star \left( x_t \right)}{\sqrt{\alpha_t}}, \quad \mu_t \left( x_t \right) := \frac{x_t + \eta_t^\star s_t \left( x_t \right)}{\sqrt{\alpha_t}}. \tag{4.7}
$$

It then boils down to bounding $K_t$, which is presented in the following lemma.

**Lemma 7.** *Suppose that $T \gg k^2 \log^3 T$. Then we have*

$$|K_t| \leq 4C_5 \frac{k^2 \log^3 T}{T} \sqrt{\frac{c_1 \log T}{T}} \mathbb{E}_{x_t \sim q_t}^{1/2} \left[ \|\varepsilon_t(x_t)\|_2^2 \right].$$

*Proof.* See Appendix A.8. $\qquad\qquad\qquad\qquad\qquad\qquad\qquad\qquad\qquad\qquad\qquad\qquad\square$

Hence we know that for $2 \leq t \leq T$,

$$\mathbb{E}_{x_t \sim q_t} \left[ \mathsf{KL}\left(p_{X_{t-1}|X_t}(\cdot \,|\, x_t) \,\|\, p_{Y_{t-1}|Y_t}(\cdot \,|\, x_t)\right) \right] - \mathbb{E}_{x_t \sim q_t}\left[\mathsf{KL}\left(p_{X_{t-1}|X_t}(\cdot \,|\, x_t) \,\|\, p_{Y_{t-1}^\star|Y_t}(\cdot \,|\, x_t)\right)\right]$$

$$\leq \frac{(1-\overline{\alpha}_t)(1-\alpha_t)}{2(\alpha_t - \overline{\alpha}_t)} \mathbb{E}_{x_t \sim q_t} \left[ \|\varepsilon_t(x_t)\|_2^2 \right] + 4C_5 \frac{1-\overline{\alpha}_t}{\alpha_t - \overline{\alpha}_t} \frac{k^2 \log^3 T}{T} \sqrt{\frac{c_1 \log T}{T}} \mathbb{E}_{x_t \sim q_t}^{1/2} \left[ \|\varepsilon_t(x_t)\|_2^2 \right]$$

$$\leq \frac{4c_1 \log T}{T} \mathbb{E}_{x_t \sim q_t} \left[ \|\varepsilon_t(x_t)\|_2^2 \right] + 8C_5 \frac{k^2 \log^3 T}{T} \sqrt{\frac{c_1 \log T}{T}} \mathbb{E}_{x_t \sim q_t}^{1/2} \left[ \|\varepsilon_t(x_t)\|_2^2 \right]. \qquad (4.8)$$

Here the first relation follows from Lemma 7 and (2.4); while the second relation follows from Lemma 8 and holds provided that $T$ is sufficiently large.

## 4.5  Step 5: putting everything together

By taking (4.5) and (4.8) collectively, we have

$$\mathbb{E}_{x_t \sim q_t} \left[ \mathsf{KL}\left(p_{X_{t-1}|X_t}(\cdot \,|\, x_t) \,\|\, p_{Y_{t-1}|Y_t}(\cdot \,|\, x_t)\right) \right]$$

$$\leq 3C_5^2 \frac{k^4 \log^6 T}{T^2} + \frac{4c_1 \log T}{T} \mathbb{E}_{x_t \sim q_t} \left[ \|\varepsilon_t(x_t)\|_2^2 \right] + 8C_5 \frac{k^2 \log^3 T}{T} \sqrt{\frac{c_1 \log T}{T}} \mathbb{E}_{x_t \sim q_t}^{1/2} \left[ \|\varepsilon_t(x_t)\|_2^2 \right]$$

$$\leq 7C_5^2 \frac{k^4 \log^6 T}{T^2} + \frac{8c_1 \log T}{T} \mathbb{E}_{x_t \sim q_t} \left[ \|\varepsilon_t(x_t)\|_2^2 \right]. \qquad (4.9)$$

Here the last relation follows from an application of the AM-GM inequality

$$8C_5 \frac{k^2 \log^3 T}{T} \sqrt{\frac{c_1 \log T}{T}} \mathbb{E}_{x_t \sim q_t}^{1/2} \left[ \|\varepsilon_t(x_t)\|_2^2 \right] \leq \frac{4c_1 \log T}{T} \mathbb{E}_{x_t \sim q_t} \left[ \|\varepsilon_t(x_t)\|_2^2 \right] + 4C_5^2 \frac{k^4 \log^6 T}{T^2}.$$

Finally we conclude that

$$\mathsf{TV}^2(q_1, p_1) \leq \mathsf{KL}\left(p_{X_T} \| p_{Y_T}\right) + \sum_{t=2}^{T} \mathbb{E}_{x_t \sim q_t} \left[ \mathsf{KL}\left(p_{X_{t-1}|X_t}(\cdot \,|\, x_t) \,\|\, p_{Y_{t-1}|Y_t}(\cdot \,|\, x_t)\right) \right]$$

$$\leq 8C_5^2 \frac{k^4 \log^6 T}{T} + \frac{8c_1 \log T}{T} \sum_{t=2}^{T} \mathbb{E}_{x_t \sim q_t} \left[ \|\varepsilon_t(x_t)\|_2^2 \right],$$

as claimed. Here the first relation follows from (3.2), while the second relation follows from the fact that $\mathsf{KL}(p_{X_T} \| p_{Y_T}) \leq T^{-100}$ provided that $T$ is sufficiently large (see Lemma 10).

# 5  Simulation study

We conducted a simple simulation to compare our coefficient design (2.4) with another design

$$\eta_t = \sigma_t^2 = 1 - \alpha_t \qquad \text{for} \qquad 1 \leq t \leq T, \qquad (5.1)$$

which has been widely adopted in theoretical analysis of diffusion model (see e.g., [14, 15]). We consider the degenerated Gaussian distribution $p_{\mathsf{data}} = \mathcal{N}(0, I_k)$ in Theorem 2 as a tractable example, and run the DDPM sampler with exact score functions (so that the error only comes from discretization). We fix the intrinsic dimension $k = 8$, and let the ambient dimension $d$ grow from 10 to $10^3$. We implement the experiment for four different number of steps $T \in \{100, 200, 500, 1000\}$. Instead of using the learning rate schedule (2.5), which is chosen mainly to facilitate analysis, we use the schedule in [11] that is commonly used in practice. Figure 1 displays the error, in terms of both the TV distance $\mathsf{TV}(q_1, p_1)$ and KL divergence $\mathsf{KL}(q_1 \| p_1)$, as the ambient dimension $d$ varies. As we can see, our design (2.4) leads to dimension-independent error while the other design (5.1) incures an error that grows as $d$ increases. This provides empirical evidence that (2.4) represents a unique coefficient design for DDPM in achieving dimension-independent error.

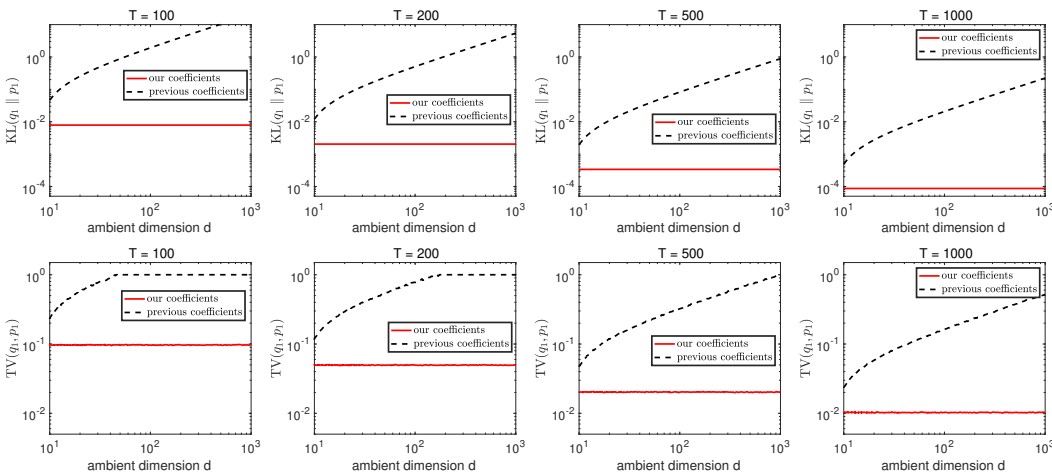

Figure 1: The KL divergence between $q_1$ and $p_1$ for $T \in \{100, 200, 500, 1000\}$, when $p_{\mathsf{data}} = \mathcal{N}(0, I_k)$. We fix the low intrinsic dimension $k = 8$, and let the ambient dimension $d$ grow from 10 to 1000.

## 6  Discussion

The present paper investigates the DDPM sampler when the target distribution is concentrated on or near low-dimensional manifolds. We identify a particular coefficient design that enables the adaptivity of the DDPM sampler to unknown low-dimensional structures and establish a dimension-free convergence rate at the order of $k^2/\sqrt{T}$ (up to logarithmic factors). We conclude this paper by pointing out several directions worthy of future investigation. To begin with, our theory yields an iteration complexity that scales quartically in the intrinsic dimension $k$, which is likely sub-optimal. Improving this dependency calls for more refined analysis tools. Recent work [15] achieved a convergence rate of order $O(d/T)$, suggesting the potential for enhancing the dependence on $T$. Furthermore, as we have discussed in the end of Section 3.2, it is not clear whether our coefficient design (2.4) is unique in terms of achieving dimension-independent error $\mathsf{TV}(q_1, p_1)$. Finally, the analysis ideas and tools developed for the DDPM sampler might be extended to study another popular DDIM sampler.

## Acknowledgments and Disclosure of Funding

Gen Li is supported in part by the Chinese University of Hong Kong Direct Grant for Research. Yuling Yan was supported in part by a Norbert Wiener Postdoctoral Fellowship from MIT.

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

# A  Proof of auxiliary lemmas for the DDPM sampler

## A.1  Preliminaries

Fix any $x_t \in \mathcal{T}_t$, there exists an index $i(x_t) \in \mathcal{I}$, two points $x_0(x_t) \in \mathcal{B}_{i(x_t)}$ and $\omega \in \mathcal{G}$ such that

$$x_t = \sqrt{\overline{\alpha}_t} x_0(x_t) + \sqrt{1 - \overline{\alpha}_t}\omega. \tag{A.1}$$

For any $r > 0$, define a set

$$\mathcal{I}(x_t; r) := \left\{ 1 \le i \le N_\varepsilon : \overline{\alpha}_t \|x_i^\star - x_{i(x_t)}^\star\|_2^2 \le r \cdot k(1 - \overline{\alpha}_t) \log T \right\}. \tag{A.2}$$

For some sufficiently large constant $C_3 > 0$, define

$$\mathcal{X}_t(x_t) := \bigcup_{i \in \mathcal{I}(x_t; C_3)} \mathcal{B}_i \quad \text{and} \quad \mathcal{Y}_t(x_t) := \bigcup_{i \notin \mathcal{I}(x_t; C_3)} \mathcal{B}_i.$$

Namely, $\mathcal{X}_t(x_t)$ (resp. $\mathcal{Y}_t(x_t)$) contains the indices of the $\varepsilon$-covering that are close (resp. far) from $\mathcal{B}_{i(x_t)}$. We require that

$$\varepsilon \ll \sqrt{\frac{1 - \overline{\alpha}_t}{\overline{\alpha}_t}} \min\left\{ 1, \sqrt{\frac{k \log T}{d}} \right\}, \tag{A.3}$$

which is guaranteed by our assumption that $\varepsilon = T^{-c_\varepsilon}$ for some sufficiently large constant $c_\varepsilon > 0$. Under this condition, for any $x, x' \in \mathcal{X}_t(x_t)$ we have

$$\|x - x'\|_2 \le \|x - x_{i(x_t)}^\star\|_2 + \|x' - x_{i(x_t)}^\star\|_2 \le 2\sqrt{\frac{C_3 k(1 - \overline{\alpha}_t)\log T}{\overline{\alpha}_t}} + 2\varepsilon \le 3\sqrt{\frac{C_3 k(1 - \overline{\alpha}_t)\log T}{\overline{\alpha}_t}}$$

Hence for any $x, x' \in \mathcal{X}_t(x_t)$ we have

$$\overline{\alpha}_t \|x - x'\|_2^2 \le 9 C_3 k (1 - \overline{\alpha}_t) \log T. \tag{A.4}$$

In addition, for any $x, x' \in \mathcal{X}$, suppose that $x \in \mathcal{B}_i$ and $x' \in \mathcal{B}_j$. For any $\omega \in \mathcal{G}$, we have

$$
\begin{aligned}
\left| \omega^\top (x - x') \right| &= \left| \omega^\top (x_i^\star - x_j^\star) \right| + \left| \omega^\top (x - x_i^\star) \right| + \left| \omega^\top (x_j^\star - x') \right| \\
&\overset{(i)}{\le} \sqrt{C_1 k \log T} \|x_i^\star - x_j^\star\|_2 + \|x - x_i^\star\|_2 \|\omega\|_2 + \|x' - x_j^\star\|_2 \|\omega\|_2 \\
&\overset{(ii)}{\le} \sqrt{C_1 k \log T} \|x_i^\star - x_j^\star\|_2 + 2\left(2\sqrt{d} + \sqrt{C_1 k \log T}\right)\varepsilon \\
&\overset{(iii)}{\le} \sqrt{C_1 k \log T}\|x - x'\|_2 + 2\sqrt{C_1 k \log T}\varepsilon + 2\left(2\sqrt{d} + \sqrt{C_1 k \log T}\right)\varepsilon \\
&\le \sqrt{C_1 k \log T}\|x - x'\|_2 + \left(4\sqrt{d} + 4\sqrt{C_1 k \log T}\right)\varepsilon. 
\end{aligned} \tag{A.5}
$$

Here step (i) follows from $\omega \in \mathcal{G}$ and the Cauchy-Schwarz inequality; steps (ii) and (iii) follows from $\|x - x_i^\star\|_2 \le \varepsilon$ and $\|x' - x_j^\star\|_2 \le \varepsilon$, as well as $\|\omega\|_2 \le \sqrt{d} + \sqrt{C_1 k \log T}$, which is a property for $\omega \in \mathcal{G}$.

## A.2  Understanding the conditional density $p_{X_t | X_0}(\cdot \,|\, x_0)$

Conditional on $X_t = x_t$, for any $1 \le i \le N_\varepsilon$ we have

$$
\begin{aligned}
\mathbb{P}\left(X_0 \in \mathcal{B}_i \,|\, X_t = x_t\right) &= \frac{\mathbb{P}\left(X_0 \in \mathcal{B}_i, X_t = x_t\right)}{p_{X_t}(x_t)} = \frac{\mathbb{P}\left(X_0 \in \mathcal{B}_i, X_t = x_t\right)}{\sum_{1 \le j \le N_\varepsilon} \mathbb{P}\left(X_0 \in \mathcal{B}_j, X_t = x_t\right)} \\
&\le \frac{\mathbb{P}\left(X_0 \in \mathcal{B}_i, X_t = x_t\right)}{\mathbb{P}\left(X_0 \in \mathcal{B}_{i(x_t)}, X_t = x_t\right)} = \frac{\mathbb{P}\left(X_0 \in \mathcal{B}_i\right)\mathbb{P}\left(X_t = x_t \,|\, X_0 \in \mathcal{B}_i\right)}{\mathbb{P}\left(X_0 \in \mathcal{B}_{i(x_t)}\right)\mathbb{P}\left(X_t = x_t \,|\, X_0 \in \mathcal{B}_{i(x_t)}\right)} \\
&\le \exp\left(C_1 k \log T\right) \cdot \frac{\mathbb{P}\left(X_t = x_t \,|\, X_0 \in \mathcal{B}_i\right)}{\mathbb{P}\left(X_t = x_t \,|\, X_0 \in \mathcal{B}_{i(x_t)}\right)} \cdot \mathbb{P}\left(X_0 \in \mathcal{B}_i\right).
\end{aligned} \tag{A.6}
$$

Here the last relation follows from $\mathbb{P}(X_0 \in \mathcal{B}_{i(x_t)}) \geq \exp(-C_1 k \log T)$ due to $i(x_t) \in \mathcal{I}$. We have

$$\mathbb{P}\left(X_t = x_t \mid X_0 \in \mathcal{B}_i\right) = \frac{\mathbb{P}\left(X_t = x_t, X_0 \in \mathcal{B}_i\right)}{\mathbb{P}\left(X_0 \in \mathcal{B}_i\right)} = \frac{1}{\mathbb{P}\left(X_0 \in \mathcal{B}_i\right)} \int_{\tilde{x} \in \mathcal{B}_i} \mathbb{P}\left(X_t = x_t, X_0 = \tilde{x}\right) d\tilde{x}$$

$$= \frac{1}{\mathbb{P}\left(X_0 \in \mathcal{B}_i\right)} \int_{\tilde{x} \in \mathcal{B}_i} \mathbb{P}\left(X_t = x_t \mid X_0 = \tilde{x}\right) \mathbb{P}\left(X_0 = \tilde{x}\right) d\tilde{x}$$

$$\leq \max_{\tilde{x} \in \mathcal{B}_i} \mathbb{P}\left(X_t = x_t \mid X_0 = \tilde{x}\right). \tag{A.7}$$

For any $\tilde{x} \in \mathcal{B}_i$, since $X_t \mid X_0 = \tilde{x} \sim \mathcal{N}(\sqrt{\overline{\alpha}_t}\tilde{x}, (1-\overline{\alpha}_t)I_d)$, we have

$$\mathbb{P}\left(X_t = x_t \mid X_0 = \tilde{x}\right) = [2\pi (1-\overline{\alpha}_t)]^{-d/2} \exp\left(-\frac{\|x_t - \sqrt{\overline{\alpha}_t}\tilde{x}\|_2^2}{2(1-\overline{\alpha}_t)}\right)$$

$$\leq [2\pi (1-\overline{\alpha}_t)]^{-d/2} \exp\left(-\frac{(\|x_t - \sqrt{\overline{\alpha}_t}x_i^\star\|_2 - \sqrt{\overline{\alpha}_t}\varepsilon)^2}{2(1-\overline{\alpha}_t)}\right). \tag{A.8}$$

Taking (A.7) and (A.8) collectively to achieve

$$\mathbb{P}\left(X_t = x_t \mid X_0 \in \mathcal{B}_i\right) \leq [2\pi (1-\overline{\alpha}_t)]^{-d/2} \exp\left(-\frac{(\|x_t - \sqrt{\overline{\alpha}_t}x_i^\star\|_2 - \sqrt{\overline{\alpha}_t}\varepsilon)^2}{2(1-\overline{\alpha}_t)}\right). \tag{A.9}$$

By similar argument in (A.7), (A.8) and (A.9), we can show that

$$\mathbb{P}\left(X_t = x_t \mid X_0 \in \mathcal{B}_{i(x_t)}\right) \geq [2\pi (1-\overline{\alpha}_t)]^{-d/2} \exp\left(-\frac{(\|x_t - \sqrt{\overline{\alpha}_t}x_{i(x_t)}^\star\|_2 + \sqrt{\overline{\alpha}_t}\varepsilon)^2}{2(1-\overline{\alpha}_t)}\right). \tag{A.10}$$

Combine (A.9) and (A.10) to achieve

$$\frac{\mathbb{P}\left(X_t = x_t \mid X_0 \in \mathcal{B}_i\right)}{\mathbb{P}\left(X_t = x_t \mid X_0 \in \mathcal{B}_{i(x_t)}\right)} \tag{A.11}$$

$$\leq \exp\left[-\frac{(\|x_t - \sqrt{\overline{\alpha}_t}x_i^\star\|_2 - \sqrt{\overline{\alpha}_t}\varepsilon)^2}{2(1-\overline{\alpha}_t)} + \frac{(\|x_t - \sqrt{\overline{\alpha}_t}x_{i(x_t)}^\star\|_2 + \sqrt{\overline{\alpha}_t}\varepsilon)^2}{2(1-\overline{\alpha}_t)}\right]$$

$$\leq \exp\left[-\frac{\|x_t - \sqrt{\overline{\alpha}_t}x_i^\star\|_2^2 - \|x_t - \sqrt{\overline{\alpha}_t}x_{i(x_t)}^\star\|_2^2 - 2\sqrt{\overline{\alpha}_t}\varepsilon(\|x_t - \sqrt{\overline{\alpha}_t}x_i^\star\|_2 + \|x_t - \sqrt{\overline{\alpha}_t}x_{i(x_t)}^\star\|_2)}{2(1-\overline{\alpha}_t)}\right].$$

Next, we will discuss the implication of the above analysis for any $i \notin \mathcal{I}(x_t; C_3)$, i.e., $\mathcal{B}_i \subseteq \mathcal{Y}_t(x_t)$.
For any $i \notin \mathcal{I}(x_t; C_3)$, we have

$$\|x_t - \sqrt{\overline{\alpha}_t}x_i^\star\|_2^2 - \|x_t - \sqrt{\overline{\alpha}_t}x_{i(x_t)}^\star\|_2^2 = \overline{\alpha}_t\|x_{i(x_t)}^\star - x_i^\star\|_2^2 + 2\sqrt{\overline{\alpha}_t}(x_{i(x_t)}^\star - x_i^\star)^\top(x_t - \sqrt{\overline{\alpha}_t}x_{i(x_t)}^\star)$$

$$\overset{(i)}{=} \overline{\alpha}_t\|x_{i(x_t)}^\star - x_i^\star\|_2^2 + 2\sqrt{\overline{\alpha}_t(1-\overline{\alpha}_t)}(x_{i(x_t)}^\star - x_i^\star)^\top\omega + 2\overline{\alpha}_t\sqrt{1-\overline{\alpha}_t}(x_{i(x_t)}^\star - x_i^\star)^\top(x_0(x_t) - x_{i(x_t)}^\star)$$

$$\overset{(ii)}{\geq} \overline{\alpha}_t\|x_{i(x_t)}^\star - x_i^\star\|_2^2 - 2\sqrt{C_1\overline{\alpha}_t(1-\overline{\alpha}_t)}\sqrt{k \log T}\|x_{i(x_t)}^\star - x_i^\star\|_2 - 2\overline{\alpha}_t\sqrt{1-\overline{\alpha}_t}\varepsilon\|x_{i(x_t)}^\star - x_i^\star\|_2$$

$$\overset{(iii)}{\geq} \frac{1}{2}\overline{\alpha}_t\|x_{i(x_t)}^\star - x_i^\star\|_2^2. \tag{A.12}$$

Here step (i) uses the decomposition (A.1); step (ii) follows from $\omega \in \mathcal{G}$, Cauchy-Schwarz inequality and the fact that $\|x_0(x_t) - x_{i(x_t)}^\star\|_2 \leq \varepsilon$; while the correctness of step (iii) is equivalent to

$$\sqrt{\overline{\alpha}_t}\|x_{i(x_t)}^\star - x_i^\star\|_2 \geq 4\sqrt{C_1(1-\overline{\alpha}_t)}\sqrt{k \log T} + 4\sqrt{\overline{\alpha}_t(1-\overline{\alpha}_t)}\varepsilon,$$

which follows from $i \notin \mathcal{I}(x_t; C_3)$, the assumptions that $C_3 \gg C_1$, and (A.3). In addition, we also have

$$\|x_t - \sqrt{\overline{\alpha}_t}x_i^\star\|_2 + \|x_t - \sqrt{\overline{\alpha}_t}x_{i(x_0)}^\star\|_2$$

$$\overset{(i)}{\leq} \sqrt{\overline{\alpha}_t}\|x_0(x_t) - x_i^\star\|_2 + \sqrt{\overline{\alpha}_t}\|x_0(x_t) - x_{i(x_t)}^\star\|_2 + 2\sqrt{1-\overline{\alpha}_t}\|\omega\|_2$$

$$\overset{(ii)}{\leq} \sqrt{\overline{\alpha}_t}\|x_{i(x_t)}^\star - x_i^\star\|_2 + 2\sqrt{\overline{\alpha}_t}\|x_0(x_t) - x_{i(x_t)}^\star\|_2 + 2\sqrt{1-\overline{\alpha}_t}\left(2\sqrt{d} + \sqrt{C_1 k \log T}\right)$$

$$\overset{(iii)}{\leq} \sqrt{\overline{\alpha}_t}\|x^{\star}_{i(x_t)} - x^{\star}_i\|_2 + 3\sqrt{1 - \overline{\alpha}_t}(2\sqrt{d} + \sqrt{C_1 k \log T}). \tag{A.13}$$

Here step (i) follows from the decomposition (A.1); step (ii) utilizes the triangle inequality; whereas step (iii) follows from $\|x_0(x_t) - x^{\star}_{i(x_t)}\|_2 \leq \varepsilon$ and the condition (A.3). We can substitute the bounds (A.12) and (A.13) into (A.11) to get

$$\frac{\mathbb{P}(X_t = x_t \mid X_0 \in \mathcal{B}_i)}{\mathbb{P}(X_t = x_t \mid X_0 \in \mathcal{B}_{i(x_t)})} \leq \exp\left(-\frac{\overline{\alpha}_t}{8(1 - \overline{\alpha}_t)}\|x^{\star}_{i(x_t)} - x^{\star}_i\|_2^2\right), \tag{A.14}$$

provided that (A.3) holds and $C_3$ is sufficiently large. Since $i \notin \mathcal{I}(x_t; C_3)$, we know that $\overline{\alpha}_t\|x^{\star}_{i(x_t)} - x^{\star}_i\|_2^2 > C_3 k(1 - \overline{\alpha}_t)\log T$, hence when $C_3 \gg C_1$, we learn from (A.6) and (A.14) that

$$\mathbb{P}\left(X_0 \in \mathcal{B}_i \mid X_t = x_t\right) \leq \exp\left(C_1 k \log T - \frac{\overline{\alpha}_t}{8(1 - \overline{\alpha}_t)}\|x^{\star}_{i(x_t)} - x^{\star}_i\|_2^2\right)\mathbb{P}\left(X_0 \in \mathcal{B}_i\right)$$

$$\leq \exp\left(-\frac{\overline{\alpha}_t}{16(1 - \overline{\alpha}_t)}\|x^{\star}_{i(x_t)} - x^{\star}_i\|_2^2\right)\mathbb{P}\left(X_0 \in \mathcal{B}_i\right). \tag{A.15}$$

### A.3 Proof of Lemma 1

It is straightforward to check that

$$\mathbb{P}\left((X_t, X_{t-1}) \in \mathcal{A}_t\right) \overset{(i)}{=} \mathbb{P}\left(X_t \in \mathcal{T}_t, W_t \in \mathcal{G}\right) \overset{(ii)}{\leq} \mathbb{P}\left(X_0 \in \cup_{i \in \mathcal{I}}\mathcal{B}_i, W_t \in \mathcal{G}, \overline{W}_t \in \mathcal{G}\right),$$

where step (i) follows from the update rule (2.1), and step (ii) follows from the relation (2.2). Therefore we have

$$\mathbb{P}\left((X_t, X_{t-1}) \notin \mathcal{A}_t\right) \leq \mathbb{P}\left(X_0 \notin \cup_{i \in \mathcal{I}}\mathcal{B}_i\right) + \mathbb{P}\left(W_t \notin \mathcal{G}\right) + \mathbb{P}\left(\overline{W}_t \notin \mathcal{G}\right). \tag{A.16}$$

By definition of the set $\mathcal{I}$, we have

$$\mathbb{P}\left(X_0 \notin \cup_{i \in \mathcal{I}}\mathcal{B}_i\right) \leq N_\varepsilon \exp\left(-C_1 k \log T\right) \leq \exp\left(C_{\mathsf{cover}} k \log T - C_1 k \log T\right)$$

$$\leq \frac{1}{3}\exp\left(-\frac{C_1}{4}k \log T\right) \tag{A.17}$$

as long as $C_1 \gg C_{\mathsf{cover}}$. In addition, since $W_t, \overline{W}_t \sim \mathcal{N}(0, I_d)$, by the definition of $\mathcal{G}$ we know that

$$\mathbb{P}\left(W_t \notin \mathcal{G}\right) \leq \mathbb{P}\left(\|W_t\|_2 > \sqrt{d} + \sqrt{C_1 k \log T}\right) + \sum_{i=1}^{N_\varepsilon}\sum_{j=1}^{N_\varepsilon}\mathbb{P}\left(|(x^{\star}_i - x^{\star}_j)^{\top}W_t| > \sqrt{C_1 k \log T}\|x^{\star}_i - x^{\star}_j\|_2\right)$$

$$\overset{(i)}{\leq} \left(N_\varepsilon^2 + 1\right)\exp\left(-\frac{C_1}{2}k \log T\right) \leq \left(\exp\left(2C_{\mathsf{cover}}k \log T\right) + 1\right)\exp\left(-\frac{C_1}{2}k \log T\right)$$

$$\overset{(ii)}{\leq} \frac{1}{3}\exp\left(-\frac{C_1}{4}k \log T\right) \tag{A.18}$$

Here step (i) follows from concentration bounds for Gaussian and chi-square variables (see Lemma 9); while step (ii) holds as long as $C_1 \gg C_{\mathsf{cover}}$. The same bound also holds for $\mathbb{P}(\overline{W}_t \notin \mathcal{G})$. Taking (A.16), (A.17) and (A.18) collectively yields

$$\mathbb{P}\left((X_t, X_{t-1}) \notin \mathcal{A}_t\right) \leq \exp\left(-\frac{C_1}{4}k \log T\right)$$

as claimed.

### A.4 Proof of Lemma 2

For any deterministic pairs $(x_t, x_{t-1})$, we have

$$p_{X_{t-1}|X_t}\left(x_{t-1} \mid x_t\right) = \frac{1}{p_{X_t}\left(x_t\right)}p_{X_{t-1}, X_t}\left(x_{t-1}, x_t\right) = \frac{p_{X_{t-1}}\left(x_{t-1}\right)}{p_{X_t}\left(x_t\right)}p_{X_t|X_{t-1}}\left(x_t \mid x_{t-1}\right). \tag{A.19}$$

Recall that $X_t \mid X_{t-1} = x_{t-1} \sim \mathcal{N}(\sqrt{\alpha_t}x_{t-1}, (1-\alpha_t)I_d)$, therefore we have

$$p_{X_t \mid X_{t-1}}(x_t \mid x_{t-1}) = [2\pi(1-\alpha_t)]^{-d/2}\exp\left(-\frac{1}{2(1-\alpha_t)}\|x_t - \sqrt{\alpha_t}x_{t-1}\|_2^2\right). \qquad (A.20)$$

Next, we analyze the density ratio $p_{X_{t-1}}(x_{t-1})/p_{X_t}(x_t)$. It would be easier to do a change of variable

$$p_{X_{t-1}}(x_{t-1}) = \alpha_t^{d/2} p_{\sqrt{\alpha_t}X_{t-1}}(\sqrt{\alpha_t}x_{t-1}). \qquad (A.21)$$

Since $\sqrt{\alpha_t}X_{t-1} \mid X_0 = x_0 \sim \mathcal{N}(\sqrt{\overline{\alpha}_t}x_0, (\alpha_t - \overline{\alpha}_t)I_d)$, we can write

$$\frac{p_{\sqrt{\alpha_t}X_{t-1}}(\sqrt{\alpha_t}x_{t-1})}{p_{X_t}(x_t)} = \frac{1}{p_{X_t}(x_t)}\int_{x_0} p_{X_0}(x_0) p_{\sqrt{\alpha_t}X_{t-1}\mid X_0}(\sqrt{\alpha_t}x_{t-1}\mid x_0)\,\mathrm{d}x_0$$

$$= \frac{1}{p_{X_t}(x_t)}\int_{x_0} p_{X_0}(x_0)[2\pi(\alpha_t-\overline{\alpha}_t)]^{-d/2}\exp\left(-\frac{\|\sqrt{\alpha_t}x_{t-1}-\sqrt{\overline{\alpha}_t}x_0\|_2^2}{2(\alpha_t-\overline{\alpha}_t)}\right)\mathrm{d}x_0. \quad (A.22)$$

We hope to connect the above quantity with the conditional density

$$p_{X_0 \mid X_t}(x_0 \mid x_t) = \frac{p_{X_0, X_t}(x_0, x_t)}{p_{X_t}(x_t)} = \frac{p_{X_0}(x_0)}{p_{X_t}(x_t)}p_{X_t \mid X_0}(x_t \mid x_0)$$

$$= \frac{p_{X_0}(x_0)}{p_{X_t}(x_t)}\frac{1}{[2\pi(1-\overline{\alpha}_t)]^{d/2}}\exp\left(-\frac{\|x_t - \sqrt{\overline{\alpha}_t}x_0\|_2^2}{2(1-\overline{\alpha}_t)}\right). \qquad (A.23)$$

Towards this, we can deduce that

$$\frac{p_{\sqrt{\alpha_t}X_{t-1}}(\sqrt{\alpha_t}x_{t-1})}{p_{X_t}(x_t)} \qquad\qquad\qquad\qquad\qquad\qquad\qquad\qquad\qquad (A.24)$$

$$\overset{\text{(i)}}{=} \left(\frac{1-\overline{\alpha}_t}{\alpha_t-\overline{\alpha}_t}\right)^{d/2}\int_{x_0}\frac{p_{X_0}(x_0)}{p_{X_t}(x_t)}[2\pi(1-\overline{\alpha}_t)]^{-d/2}\exp\left(-\frac{\|\sqrt{\alpha_t}x_{t-1}-\sqrt{\overline{\alpha}_t}x_0\|_2^2}{2(\alpha_t-\overline{\alpha}_t)}\right)\mathrm{d}x_0$$

$$\overset{\text{(ii)}}{=} \left(\frac{1-\overline{\alpha}_t}{\alpha_t-\overline{\alpha}_t}\right)^{d/2}\int_{x_0}p_{X_0\mid X_t}(x_0 \mid x_t)\exp\left(\frac{\|x_t-\sqrt{\overline{\alpha}_t}x_0\|_2^2}{2(1-\overline{\alpha}_t)}-\frac{\|\sqrt{\alpha_t}x_{t-1}-\sqrt{\overline{\alpha}_t}x_0\|_2^2}{2(\alpha_t-\overline{\alpha}_t)}\right)\mathrm{d}x_0,$$

where step (i) follows from (A.22) and step (ii) utilizes (A.23). The terms in the exponent can be rearranged into

$$\frac{\|x_t - \sqrt{\overline{\alpha}_t}x_0\|_2^2}{2(1-\overline{\alpha}_t)} - \frac{\|\sqrt{\alpha_t}x_{t-1}-\sqrt{\overline{\alpha}_t}x_0\|_2^2}{2(\alpha_t-\overline{\alpha}_t)}$$

$$= \frac{\|x_t - \sqrt{\overline{\alpha}_t}x_0\|_2^2 - \|\sqrt{\alpha_t}x_{t-1}-\sqrt{\overline{\alpha}_t}x_0\|_2^2}{2(\alpha_t-\overline{\alpha}_t)} - \frac{(1-\alpha_t)\|x_t-\sqrt{\overline{\alpha}_t}x_0\|_2^2}{2(\alpha_t-\overline{\alpha}_t)(1-\overline{\alpha}_t)}$$

$$= -\frac{\|\sqrt{\alpha_t}x_{t-1}-x_t\|_2^2 + 2(\sqrt{\alpha_t}x_{t-1}-x_t)^\top(x_t-\sqrt{\overline{\alpha}_t}x_0)}{2(\alpha_t-\overline{\alpha}_t)} - \frac{(1-\alpha_t)\|x_t-\sqrt{\overline{\alpha}_t}x_0\|_2^2}{2(\alpha_t-\overline{\alpha}_t)(1-\overline{\alpha}_t)}$$

$$= -\frac{\|\sqrt{\alpha_t}x_{t-1}-x_t\|_2^2 + 2(\sqrt{\alpha_t}x_{t-1}-x_t)^\top(x_t-\sqrt{\overline{\alpha}_t}\widehat{x}_0)}{2(\alpha_t-\overline{\alpha}_t)} - \frac{(1-\alpha_t)\|x_t-\sqrt{\overline{\alpha}_t}\widehat{x}_0\|_2^2}{2(\alpha_t-\overline{\alpha}_t)(1-\overline{\alpha}_t)} + \Delta_{x_t, x_{t-1}}(x_0)$$

where we define

$$\widehat{x}_0 := \mathbb{E}[X_0 \mid X_t = x_t] = \int_{x_0} x_0 p_{X_0 \mid X_t}(x_0 \mid x_t)\,\mathrm{d}x_0,$$

and

$$\Delta_{x_t, x_{t-1}}(x_0) := -\frac{\sqrt{\alpha_t}}{\alpha_t-\overline{\alpha}_t}(\sqrt{\alpha_t}x_{t-1}-x_t)^\top(\widehat{x}_0-x_0)$$

$$-\frac{(1-\alpha_t)\sqrt{\overline{\alpha}_t}}{(\alpha_t-\overline{\alpha}_t)(1-\overline{\alpha}_t)}(x_t-\sqrt{\overline{\alpha}_t}\widehat{x}_0)^\top(\widehat{x}_0-x_0) - \frac{(1-\alpha_t)\overline{\alpha}_t}{2(\alpha_t-\overline{\alpha}_t)(1-\overline{\alpha}_t)}\|\widehat{x}_0-x_0\|_2^2.$$

Substituting the above relation into (A.24) yields

$$\frac{p_{\sqrt{\alpha_t}X_{t-1}}(\sqrt{\alpha_t}x_{t-1})}{p_{X_t}(x_t)} = \left(\frac{1-\overline{\alpha}_t}{\alpha_t-\overline{\alpha}_t}\right)^{d/2}\exp\left[-\frac{\|\sqrt{\alpha_t}x_{t-1}-x_t\|_2^2}{2(\alpha_t-\overline{\alpha}_t)}\right]$$

$$\cdot \exp\left[-\frac{\left(\sqrt{\alpha_t}x_{t-1} - x_t\right)^\top \left(x_t - \sqrt{\overline{\alpha}_t}\widehat{x}_0\right)}{\alpha_t - \overline{\alpha}_t} - \frac{(1-\alpha_t)\left\|x_t - \sqrt{\overline{\alpha}_t}\widehat{x}_0\right\|_2^2}{2\left(\alpha_t - \overline{\alpha}_t\right)\left(1 - \overline{\alpha}_t\right)}\right]$$

$$\cdot \int_{x_0} p_{X_0 \mid X_t}\left(x_0 \mid x_t\right) \exp\left(\Delta_{x_t, x_{t-1}}\left(x_0\right)\right) \mathrm{d}x_0. \tag{A.25}$$

Therefore we have

$$p_{X_{t-1} \mid X_t}\left(x_{t-1} \mid x_t\right) \overset{\text{(i)}}{=} \alpha_t^{d/2} \frac{p_{\sqrt{\alpha_t}X_{t-1}}\left(\sqrt{\alpha_t}x_{t-1}\right)}{p_{X_t}\left(x_t\right)} p_{X_t \mid X_{t-1}}\left(x_t \mid x_{t-1}\right) \tag{A.26}$$

$$\overset{\text{(ii)}}{=} \alpha_t^{d/2} \left(\frac{1 - \overline{\alpha}_t}{\alpha_t - \overline{\alpha}_t}\right)^{d/2} \exp\left(-\frac{\left(\sqrt{\alpha_t}x_{t-1} - x_t\right)^\top \left(x_t - \sqrt{\overline{\alpha}_t}\widehat{x}_0\right)}{\alpha_t - \overline{\alpha}_t} - \frac{(1-\alpha_t)\left\|x_t - \sqrt{\overline{\alpha}_t}\widehat{x}_0\right\|_2^2}{2\left(\alpha_t - \overline{\alpha}_t\right)\left(1 - \overline{\alpha}_t\right)}\right)$$

$$\cdot \left[2\pi\left(1 - \alpha_t\right)\right]^{-d/2} \exp\left(-\frac{(1-\overline{\alpha}_t)\left\|x_t - \sqrt{\alpha_t}x_{t-1}\right\|_2^2}{2\left(1 - \alpha_t\right)\left(\alpha_t - \overline{\alpha}_t\right)}\right) \int_{x_0} p_{X_0 \mid X_t}\left(x_0 \mid x_t\right) \exp\left(\Delta_{x_t, x_{t-1}}\left(x_0\right)\right) \mathrm{d}x_0$$

$$\overset{\text{(iii)}}{=} \frac{\alpha_t^{d/2}}{\left(2\pi\sigma_t^{\star 2}\right)^{d/2}} \exp\left(-\frac{\left\|\sqrt{\alpha_t}x_{t-1} - x_t - \eta_t^\star s_t^\star\left(x_t\right)\right\|_2^2}{2\sigma_t^{\star 2}}\right) \int_{x_0} p_{X_0 \mid X_t}\left(x_0 \mid x_t\right) \exp\left(\Delta_{x_t, x_{t-1}}\left(x_0\right)\right) \mathrm{d}x_0.$$

Here step (i) follows from (A.19) and (A.21); step (ii) follows from (A.20) and (A.25); whereas step (iii) follows from the definition of $\eta_t^\star$ and $\sigma_t^\star$ (cf. (2.4)) as well as the fact that

$$s_t^\star\left(x_t\right) = -\frac{1}{1 - \overline{\alpha}_t} \int_{x_0} p_{X_0 \mid X_t}\left(x_0 \mid x_t\right)\left(x - \sqrt{\overline{\alpha}_t}x_0\right)\mathrm{d}x_0 = -\frac{1}{1 - \overline{\alpha}_t}\left(x_t - \sqrt{\overline{\alpha}_t}\widehat{x}_0\right).$$

## A.5 Proof of Lemma 3

For any $(x_t, x_{t-1}) \in \mathcal{A}_t$, we know that $\omega' := (x_t - \sqrt{\alpha_t}x_{t-1})/\sqrt{1 - \alpha_t} \in \mathcal{G}$. We will upper bound the integral with two terms

$$\int_{x_0} p_{X_0 \mid X_t}\left(x_0 \mid x_t\right) \exp\left(\Delta\left(x_t, x_{t-1}, x_0\right)\right) \mathrm{d}x_0 = \underbrace{\int_{\mathcal{X}_t(x_t)} p_{X_0 \mid X_t}\left(x_0 \mid x_t\right) \exp\left(\Delta\left(x_t, x_{t-1}, x_0\right)\right) \mathrm{d}x_0}_{=:I_1}$$

$$+ \underbrace{\int_{\mathcal{Y}_t(x_t)} p_{X_0 \mid X_t}\left(x_0 \mid x_t\right) \exp\left(\Delta\left(x_t, x_{t-1}, x_0\right)\right) \mathrm{d}x_0}_{=:I_2},$$

where we recall that

$$\Delta\left(x_t, x_{t-1}, x_0\right) = \underbrace{\frac{\sqrt{\overline{\alpha}_t}\left(1 - \alpha_t\right)}{\alpha_t - \overline{\alpha}_t}\left(\widehat{x}_0 - x_0\right)^\top \omega'}_{=:\Delta_1(x_0)} - \underbrace{\frac{(1-\alpha_t)\sqrt{\overline{\alpha}_t}}{\left(\alpha_t - \overline{\alpha}_t\right)\sqrt{1 - \overline{\alpha}_t}}\left(\widehat{x}_0 - x_0\right)^\top \omega}_{=:\Delta_2(x_0)}$$

$$- \underbrace{\frac{(1-\alpha_t)\overline{\alpha}_t}{\left(\alpha_t - \overline{\alpha}_t\right)\left(1 - \overline{\alpha}_t\right)}\left(x_0(x_t) - \widehat{x}_0\right)^\top\left(\widehat{x}_0 - x_0\right)}_{=:\Delta_3(x_0)} - \underbrace{\frac{(1-\alpha_t)\overline{\alpha}_t}{2\left(\alpha_t - \overline{\alpha}_t\right)\left(1 - \overline{\alpha}_t\right)}\left\|\widehat{x}_0 - x_0\right\|_2^2}_{=:\Delta_4(x_0)}.$$

In what follows, we will use $\Delta(x_0)$ instead of $\Delta\left(x_t, x_{t-1}, x_0\right)$ when there is no confusion. Since $(x_t, x_{t-1}) \in \mathcal{A}_t$, we know that $\omega' := (x_t - \sqrt{\alpha_t}x_{t-1})/\sqrt{1 - \alpha_t} \in \mathcal{G}$. We decompose $\widehat{x}_0$ into

$$\widehat{x}_0 = \int_{x_0} x_0' p_{X_0 \mid X_t}\left(x_0' \mid x_t\right) \mathrm{d}x_0' \tag{A.27}$$

$$= \underbrace{x_{i(x_t)}^\star + \int_{\mathcal{X}_t(x_t)}\left(x_0' - x_{i(x_t)}^\star\right)p_{X_0 \mid X_t}\left(x_0' \mid x_t\right)\mathrm{d}x_0'}_{=:\overline{x}_0} + \underbrace{\int_{\mathcal{Y}_t(x_t)}\left(x_0' - x_{i(x_t)}^\star\right)p_{X_0 \mid X_t}\left(x_0' \mid x_t\right)\mathrm{d}x_0'}_{=:\delta}.$$

Since $\mathcal{X}_t(x_t)$ is a ball in $\mathbb{R}^d$ centered at $x_{i(x_t)}^\star$, it is straightforward to check that $\overline{x}_0 \in \mathcal{X}_t(x_t)$. We also have

$$\|\delta\|_2 \leq \int_{\mathcal{Y}_t(x_t)}\left\|x_0' - x_{i(x_t)}^\star\right\|_2 p_{X_0 \mid X_t}\left(x_0 \mid x_t\right)\mathrm{d}x_0'$$

$$\overset{(i)}{\leq} \sum_{i \notin \mathcal{I}(x_t; C_3)} \left( \|x_i^\star - x_{i(x_t)}^\star\|_2 + \varepsilon \right) \mathbb{P}\left( X_0 \in \mathcal{B}_i \mid X_t = x_t \right)$$

$$\overset{(ii)}{\leq} \sum_{i \notin \mathcal{I}(x_t; C_3)} \left( \|x_i^\star - x_{i(x_t)}^\star\|_2 + \varepsilon \right) \exp\left( -\frac{\overline{\alpha}_t}{16(1 - \overline{\alpha}_t)} \|x_{i(x_t)}^\star - x_i^\star\|_2^2 \right) \mathbb{P}\left( X_0 \in \mathcal{B}_i \right).$$

Here step (i) holds since for any $i \notin \mathcal{I}(x_t; C_3)$ and $x_0' \in \mathcal{B}_i$,

$$\|x_0' - x_{i(x_t)}^\star\|_2 \leq \|x_i^\star - x_{i(x_t)}^\star\|_2 + \|x_0' - x_i^\star\|_2 \leq \|x_i^\star - x_{i(x_t)}^\star\|_2 + \varepsilon;$$

while step (ii) follows from (A.15). For any $i \notin \mathcal{I}(x_t; C_3)$, we know that $\overline{\alpha}_t \|x_{i(x_t)}^\star - x_i^\star\|_2^2 > C_3 k(1 - \overline{\alpha}_t) \log T$, hence we can check that

$$\left( \|x_i^\star - x_{i(x_t)}^\star\|_2 + \varepsilon \right) \exp\left( -\frac{\overline{\alpha}_t}{16(1 - \overline{\alpha}_t)} \|x_{i(x_t)}^\star - x_i^\star\|_2^2 \right)$$

$$\leq \left( \sqrt{\frac{C_3 k(1 - \overline{\alpha}_t) \log T}{\overline{\alpha}_t}} + \varepsilon \right) \exp\left( -\frac{C_3 k \log T}{16} \right) \leq \sqrt{\frac{1 - \overline{\alpha}_t}{\overline{\alpha}_t}} \exp\left( -\frac{C_3 k \log T}{32} \right)$$

as long as $C_3$ is sufficiently large and the condition (A.3) holds. Therefore

$$\|\delta\|_2 \leq \sum_{i \notin \mathcal{I}(x_t; C_3)} \sqrt{\frac{1 - \overline{\alpha}_t}{\overline{\alpha}_t}} \exp\left( -\frac{C_3 k \log T}{32} \right) \mathbb{P}\left( X_0 \in \mathcal{B}_i \right)$$

$$\leq \sqrt{\frac{1 - \overline{\alpha}_t}{\overline{\alpha}_t}} \exp\left( -\frac{C_3 k \log T}{32} \right). \tag{A.28}$$

### A.5.1 Step 1: deriving an upper bound for $\Delta(x_0)$

Suppose that $x_0 \in \mathcal{B}_i$ for some $1 \leq i \leq N_\varepsilon$ (notice that here we are not requiring that $i \in \mathcal{I}$). We will bound each of $|\Delta_i(x_0)|$ for $i = 1, 2, 3, 4$. We first record two basic facts about the step sizes, which are immediate consequences of Lemma 8:

$$\frac{\sqrt{\overline{\alpha}_t(1 - \alpha_t)}}{\alpha_t - \overline{\alpha}_t} = \sqrt{\frac{\overline{\alpha}_t}{1 - \overline{\alpha}_t}} \sqrt{1 + \frac{1 - \alpha_t}{\alpha_t - \overline{\alpha}_t}} \sqrt{\frac{1 - \alpha_t}{\alpha_t - \overline{\alpha}_t}} \leq \sqrt{\frac{\overline{\alpha}_t}{1 - \overline{\alpha}_t}} \sqrt{1 + \frac{8 c_1 \log T}{T}} \sqrt{\frac{8 c_1 \log T}{T}}$$

$$\leq 3 \sqrt{\frac{c_1 \log T}{T}} \sqrt{\frac{\overline{\alpha}_t}{1 - \overline{\alpha}_t}}, \tag{A.29}$$

as long as $T$ is sufficiently large, and

$$\frac{(1 - \alpha_t) \sqrt{\overline{\alpha}_t}}{(\alpha_t - \overline{\alpha}_t) \sqrt{1 - \overline{\alpha}_t}} \leq \frac{8 c_1 \log T}{T} \sqrt{\frac{\overline{\alpha}_t}{1 - \overline{\alpha}_t}}. \tag{A.30}$$

We learn from (A.5) that

$$\max\left\{ \left| (\widehat{x}_0 - x_0)^\top \omega \right|, \left| (\widehat{x}_0 - x_0)^\top \omega' \right| \right\} \leq \sqrt{C_1 k \log T} \|\widehat{x}_0 - x_0\|_2 + \left( 4\sqrt{d} + 4\sqrt{C_1 k \log T} \right) \varepsilon. \tag{A.31}$$

We also have

$$\|\widehat{x}_0 - x_0\|_2 \leq \|\overline{x}_0 - x_0\|_2 + \|\delta\|_2 \overset{(i)}{\leq} \|x_{i(x_t)}^\star - x_i^\star\|_2 + \|x_{i(x_t)}^\star - \overline{x}_0\|_2 + \varepsilon + \|\delta\|_2$$

$$\overset{(ii)}{\leq} \|x_{i(x_t)}^\star - x_i^\star\|_2 + 3\sqrt{\frac{C_3 k(1 - \overline{\alpha}_t) \log T}{\overline{\alpha}_t}} + \varepsilon + \sqrt{\frac{1 - \overline{\alpha}_t}{\overline{\alpha}_t}} \exp\left( -\frac{C_3 k \log T}{32} \right)$$

$$\overset{(iii)}{\leq} \|x_{i(x_t)}^\star - x_i^\star\|_2 + 4\sqrt{\frac{C_3 k(1 - \overline{\alpha}_t) \log T}{\overline{\alpha}_t}}. \tag{A.32}$$

Here step (i) holds since $x_0 \in \mathcal{B}_i$, hence $\|x_0 - x_i^\star\|_2 \leq \varepsilon$; step (ii) follows from (A.4) and the fact that $x_{i(x_t)}^\star, \overline{x}_0 \in \mathcal{X}_t(x_t)$; while step (iii) follows from (A.3) and holds provided that $C_3$ is sufficiently large. Then we have

$$|\Delta_1(x_0)| \leq \frac{\sqrt{\overline{\alpha}_t(1 - \alpha_t)}}{\alpha_t - \overline{\alpha}_t} \left| (\widehat{x}_0 - x_0)^\top \omega \right| \tag{A.33a}$$

$$\overset{(a)}{\leq} 3\sqrt{\frac{c_1 \log T}{T}}\sqrt{\frac{\overline{\alpha}_t}{1-\overline{\alpha}_t}}\left(\sqrt{C_1 k \log T}\|\widehat{x}_0 - x_0\|_2 + \left(4\sqrt{d} + 4\sqrt{C_1 k \log T}\right)\varepsilon\right)$$

$$\overset{(b)}{\leq} 4\sqrt{\frac{c_1 \log T}{T}}\sqrt{\frac{\overline{\alpha}_t}{1-\overline{\alpha}_t}}\left(\sqrt{C_1 k \log T}\|x_{i(x_t)}^\star - x_i^\star\|_2 + 4\sqrt{C_1 C_3}k\log T\sqrt{\frac{1-\overline{\alpha}_t}{\overline{\alpha}_t}}\right).$$

Here step (a) follows from (A.29) and (A.31); while step (b) utilizes (A.32), (A.3). Similarly we can use (A.30) to show that

$$|\Delta_2(x_0)| \leq \frac{9c_1 \log T}{T}\sqrt{\frac{\overline{\alpha}_t}{1-\overline{\alpha}_t}}\left(\sqrt{C_1 k \log T}\|x_{i(x_t)}^\star - x_i^\star\|_2 + 4\sqrt{C_1 C_3}k\log T\sqrt{\frac{1-\overline{\alpha}_t}{\overline{\alpha}_t}}\right).$$
(A.33b)

Notice that

$$\left|(x_0(x_t) - \widehat{x}_0)^\top(\widehat{x}_0 - x_0)\right| \overset{(i)}{\leq} \|x_0(x_t) - \widehat{x}_0\|_2 \|\widehat{x}_0 - x_0\|_2 \leq (\|x_0(x_t) - \overline{x}_0\|_2 + \|\delta\|_2)\|\widehat{x}_0 - x_0\|_2$$

$$\overset{(ii)}{\leq} \left[3\sqrt{\frac{C_3 k(1-\overline{\alpha}_t)\log T}{\overline{\alpha}_t}} + \sqrt{\frac{1-\overline{\alpha}_t}{\overline{\alpha}_t}}\exp\left(-\frac{C_3 k \log T}{32}\right)\right]\|\widehat{x}_0 - x_0\|_2$$

$$\overset{(iii)}{\leq} 4\sqrt{\frac{C_3 k(1-\overline{\alpha}_t)\log T}{\overline{\alpha}_t}}\|\widehat{x}_0 - x_0\|_2,$$

where step (i) utilizes the Cauchy-Schwarz inequality; step (ii) follows from (A.4), (A.28) and the fact that $x_0(x_t), \overline{x}_0 \in \mathcal{X}_t(x_t)$; step (iii) holds provided that $C_3$ is sufficiently large. Therefore we have

$$|\Delta_3(x_0)| \leq \frac{(1-\alpha_t)\overline{\alpha}_t}{(\alpha_t - \overline{\alpha}_t)(1-\overline{\alpha}_t)}\left|(x_0(x_t) - \widehat{x}_0)^\top(\widehat{x}_0 - x_0)\right|$$
(A.33c)

$$\leq \frac{(1-\alpha_t)\overline{\alpha}_t}{(\alpha_t - \overline{\alpha}_t)(1-\overline{\alpha}_t)} \cdot 4\sqrt{\frac{C_3 k(1-\overline{\alpha}_t)\log T}{\overline{\alpha}_t}}\|\widehat{x}_0 - x_0\|_2$$

$$\leq 32 c_1 \sqrt{C_3}\frac{\log T}{T}\sqrt{\frac{\overline{\alpha}_t}{1-\overline{\alpha}_t}}\sqrt{k\log T}\left(\|x_{i(x_t)}^\star - x_i^\star\|_2 + 4\sqrt{\frac{C_3 k(1-\overline{\alpha}_t)\log T}{\overline{\alpha}_t}}\right),$$

where the last relation follows from Lemma 8 and (A.32). Finally we have

$$|\Delta_4(x_0)| \leq \frac{(1-\alpha_t)\overline{\alpha}_t}{2(\alpha_t - \overline{\alpha}_t)(1-\overline{\alpha}_t)}\|\widehat{x}_0 - x_0\|_2^2$$

$$\leq \frac{8c_1 \log T}{T}\frac{\overline{\alpha}_t}{1-\overline{\alpha}_t}\left(\|x_{i(x_t)}^\star - x_i^\star\|_2^2 + 16\frac{C_3 k(1-\overline{\alpha}_t)\log T}{\overline{\alpha}_t}\right).$$
(A.33d)

Here step (a) follows from (A.32), step (b) follows from (A.4) and the fact that $x_{i(x_t)}^\star, x_i^\star \in \mathcal{X}_t(x_t)$ Taking the bounds in (A.33) collectively leads to

$$|\Delta(x_0)| \leq 5\sqrt{c_1 C_1}\sqrt{\frac{k}{T}}\log T\left(\sqrt{\frac{\overline{\alpha}_t}{1-\overline{\alpha}_t}}\|x_{i(x_t)}^\star - x_i^\star\|_2 + 4\sqrt{C_3 k \log T}\right)$$

$$+ \frac{8c_1 \log T}{T}\frac{\overline{\alpha}_t}{1-\overline{\alpha}_t}\|x_{i(x_t)}^\star - x_i^\star\|_2^2.$$
(A.34)

provided that $T$ is sufficiently large.

### A.5.2 Step 2: bounding $I_1$

For $x_0 \in \mathcal{X}_t(x_t)$, we know that $x_0 \in \mathcal{B}_i$ for some $i \in \mathcal{I}(x_t; C_3)$, hence $\overline{\alpha}_t\|x_i^\star - x_{i(x_t)}^\star\|_2^2 \leq C_3 k(1-\overline{\alpha}_t)\log T$. This combined with (A.34) gives

$$|\Delta(x_0)| \leq 25\sqrt{c_1 C_1 C_3}\sqrt{\frac{k^2 \log^3 T}{T}} + \frac{8c_1 C_3 k \log^2 T}{T} \leq 26\sqrt{c_1 C_1 C_3}\sqrt{\frac{k^2 \log^3 T}{T}}$$
(A.35)

provided that $T \gg k^2 \log^3 T$. Similarly we can check that for each $1 \leq i \leq 4$, $|\Delta_i(x_0)| \leq 1$. Then we know that for $x_0 \in \mathcal{X}_t(x_t)$, we have $\exp(\Delta(x_0)) \leq 1 + \Delta(x_0) + \Delta^2(x_0)$ as long as $T \gg k^2 \log^3 T$. Hence

$$
I_1 \leq 1 + \int_{\mathcal{X}_t(x_t)} p_{X_0|X_t}(x_0 \mid x_t) \Delta(x_0) \, dx_0 + \int_{\mathcal{X}_t(x_t)} p_{X_0|X_t}(x_0 \mid x_t) \Delta^2(x_0) \, dx_0
$$

$$
= 1 + \int p_{X_0|X_t}(x_0 \mid x_t) \Delta_1(x_0) \, dx_0 - \int_{\mathcal{Y}_t(x_t)} p_{X_0|X_t}(x_0 \mid x_t) \Delta_1(x_0) \, dx_0
$$

$$
+ \int_{\mathcal{X}_t(x_t)} p_{X_0|X_t}(x_0 \mid x_t) \left[ \Delta_2(x_0) + \Delta_3(x_0) + \Delta_4(x_0) + \Delta^2(x_0) \right] dx_0
$$

$$
\leq 1 + \underbrace{\int_{\mathcal{Y}_t(x_t)} p_{X_0|X_t}(x_0 \mid x_t) |\Delta_1(x_0)| \, dx_0}_{=:I_{1,1}}
$$

$$
+ \underbrace{\int_{\mathcal{X}_t(x_t)} p_{X_0|X_t}(x_0 \mid x_t) \left[ |\Delta_2(x_0) + \Delta_3(x_0) + \Delta_4(x_0)| + \Delta^2(x_0) \right] dx_0}_{=:I_{1,2}}.
$$

Here the last step follows from the fact that $\int p_{X_0|X_t}(x_0 \mid x_t) \Delta_1(x_0) dx_0 = 0$. The integral $I_{1,1}$ can be upper bounded similar to $I_2$, hence we defer its analysis to the next section. For the integral $I_{1,2}$, we have

$$
I_{1,2} \leq \max_{x_0 \in \mathcal{X}_t(x_t)} \left\{ |\Delta_2(x_0) + \Delta_3(x_0) + \Delta_4(x_0)| + \Delta^2(x_0) \right\}
$$

$$
\overset{(i)}{\leq} \max_{x_0 \in \mathcal{X}_t(x_t)} \left\{ 4\Delta_1^2(x_0) + 5|\Delta_2(x_0)| + 5|\Delta_3(x_0)| + 5|\Delta_4(x_0)| \right\}
$$

$$
\overset{(ii)}{\leq} 128 \frac{c_1 \log T}{T} \frac{\overline{\alpha}_t}{1 - \overline{\alpha}_t} \left( C_1 k \log T \frac{C_3 k (1 - \overline{\alpha}_t) \log T}{\overline{\alpha}_t} + 16 C_1 C_3 k^2 \log^2 T \frac{1 - \overline{\alpha}_t}{\overline{\alpha}_t} \right)
$$

$$
+ \frac{45 c_1 \log T}{T} \sqrt{\frac{\overline{\alpha}_t}{1 - \overline{\alpha}_t}} \left( \sqrt{C_1 k \log T} \sqrt{\frac{C_3 k (1 - \overline{\alpha}_t) \log T}{\overline{\alpha}_t}} + 4\sqrt{C_1 C_3} k \log T \sqrt{\frac{1 - \overline{\alpha}_t}{\overline{\alpha}_t}} \right)
$$

$$
+ 160 c_1 \sqrt{C_3} \frac{\log T}{T} \sqrt{\frac{\overline{\alpha}_t}{1 - \overline{\alpha}_t}} \sqrt{k \log T} \left( \sqrt{\frac{C_3 k (1 - \overline{\alpha}_t) \log T}{\overline{\alpha}_t}} + 4\sqrt{\frac{C_3 k (1 - \overline{\alpha}_t) \log T}{\overline{\alpha}_t}} \right)
$$

$$
+ \frac{40 c_1 \log T}{T} \frac{\overline{\alpha}_t}{1 - \overline{\alpha}_t} \left( \frac{C_3 k (1 - \overline{\alpha}_t) \log T}{\overline{\alpha}_t} + 16 \frac{C_3 k (1 - \overline{\alpha}_t) \log T}{\overline{\alpha}_t} \right)
$$

$$
\leq 2181 c_1 C_1 C_3 \frac{k^2 \log^3 T}{T}. \tag{A.36}
$$

Here step (i) follows from the Cauchy-Schwarz inequality and the facts that $|\Delta_i(x_0)| \leq 1$ for $i = 2, 3, 4$, while step (ii) follows from the bounds (A.33) and the fact that $\overline{\alpha}_t \|x_i^\star - x_{i(x_t)}^\star\|_2^2 \leq C_3 k(1 - \overline{\alpha}_t) \log T$.

### A.5.3 Step 3: bounding $I_2$.

For $x_0 \in \mathcal{Y}_t(x_t)$, we know that $x_0 \in \mathcal{B}_i$ for some $i \notin \mathcal{I}(x_t; C_3)$, hence $\overline{\alpha}_t \|x_i^\star - x_{i(x_t)}^\star\|_2^2 > C_3 k(1 - \overline{\alpha}_t) \log T$. This combined with (A.34) gives

$$
|\Delta(x_0)| \leq \left( 5\sqrt{\frac{c_1 C_1}{C_3}} \sqrt{\frac{\log T}{T}} + 20\sqrt{\frac{c_1 C_1}{C_3}} \sqrt{\frac{\log T}{T}} + \frac{8 c_1 \log T}{T} \right) \frac{\overline{\alpha}_t}{1 - \overline{\alpha}_t} \|x_{i(x_t)}^\star - x_i^\star\|_2^2
$$

$$
\leq 25\sqrt{\frac{c_1 C_1}{C_3}} \sqrt{\frac{\log T}{T}} \frac{\overline{\alpha}_t}{1 - \overline{\alpha}_t} \|x_{i(x_t)}^\star - x_i^\star\|_2^2 \tag{A.37}
$$

as long as $T$ is sufficiently large. Therefore we have

$$
I_2 = \int_{\mathcal{Y}_t(x_t)} p_{X_0|X_t}(x_0 \mid x_t) \exp(\Delta(x_0)) \, dx_0 \tag{A.38}
$$

$$\leq \sum_{i \notin \mathcal{I}(x_t; C_3)} \mathbb{P}\left(X_0 \in \mathcal{B}_i \mid X_t = x_t\right) \max_{x_0 \in \mathcal{B}_i} \exp\left(\Delta\left(x_0\right)\right)$$

$$\overset{\text{(i)}}{\leq} \sum_{i \notin \mathcal{I}(x_t; C_3)} \exp\left(-\left(\frac{1}{16} - 25\sqrt{\frac{c_1 C_1}{C_3}}\sqrt{\frac{\log T}{T}}\right)\frac{\overline{\alpha}_t}{1 - \overline{\alpha}_t}\|x^\star_{i(x_t)} - x^\star_i\|_2^2\right)\mathbb{P}\left(X_0 \in \mathcal{B}_i\right)$$

$$\overset{\text{(ii)}}{\leq} \sum_{i \notin \mathcal{I}(x_t; C_3)} \exp\left(-\frac{\overline{\alpha}_t}{32\left(1 - \overline{\alpha}_t\right)}\|x^\star_{i(x_t)} - x^\star_i\|_2^2\right)\mathbb{P}\left(X_0 \in \mathcal{B}_i\right) \overset{\text{(iii)}}{\leq} \exp\left(-\frac{C_3}{32}k \log T\right).$$

Here step (i) follows from (A.15) and (A.37); step (ii) holds as long as $T$ is sufficiently large; while step (iii) uses the fact that $\overline{\alpha}_t\|x^\star_i - x^\star_{i(x_t)}\|_2^2 > C_3 k(1 - \overline{\alpha}_t)\log T$ for $i \notin \mathcal{I}(x_t; C_3)$. By similar analysis, we can show that

$$I_{1,1} = \int_{\mathcal{Y}_t(x_t)} p_{X_0|X_t}\left(x_0 \mid x_t\right)|\Delta_1\left(x_0\right)|\,\mathrm{d}x_0 \leq \sum_{i \notin \mathcal{I}(x_t; C_3)} \mathbb{P}\left(X_0 \in \mathcal{B}_i \mid X_t = x_t\right)\max_{x_0 \in \mathcal{B}_i}|\Delta_1\left(x_0\right)|$$

$$\overset{\text{(a)}}{\leq} 20\sqrt{\frac{c_1 C_1 k \log^2 T}{T}}\sqrt{\frac{\overline{\alpha}_t}{1 - \overline{\alpha}_t}} \sum_{i \notin \mathcal{I}(x_t; C_3)} \mathbb{P}\left(X_0 \in \mathcal{B}_i \mid X_t = x_t\right)\|x^\star_{i(x_t)} - x^\star_i\|_2$$

$$\overset{\text{(b)}}{\leq} 20\sqrt{\frac{c_1 C_1 k \log^2 T}{T}}\sqrt{\frac{\overline{\alpha}_t}{1 - \overline{\alpha}_t}} \sum_{i \notin \mathcal{I}(x_t; C_3)} \exp\left(-\frac{\overline{\alpha}_t}{16\left(1 - \overline{\alpha}_t\right)}\|x^\star_{i(x_t)} - x^\star_i\|_2^2\right)\mathbb{P}\left(X_0 \in \mathcal{B}_i\right)\|x^\star_{i(x_t)} - x^\star_i\|_2$$

$$\overset{\text{(c)}}{\leq} 20\sqrt{\frac{c_1 C_1 k \log^2 T}{T}} \sum_{i \notin \mathcal{I}(x_t; C_3)} \exp\left(-\frac{\overline{\alpha}_t}{32\left(1 - \overline{\alpha}_t\right)}\|x^\star_{i(x_t)} - x^\star_i\|_2^2\right)\mathbb{P}\left(X_0 \in \mathcal{B}_i\right)$$

$$\overset{\text{(d)}}{\leq} 20\sqrt{\frac{c_1 C_1 k \log^2 T}{T}}\exp\left(-\frac{C_3}{32}k \log T\right). \tag{A.39}$$

Here step (a) follows from (A.33a) and the fact that $\overline{\alpha}_t\|x^\star_i - x^\star_{i(x_t)}\|_2^2 > C_3 k(1 - \overline{\alpha}_t)\log T$ for $i \notin \mathcal{I}(x_t; C_3)$; step (b) follows from (A.15); step (c) holds provided that $C_3$ is sufficiently large; step (d) follows again from the fact that $\overline{\alpha}_t\|x^\star_i - x^\star_{i(x_t)}\|_2^2 > C_3 k(1 - \overline{\alpha}_t)\log T$ for $i \notin \mathcal{I}(x_t; C_3)$.

### A.5.4   Step 4: putting everything together

Taking (A.36), (A.38) and (A.39) collectively, we have

$$\int_{x_0} p_{X_0|X_t}\left(x_0 \mid x_t\right)\exp\left(\Delta\left(x_t, x_{t-1}, x_0\right)\right)\mathrm{d}x_0 = I_1 + I_2 \leq 1 + I_{1,1} + I_{1,2} + I_2$$

$$\leq 1 + 2182 c_1 C_1 C_3 \frac{k^2 \log^3 T}{T},$$

provided that $T$ is sufficiently large. By similar argument, i.e., using the lower bounding $\exp(\Delta(x_0)) \geq 1 + \Delta(x_0) - \Delta^2(x_0)$ in Step 2 and repeat the same analysis, we can show that

$$\int_{x_0} p_{X_0|X_t}\left(x_0 \mid x_t\right)\exp\left(\Delta\left(x_0\right)\right)\mathrm{d}x_0 \geq 1 - 2182 c_1 C_1 C_3 \frac{k^2 \log^3 T}{T}.$$

This gives the desired result.

### A.6   Proof of Lemma 4

Recall that

$$\log \frac{p_{X_{t-1}|X_t}\left(x_{t-1} \mid x_t\right)}{p_{Y^\star_{t-1}|Y_t}\left(x_{t-1} \mid x_t\right)} = \log\left[\int_{x_0} p_{X_0|X_t}\left(x_0 \mid x_t\right)\exp\left(\Delta\left(x_t, x_{t-1}, x_0\right)\right)\mathrm{d}x_0\right].$$

For any $x_0 \in \mathcal{X}$, by the definition of $\Delta\left(x_t, x_{t-1}, x_0\right)$ in (4.4), we have

$$|\Delta\left(x_t, x_{t-1}, x_0\right)|$$

$$\leq \frac{\sqrt{\overline{\alpha}_t}}{\alpha_t - \overline{\alpha}_t} \|\sqrt{\alpha_t} x_{t-1} - x_t\|_2 \|\widehat{x}_0 - x_0\|_2 + \frac{(1-\alpha_t)\sqrt{\overline{\alpha}_t}}{(\alpha_t - \overline{\alpha}_t)(1-\overline{\alpha}_t)} \|x_t - \sqrt{\overline{\alpha}_t}\widehat{x}_0\|_2 \|\widehat{x}_0 - x_0\|_2$$

$$+ \frac{(1-\alpha_t)\overline{\alpha}_t}{2(\alpha_t - \overline{\alpha}_t)(1-\overline{\alpha}_t)} \|\widehat{x}_0 - x_0\|_2^2$$

$$\overset{(i)}{\leq} 2R \frac{\sqrt{\overline{\alpha}_t}}{\alpha_t - \overline{\alpha}_t} \|\sqrt{\alpha_t} x_{t-1} - x_t\|_2 + 2R \frac{(1-\alpha_t)\sqrt{\overline{\alpha}_t}}{(\alpha_t - \overline{\alpha}_t)(1-\overline{\alpha}_t)} \|x_t\|_2 + \frac{(1-\alpha_t)\overline{\alpha}_t}{(\alpha_t - \overline{\alpha}_t)(1-\overline{\alpha}_t)} 4R^2$$

$$\overset{(ii)}{\leq} 4RT^{c_0} \|\sqrt{\alpha_t} x_{t-1} - x_t\|_2 + 16c_1 RT^{c_0-1} \log T \|x_t\|_2 + 32c_1 R^2 T^{c_0-1} \log T.$$

Here step (i) follows from $\widehat{x}_0, x_0 \in \mathcal{X}$, hence $\max\{\|\widehat{x}_0\|_2, \|x_0\|_2\} \leq R$; while step (ii) follows from the facts that, for $2 \leq t \leq T$,

$$\frac{\sqrt{\overline{\alpha}_t}}{\alpha_t - \overline{\alpha}_t} \leq \frac{1}{\alpha_t - \prod_{i=1}^t \alpha_i} = \frac{1}{\alpha_t \left(1 - \prod_{i=1}^{t-1} \alpha_i\right)} \leq \frac{2}{1-\alpha_1} \leq 2T^{c_0},$$

and in view of Lemma 8,

$$\frac{(1-\alpha_t)\overline{\alpha}_t}{(\alpha_t - \overline{\alpha}_t)(1-\overline{\alpha}_t)} \leq \frac{(1-\alpha_t)\sqrt{\overline{\alpha}_t}}{(\alpha_t - \overline{\alpha}_t)(1-\overline{\alpha}_t)} \leq \frac{8c_1 \log T}{T} \frac{1}{1-\overline{\alpha}_t} \leq 8c_1 T^{c_0-1} \log T.$$

Hence we have

$$\left| \log \frac{p_{X_{t-1}|X_t}(x_{t-1} \mid x_t)}{p_{Y_{t-1}^\star|Y_t}(x_{t-1} \mid x_t)} \right|$$

$$= \left| \log \left[ \int_{x_0} p_{X_0|X_t}(x_0 \mid x_t) \exp\left(\Delta(x_t, x_{t-1}, x_0)\right) dx_0 \right] \right| \leq \sup_{x_0 \in \mathcal{X}} |\Delta(x_t, x_{t-1}, x_0)|$$

$$\leq 4RT^{c_0} \|\sqrt{\alpha_t} x_{t-1} - x_t\|_2 + 16c_1 RT^{c_0-1} \log T \|x_t\|_2 + 32c_1 R^2 T^{c_0-1} \log T$$

$$\leq T^{c_0 + 2c_R} \left(\|\sqrt{\alpha_t} x_{t-1} - x_t\|_2 + \|x_t\|_2 + 1\right)$$

as long as $T$ is sufficiently large.

### A.7 Proof of Lemma 5

Regarding $\Delta_{t,1}$, we first utilize Lemma 3 to show that for any $(x_t, x_{t-1}) \in \mathcal{A}_t$,

$$\left| 1 - \frac{p_{Y_{t-1}^\star|Y_t}(x_{t-1} \mid x_t)}{p_{X_{t-1}|X_t}(x_{t-1} \mid x_t)} \right| \leq C_5 \frac{k^2 \log^3 T}{T}.$$

Since $\log(1-x) \geq -x - x^2$ holds for any $x \in [-1/2, 1/2]$, we know that when $T \gg k^2 \log^3 T$, we have

$$p_{X_{t-1}|X_t}(x_{t-1} \mid x_t) \log \frac{p_{X_{t-1}|X_t}(x_{t-1} \mid x_t)}{p_{Y_{t-1}^\star|Y_t}(x_{t-1} \mid x_t)}$$

$$= -p_{X_{t-1}|X_t}(x_{t-1} \mid x_t) \log \left[ 1 - \left( 1 - \frac{p_{Y_{t-1}^\star|Y_t}(x_{t-1} \mid x_t)}{p_{X_{t-1}|X_t}(x_{t-1} \mid x_t)} \right) \right]$$

$$\leq p_{X_{t-1}|X_t}(x_{t-1} \mid x_t) \left[ 1 - \frac{p_{Y_{t-1}^\star|Y_t}(x_{t-1} \mid x_t)}{p_{X_{t-1}|X_t}(x_{t-1} \mid x_t)} + \left( 1 - \frac{p_{Y_{t-1}^\star|Y_t}(x_{t-1} \mid x_t)}{p_{X_{t-1}|X_t}(x_{t-1} \mid x_t)} \right)^2 \right]$$

$$= p_{X_{t-1}|X_t}(x_{t-1} \mid x_t) - p_{Y_{t-1}^\star|Y_t}(x_{t-1} \mid x_t) + p_{X_{t-1}|X_t}(x_{t-1} \mid x_t) C_5^2 \frac{k^4 \log^6 T}{T^2}$$

Hence we have

$$\Delta_{t,1} \leq \int_{(x_t, x_{t-1}) \in \mathcal{A}_t} \left[ -p_{Y_{t-1}^\star|Y_t}(x_{t-1} \mid x_t) + p_{X_{t-1}|X_t}(x_{t-1} \mid x_t) \right] p_{X_t}(x_t) \, dx_{t-1} dx_t + C_5^2 \frac{k^4 \log^6 T}{T^2}$$

$$= \int_{(x_t, x_{t-1}) \in \mathcal{A}_t^c} \left[ p_{Y_{t-1}^\star|Y_t}(x_{t-1} \mid x_t) - p_{X_{t-1}|X_t}(x_{t-1} \mid x_t) \right] p_{X_t}(x_t) \, dx_{t-1} dx_t + C_5^2 \frac{k^4 \log^6 T}{T^2}$$

$$\leq \underbrace{\int_{(x_t, x_{t-1}) \in \mathcal{A}_t^c} p_{Y_{t-1}^\star | Y_t}(x_{t-1} \mid x_t) \, p_{X_t}(x_t) \, \mathrm{d}x_{t-1} \mathrm{d}x_t}_{=: \Delta_{t,3}} + C_5^2 \frac{k^4 \log^6 T}{T^2}.$$

Here the penultimate step follows from the fact that

$$\int p_{Y_{t-1}^\star | Y_t}(x_{t-1} \mid x_t) \, p_{X_t}(x_t) \, \mathrm{d}x_{t-1} \mathrm{d}x_t = \int p_{X_{t-1} | X_t}(x_{t-1} \mid x_t) \, p_{X_t}(x_t) \, \mathrm{d}x_{t-1} \mathrm{d}x_t = 1.$$

It boils down to bounding $\Delta_{t,3}$. In view of (A.26), we know that

$$\Delta_{t,3} = \int p_{Y_{t-1}^\star | Y_t}(x_{t-1} \mid x_t) \, p_{X_t}(x_t) \mathbb{1}\{x_t \notin \mathcal{T}_t\} \, \mathrm{d}x_{t-1} \mathrm{d}x_t$$

$$+ \int p_{Y_{t-1}^\star | Y_t}(x_{t-1} \mid x_t) \, p_{X_t}(x_t) \mathbb{1}\left\{x_t \in \mathcal{T}_t, \frac{x_t - \sqrt{\alpha_t}x_{t-1}}{\sqrt{1 - \alpha_t}} \notin \mathcal{G}\right\} \mathrm{d}x_{t-1} \mathrm{d}x_t$$

$$= \mathbb{P}(X_t \notin \mathcal{T}_t) + \mathbb{P}\left(X_t \in \mathcal{T}_t, \frac{X_t - (X_t + \eta_t^\star s_t^\star(X_t) + \sigma_t^\star Z)}{\sqrt{1 - \alpha_t}} \notin \mathcal{G}\right) \quad \text{where} \quad Z \sim \mathcal{N}(0, I_d)$$

$$= \mathbb{P}(X_t \notin \mathcal{T}_t) + \mathbb{P}\left(X_t \in \mathcal{T}_t, -\frac{\eta_t^\star s_t^\star(X_t) + \sigma_t^\star Z}{\sqrt{1 - \alpha_t}} \notin \mathcal{G}\right)$$

Here we use the fact that $Y_{t-1}^\star \mid Y_t = x_t \sim \mathcal{N}\big((x_t + \eta_t^\star s_t^\star(x_t))/\sqrt{\alpha_t}, (\sigma_t^{\star 2}/\alpha_t)I_d\big)$. Notice that

$$-\frac{\eta_t^\star s_t^\star(X_t) + \sigma_t^\star Z}{\sqrt{1 - \alpha_t}} = -\sqrt{1 - \alpha_t} s_t^\star(X_t) - \sqrt{\frac{\alpha_t - \overline{\alpha}_t}{1 - \overline{\alpha}_t}} Z.$$

The following claim is cricial for understanding this random variable.

***Claim*** 1. For any $x_t \in \mathcal{T}_t$, we have

$$\left\|\sqrt{1 - \alpha_t} s_t^\star(x_t)\right\|_2 \leq \frac{1}{2}\big(\sqrt{d} + \sqrt{C_1 k \log T}\big),$$

and for any $1 \leq i \leq j \leq N_\varepsilon$,

$$\sqrt{1 - \alpha_t}|(x_i^\star - x_j^\star)^\top s_t^\star(x_t)| \leq \frac{1}{2}\sqrt{C_1 k \log T}\|x_i^\star - x_j^\star\|_2.$$

*Proof.* See Appendix A.7.1. $\qquad\qquad\square$

Since $Z \sim \mathcal{N}(0, I_d)$, in view of Lemma 9, with probability exceeding $1 - \exp\left(-(C_1/64)k \log T\right)$,

$$\sqrt{\frac{\alpha_t - \overline{\alpha}_t}{1 - \overline{\alpha}_t}} \|Z\|_2 \leq \|Z\|_2 \leq \sqrt{d} + \frac{1}{2}\sqrt{C_1 k \log T}$$

and for any $1 \leq i \leq j \leq N_\varepsilon$,

$$\sqrt{\frac{\alpha_t - \overline{\alpha}_t}{1 - \overline{\alpha}_t}}|(x_i^\star - x_j^\star)^\top Z| \leq |(x_i^\star - x_j^\star)^\top Z| \leq \frac{1}{2}\sqrt{C_1 k \log T}\|x_i^\star - x_j^\star\|_2.$$

These combined with Claim 1 allow us to show that

$$\mathbb{P}\left(X_t \in \mathcal{T}_t, -\frac{\eta_t^\star s_t^\star(X_t) + \sigma_t^\star Z}{\sqrt{1 - \alpha_t}} \notin \mathcal{G}\right) \leq \exp\left(-\frac{C_1}{64}k \log T\right).$$

Taking the above inequality collectively with Lemma 1 gives

$$\Delta_{t,3} \leq \exp\left(-\frac{C_1}{4}k \log T\right) + \exp\left(-\frac{C_1}{64}k \log T\right) \leq 2\exp\left(-\frac{C_1}{64}k \log T\right).$$

Hence we have

$$\Delta_{t,1} \leq \Delta_{t,3} + C_5^2 \frac{k^4 \log^6 T}{T^2} \leq 2\exp\left(-\frac{C_1}{64}k \log T\right) + C_5^2 \frac{k^4 \log^6 T}{T^2} \leq 2C_5^2 \frac{k^4 \log^6 T}{T^2}$$

as long as $T$ is sufficiently large.

### A.7.1 Proof of Claim 1

Consider the decomposition $x_t = \sqrt{\overline{\alpha}_t}x_0(x_t) + \sqrt{1-\overline{\alpha}_t}\omega$ as in Appendix A.1, where $x_0(x_t) \in \mathcal{B}_{i(x_t)}$ for some $i(x_t) \in \mathcal{I}$ and $\omega \in \mathcal{G}$. Notice that

$$s_t^\star(x_t) = -\frac{1}{1-\overline{\alpha}_t}\left(x_t - \sqrt{\overline{\alpha}_t}\widehat{x}_0\right) = -\frac{1}{1-\overline{\alpha}_t}\left[\sqrt{\overline{\alpha}_t}x_0(x_t) + \sqrt{1-\overline{\alpha}_t}\omega - \sqrt{\overline{\alpha}_t}(\overline{x}_0 + \delta)\right]$$

$$= -\frac{\sqrt{\overline{\alpha}_t}}{1-\overline{\alpha}_t}(x_0(x_t) - \overline{x}_0) - \frac{1}{\sqrt{1-\overline{\alpha}_t}}\omega + \frac{\sqrt{\overline{\alpha}_t}}{1-\overline{\alpha}_t}\delta.$$

where $\widehat{x}_0 := \mathbb{E}[X_0 \,|\, X_t = x_t]$ is defined in (4.3), whereas $\overline{x}_0 \in \mathcal{X}_t(x_t)$ and $\delta$ are defined in (A.27). Therefore we can check that

$$\left\|\sqrt{1-\overline{\alpha}_t}s_t^\star(x_t)\right\|_2 \leq \frac{\sqrt{\overline{\alpha}_t(1-\overline{\alpha}_t)}}{1-\overline{\alpha}_t}\|x_0(x_t) - \overline{x}_0\|_2 + \sqrt{\frac{1-\overline{\alpha}_t}{1-\overline{\alpha}_t}}\|\omega\|_2 + \frac{\sqrt{\overline{\alpha}_t(1-\overline{\alpha}_t)}}{1-\overline{\alpha}_t}\|\delta\|_2$$

$$\overset{(i)}{\leq} \frac{\sqrt{\overline{\alpha}_t(1-\overline{\alpha}_t)}}{1-\overline{\alpha}_t}3\sqrt{\frac{C_3k(1-\overline{\alpha}_t)\log T}{\overline{\alpha}_t}} + \sqrt{\frac{1-\overline{\alpha}_t}{1-\overline{\alpha}_t}}\left(\sqrt{d} + \sqrt{C_1k\log T}\right) + \sqrt{\frac{1-\overline{\alpha}_t}{1-\overline{\alpha}_t}}\exp\left(-\frac{C_3k\log T}{32}\right)$$

$$\overset{(ii)}{\leq} \sqrt{\frac{8c_1\log T}{T}}\left[3\sqrt{C_3k\log T} + \sqrt{d} + \sqrt{C_1k\log T} + \exp\left(-\frac{C_3k\log T}{32}\right)\right]$$

$$\overset{(iii)}{\leq} \frac{1}{2}\left(\sqrt{d} + \sqrt{C_1k\log T}\right).$$

Here step (i) follows from (A.4), the fact that $\omega \in \mathcal{G}$, and (A.28); step (ii) follows from Lemma 8; while step (iii) holds provided that $T$ is sufficiently large. In addition, for any $1 \leq i \leq j \leq N_\varepsilon$ we have

$$\sqrt{1-\overline{\alpha}_t}|(x_i^\star - x_j^\star)^\top s_t^\star(x_t)|$$

$$\overset{(a)}{\leq} \frac{\sqrt{\overline{\alpha}_t(1-\overline{\alpha}_t)}}{1-\overline{\alpha}_t}\|x_0(x_t) - \overline{x}_0\|_2\|x_i^\star - x_j^\star\|_2 + \sqrt{\frac{1-\overline{\alpha}_t}{1-\overline{\alpha}_t}}|\omega^\top(x_i^\star - x_j^\star)^\top|$$

$$+ \frac{\sqrt{\overline{\alpha}_t(1-\overline{\alpha}_t)}}{1-\overline{\alpha}_t}\|\delta\|_2\|x_i^\star - x_j^\star\|_2$$

$$\overset{(b)}{\leq} \frac{\sqrt{\overline{\alpha}_t(1-\overline{\alpha}_t)}}{1-\overline{\alpha}_t}3\sqrt{\frac{C_3k(1-\overline{\alpha}_t)\log T}{\overline{\alpha}_t}}\|x_i^\star - x_j^\star\|_2 + \sqrt{\frac{1-\overline{\alpha}_t}{1-\overline{\alpha}_t}}\sqrt{C_1k\log T}\|x_i^\star - x_j^\star\|_2$$

$$+ \frac{\sqrt{\overline{\alpha}_t(1-\overline{\alpha}_t)}}{1-\overline{\alpha}_t}\sqrt{\frac{1-\overline{\alpha}_t}{\overline{\alpha}_t}}\exp\left(-\frac{C_3k\log T}{32}\right)\|x_i^\star - x_j^\star\|_2$$

$$\overset{(c)}{\leq} \sqrt{\frac{8c_1\log T}{T}}\left[3\sqrt{C_3k\log T} + \sqrt{C_1k\log T} + \exp\left(-\frac{C_3k\log T}{32}\right)\right]\|x_i^\star - x_j^\star\|_2$$

$$\overset{(d)}{\leq} \frac{1}{2}\sqrt{C_1k\log T}\|x_i^\star - x_j^\star\|_2.$$

Here step (a) utilizes the Cauchy-Schwarz inequality; step (b) follows from (A.4), the fact that $\omega \in \mathcal{G}$, and (A.28); step (c) follows from Lemma 8; while step (d) holds when $T$ is sufficiently large.

### A.8 Proof of Lemma 6

We can upper bound $|\Delta_{t,2}|$ by

$$|\Delta_{t,2}| \overset{(i)}{\leq} T^{c_0+2c_R}\int(\|\sqrt{\overline{\alpha}_t}x_{t-1} - x_t\|_2 + \|x_t\|_2 + 1)\,p_{X_{t-1},X_t}(x_{t-1},x_t)\,\mathbb{1}\{(x_t,x_{t-1}) \notin \mathcal{A}_t\}\,\mathrm{d}x_{t-1}\mathrm{d}x_t$$

$$= T^{c_0+2c_R}\mathbb{E}\left[(\|\sqrt{\overline{\alpha}_t}X_{t-1} - X_t\|_2 + \|X_t\|_2 + 1)\mathbb{1}\{(X_t,X_{t-1}) \notin \mathcal{A}_t\}\right]$$

$$\overset{(ii)}{=} T^{c_0+2c_R}\mathbb{E}\left[\left(\sqrt{1-\overline{\alpha}_t}\|W_t\|_2 + \|X_t\|_2 + 1\right)\mathbb{1}\{(X_t,X_{t-1}) \notin \mathcal{A}_t\}\right]$$

$$\overset{(iii)}{\leq} T^{c_0+2c_R}\sqrt{1-\overline{\alpha}_t}\mathbb{E}^{1/2}\left[\|W_t\|_2^2\right]\mathbb{P}^{1/2}\left((X_t,X_{t-1}) \notin \mathcal{A}_t\right) + T^{c_0+2c_R}\mathbb{E}^{1/2}\left[\|X_t\|_2^2\right]\mathbb{P}^{1/2}\left((X_t,X_{t-1}) \notin \mathcal{A}_t\right)$$

$$+ T^{c_0+2c_R}\mathbb{P}\left((X_t,X_{t-1}) \notin \mathcal{A}_t\right). \tag{A.40}$$

Here step (i) follows from Lemma 4; step (ii) follows from the update rule (2.1); step (iii) utilizes the Cauchy-Schwarz inequality. In view of (2.2), we have

$$
\begin{aligned}
\mathbb{E}\left[\|X_t\|_2^2\right] = \mathbb{E}\left[\|\sqrt{\overline{\alpha}_t}X_0 + \sqrt{1-\overline{\alpha}_t}\,\overline{W}_t\|_2^2\right] &\leq \mathbb{E}\left[2\overline{\alpha}_t R^2 + 2\left(1-\overline{\alpha}_t\right)\|\overline{W}_t\|_2^2\right] \\
&= 2\overline{\alpha}_t R^2 + 2\left(1-\overline{\alpha}_t\right)d \leq 2R^2 + 2d,
\end{aligned}
\tag{A.41}
$$

where we use the fact that $\mathbb{E}[\|\overline{W}_t\|_2^2] = d$. Then we have

$$
\begin{aligned}
|\Delta_{t,2}| &\overset{(a)}{\leq} T^{c_0+2c_R}\left(\sqrt{d\left(1-\alpha_t\right)} + \sqrt{2R^2+2d}\right)\mathbb{P}^{1/2}\left((X_t, X_{t-1}) \notin \mathcal{A}_t\right) + T^{c_0+2c_R}\mathbb{P}\left((X_t, X_{t-1}) \notin \mathcal{A}_t\right) \\
&\overset{(b)}{\leq} T^{c_0+2c_R}\left(2R+3\sqrt{d}\right)\exp\left(-\frac{C_1}{8}k\log T\right) + T^{c_0+2c_R}\exp\left(-\frac{C_1}{4}k\log T\right) \\
&\overset{(c)}{\leq} \exp\left(-\frac{C_1}{16}k\log T\right)
\end{aligned}
$$

Here step (a) utilizes (A.41) and the fact that $\mathbb{E}[\|W_t\|_2^2] = d$; step (b) follows from Lemma 1; while step (c) makes use of the assumption that $k \geq \log d$ and holds provided that $C_1 \gg c_0 + c_R$.

## A.9 Proof of Lemma 7

We first decompose $K_t$ into

$$
\begin{aligned}
K_t &= \int p_{X_{t-1}|X_t}\left(x_{t-1}\,|\,x_t\right)p_{X_t}\left(x_t\right)\left(x_{t-1} - \mu_t^\star\left(x_t\right)\right)^\top \varepsilon_t\left(x_t\right)\mathrm{d}x_{t-1}\mathrm{d}x_t \\
&\overset{(i)}{=} \int \left(p_{X_{t-1}|X_t}\left(x_{t-1}\,|\,x_t\right) - p_{Y_{t-1}^\star|Y_t}\left(x_{t-1}\,|\,x_t\right)\right)p_{X_t}\left(x_t\right)\left(x_{t-1} - \mu_t^\star\left(x_t\right)\right)^\top \varepsilon_t\left(x_t\right)\mathrm{d}x_{t-1}\mathrm{d}x_t \\
&\overset{(ii)}{=} \left(\int_{\mathcal{A}_t} + \int_{\mathcal{A}_t^c}\right)\left(p_{X_{t-1}|X_t}\left(x_{t-1}\,|\,x_t\right) - p_{Y_{t-1}^\star|Y_t}\left(x_{t-1}\,|\,x_t\right)\right)p_{X_t}\left(x_t\right)\left(x_{t-1} - \mu_t^\star\left(x_t\right)\right)^\top \varepsilon_t\left(x_t\right)\mathrm{d}x_{t-1}\mathrm{d}x_t \\
&=: K_{t,1} + K_{t,2}.
\end{aligned}
$$

Here step (i) follows from the fact that $\int p_{Y_{t-1}^\star|Y_t}\left(x_{t-1}\,|\,x_t\right)\left(x_{t-1} - \mu_t^\star(x_t)\right)\mathrm{d}x_{t-1} = 0$ for any $x_t \in \mathbb{R}^d$, and $K_1$ and $K_2$ are defined to be the two integrals over $\mathcal{A}_t$ and $\mathcal{A}_t^c$ in step (ii). The following two claims provide bounds for the two integrals $K_{t,1}$ and $K_{t,2}$ respectively.

***Claim* 2.** Suppose that $T \gg k^2\log^3 T$. Then for each $2 \leq t \leq T$, we have

$$
|K_{t,1}| \leq 3C_5\frac{k^2\log^3 T}{T}\sqrt{\frac{c_1\log T}{T}}\mathbb{E}_{x_t\sim q_t}\left[\|\varepsilon_t\left(x_t\right)\|_2\right].
$$

*Proof.* See Appendix A.9.1. □

***Claim* 3.** Suppose that $T \gg 1$. Then for each $2 \leq t \leq T$, we have

$$
|K_{t,2}| \leq 2\exp\left(-\frac{C_1}{32}k\log T\right)\mathbb{E}_{x_t\sim q_t}^{1/2}\left[\|\varepsilon_t\left(x_t\right)\|_2^2\right].
$$

*Proof.* See Appendix A.9.2. □

Then we conclude that

$$
\begin{aligned}
|K_t| &\leq |K_{t,1}| + |K_{t,2}| \\
&\overset{(a)}{\leq} 3C_5\frac{k^2\log^3 T}{T}\sqrt{\frac{c_1\log T}{T}}\mathbb{E}_{x_t\sim q_t}\left[\|\varepsilon_t\left(x_t\right)\|_2\right] + 2\exp\left(-\frac{C_1}{32}k\log T\right)\mathbb{E}_{x_t\sim q_t}^{1/2}\left[\|\varepsilon_t\left(x_t\right)\|_2^2\right] \\
&\overset{(b)}{\leq} 4C_5\frac{k^2\log^3 T}{T}\sqrt{\frac{c_1\log T}{T}}\mathbb{E}_{x_t\sim q_t}^{1/2}\left[\|\varepsilon_t\left(x_t\right)\|_2^2\right]
\end{aligned}
$$

as claimed. Here step (a) follows from Claim 2 and Claim 3; while step (b) utilizes Jensen's inequality, and holds provided that $T$ is sufficiently large.

### A.9.1 Proof of Claim 2

The term $K_{t,1}$ can be upper bounded by

$$
\begin{aligned}
|K_{t,1}| &= \left| \int_{\mathcal{A}_t} \left( \frac{p_{X_{t-1}|X_t}(x_{t-1}\,|\,x_t)}{p_{Y_{t-1}^\star|Y_t}(x_{t-1}\,|\,x_t)} - 1 \right) p_{Y_{t-1}^\star|Y_t}(x_{t-1}\,|\,x_t)\, p_{X_t}(x_t)\,(x_{t-1} - \mu_t^\star(x_t))^\top \varepsilon_t(x_t)\,\mathrm{d}x_{t-1}\mathrm{d}x_t \right| \\
&\stackrel{\text{(i)}}{\leq} \int_{\mathcal{A}_t} \left| 1 - \frac{p_{X_{t-1}|X_t}(x_{t-1}\,|\,x_t)}{p_{Y_{t-1}^\star|Y_t}(x_{t-1}\,|\,x_t)} \right| p_{Y_{t-1}^\star|Y_t}(x_{t-1}\,|\,x_t)\, p_{X_t}(x_t) \left| (x_{t-1} - \mu_t^\star(x_t))^\top \varepsilon_t(x_t) \right| \mathrm{d}x_{t-1}\mathrm{d}x_t \\
&\stackrel{\text{(ii)}}{\leq} C_5 \frac{k^2 \log^3 T}{T} \int_{\mathcal{A}_t} p_{Y_{t-1}^\star|Y_t}(x_{t-1}\,|\,x_t)\, p_{X_t}(x_t) \left| (x_{t-1} - \mu_t^\star(x_t))^\top \varepsilon_t(x_t) \right| \mathrm{d}x_{t-1}\mathrm{d}x_t \\
&\stackrel{\text{(iii)}}{=} C_5 \frac{k^2 \log^3 T}{T} \mathbb{E}\left[ \frac{\sigma_t^\star}{\sqrt{\alpha_t}} \left| Z^\top \varepsilon_t(X_t) \right| \mathbb{1}\left\{ \left( X_t, \frac{X_t + \eta_t s_t^\star(X_t) + \sigma_t^\star Z}{\sqrt{\alpha_t}} \right) \in \mathcal{A}_t \right\} \right] \\
&\leq C_5 \frac{k^2 \log^3 T}{T} \sqrt{\frac{(1-\alpha_t)(\alpha_t - \overline{\alpha}_t)}{\alpha_t(1-\overline{\alpha}_t)}} \mathbb{E}\left[ \left| Z^\top \varepsilon_t(X_t) \right| \right] \stackrel{\text{(iv)}}{\leq} C_5 \frac{k^2 \log^3 T}{T} \sqrt{\frac{8c_1 \log T}{T}} \frac{2}{\sqrt{2\pi}} \mathbb{E}\left[ \|\varepsilon_t(X_t)\|_2 \right] \\
&\leq 3C_5 \frac{k^2 \log^3 T}{T} \sqrt{\frac{c_1 \log T}{T}} \mathbb{E}_{x_t \sim q_t}\left[ \|\varepsilon_t(x_t)\|_2 \right].
\end{aligned}
$$

Here step (i) follows from Jensen's inequality; step (ii) utilizes Lemma 3; step (iii) follows from the definition of $Y_t^\star$ in (4.1) and of $\mu_t^\star$ in (4.7), and $Z_t \sim \mathcal{N}(0, I_d)$ is independent of $X_t$; step (iv) follows from Lemma 8 and the fact that $Z_t^\top \varepsilon_t(X_t)\,|\,X_t \sim \mathcal{N}(0, \|\varepsilon_t(X_t)\|_2^2)$ and hence

$$
\mathbb{E}\left[ \left| Z_t^\top \varepsilon_t(X_t) \right| \right] = \mathbb{E}\left[ \mathbb{E}\left[ \left| Z_t^\top \varepsilon_t(X_t) \right| \,|\, X_t \right] \right] = \frac{2}{\sqrt{2\pi}} \mathbb{E}\left[ \|\varepsilon_t(X_t)\|_2 \right].
$$

### A.9.2 Proof of Claim 3

The term $K_{t,1}$ can be upper bounded by

$$
\begin{aligned}
|K_{t,2}| &\leq \int_{\mathcal{A}_t^c} \left( p_{X_{t-1}|X_t}(x_{t-1}\,|\,x_t) + p_{Y_{t-1}^\star|Y_t}(x_{t-1}\,|\,x_t) \right) p_{X_t}(x_t) \|x_{t-1} - \mu_t^\star(x_t)\|_2 \|\varepsilon_t(x_t)\|_2 \,\mathrm{d}x_{t-1}\mathrm{d}x_t \\
&\leq \underbrace{\left[ \int_{\mathcal{A}_t^c} \left( p_{X_{t-1}|X_t}(x_{t-1}\,|\,x_t) + p_{Y_{t-1}^\star|Y_t}(x_{t-1}\,|\,x_t) \right) p_{X_t}(x_t) \|x_{t-1} - \mu_t^\star(x_t)\|_2^2 \,\mathrm{d}x_{t-1}\mathrm{d}x_t \right]^{1/2}}_{=:\gamma_1} \\
&\quad \cdot \underbrace{\left[ \int_{\mathcal{A}_t^c} \left( p_{X_{t-1}|X_t}(x_{t-1}\,|\,x_t) + p_{Y_{t-1}^\star|Y_t}(x_{t-1}\,|\,x_t) \right) p_{X_t}(x_t) \|\varepsilon_t(x_t)\|_2^2 \,\mathrm{d}x_{t-1}\mathrm{d}x_t \right]^{1/2}}_{=:\gamma_2}.
\end{aligned}
$$

The second term $\gamma_2$ can be easily bounded by

$$
\gamma_2 \leq \sqrt{2} \mathbb{E}_{x_t \sim q_t}^{1/2}\left[ \|\varepsilon_t(x_t)\|_2^2 \right].
$$

In what follows, we will bound the first term $\gamma_1$. Note that

$$
\gamma_1^2 = \underbrace{\int_{\mathcal{A}_t^c} p_{X_{t-1},X_t}(x_{t-1}, x_t) \|x_{t-1} - \mu_t^\star(x_t)\|_2^2 \,\mathrm{d}x_{t-1}\mathrm{d}x_t}_{=:\gamma_{1,1}}
$$
$$
+ \underbrace{\int_{\mathcal{A}_t^c} p_{Y_{t-1}^\star|Y_t}(x_{t-1}\,|\,x_t)\, p_{X_t}(x_t) \|x_{t-1} - \mu_t^\star(x_t)\|_2^2 \,\mathrm{d}x_{t-1}\mathrm{d}x_t}_{=:\gamma_{1,2}}.
$$

We have

$$
\gamma_{1,1} = \mathbb{E}\left[ \|X_{t-1} - \mu_t^\star(X_t)\|_2^2 \mathbb{1}\left\{ (X_t, X_{t-1}) \notin \mathcal{A}_t \right\} \right] \stackrel{\text{(i)}}{\leq} \mathbb{E}^{1/2}\left[ \|X_{t-1} - \mu_t^\star(X_t)\|_2^4 \right] \mathbb{P}^{1/2}\left( (X_t, X_{t-1}) \notin \mathcal{A}_t \right)
$$

$$\overset{\text{(ii)}}{\leq} \alpha_t^{-2} \mathbb{E}^{1/2} \left[ \left\| \sqrt{1-\alpha_t} W_t + \eta_t^\star s_t^\star (X_t) \right\|_2^4 \right] \exp\left( -\frac{C_1}{8} k \log T \right)$$

$$\overset{\text{(iii)}}{\leq} 4 \mathbb{E}^{1/2} \left[ \left\| \sqrt{1-\alpha_t} W_t + \eta_t^\star s_t^\star (X_t) \right\|_2^4 \right] \exp\left( -\frac{C_1}{8} k \log T \right)$$

Here step (i) follows from Cauchy-Schwarz inequality; step (ii) follows from Lemma 1 and the definition of $\mu_t^\star$ in (4.7); while step (iii) uses the fact that $\alpha_t \geq 1/2$ (see Lemma 8). Recall the definition of $s_t^\star(\cdot)$

$$s_t^\star (x_t) = -\frac{1}{1-\overline{\alpha}_t} \left( x_t - \sqrt{\overline{\alpha}_t} \mathbb{E}\left[ X_0 \mid X_t = x_t \right] \right),$$

which leads to the following upper bound

$$\| s_t^\star (X_t) \| \leq \frac{1}{1-\overline{\alpha}_t} \| X_t \|_2 + \frac{\sqrt{\overline{\alpha}_t}}{1-\overline{\alpha}_t} R = \frac{1}{1-\overline{\alpha}_t} \left\| \sqrt{\overline{\alpha}_t} X_0 + \sqrt{1-\overline{\alpha}_t} \, \overline{W}_t \right\|_2 + \frac{\sqrt{\overline{\alpha}_t}}{1-\overline{\alpha}_t} R$$

$$\leq \frac{1}{\sqrt{1-\overline{\alpha}_t}} \left\| \overline{W}_t \right\|_2 + 2 \frac{\sqrt{\overline{\alpha}_t}}{1-\overline{\alpha}_t} R. \tag{A.42}$$

Hence we have

$$\mathbb{E}\left[ \left\| \sqrt{1-\alpha_t} W_t + \eta_t^\star s_t^\star (X_t) \right\|_2^4 \right] \overset{\text{(i)}}{\leq} 8 (1-\alpha_t)^2 \mathbb{E}\left[ \| W_t \|_2^4 \right] + (1-\alpha_t)^4 \mathbb{E}\left[ \| s_t^\star (X_t) \|_2^4 \right]$$

$$\overset{\text{(ii)}}{\leq} 8 \left( \frac{c_1 \log T}{T} \right)^2 \mathbb{E}\left[ \| W_t \|_2^4 \right] + \left( \frac{8 c_1 \log T}{T} \right)^4 \mathbb{E}\left[ \left( \left\| \overline{W}_t \right\|_2 + R \right)^4 \right]$$

$$\overset{\text{(iii)}}{\leq} \frac{1}{16} \left( d^2 + R^4 \right).$$

Here step (i) follows from the elementary inequality $8(x^4 + y^4) \geq (x+y)^2$; step (ii) follows from Lemma 8 and (A.42); step (iii) follows from $W_t, \overline{W}_t \sim \mathcal{N}(0, I_d)$ and the proviso that $T$ being sufficiently large. Hence we have

$$\gamma_{1,1} \leq \sqrt{d^2 + R^4} \exp\left( -\frac{C_1}{8} k \log T \right) \leq \exp\left( -\frac{C_0}{16} k \log T \right)$$

as long as $C_0 \gg c_R$ and $k \geq \log d$. Regarding $\gamma_{1,2}$, we have

$$\gamma_{1,2} \overset{\text{(i)}}{=} \mathbb{E}\left[ \left\| \frac{X_t + \eta_t s_t^\star (X_t) + \sigma_t^\star Z_t}{\sqrt{\alpha_t}} - \frac{X_t + \eta_t^\star s_t^\star (X_t)}{\sqrt{\alpha_t}} \right\|_2^2 \mathbb{1}\left\{ (X_t, X_{t-1}) \notin \mathcal{A}_t \right\} \right]$$

$$= \frac{\sigma_t^{\star 2}}{\alpha_t} \mathbb{E}\left[ \| Z_t \|_2^2 \mathbb{1}\left\{ (X_t, X_{t-1}) \notin \mathcal{A}_t \right\} \right] \overset{\text{(ii)}}{\leq} \frac{(1-\alpha_t)(\alpha_t - \overline{\alpha}_t)}{(1-\overline{\alpha}_t)\alpha_t} \mathbb{E}^{1/2} \left[ \| Z_t \|_2^4 \right] \mathbb{P}^{1/2}\left( (X_t, X_{t-1}) \notin \mathcal{A}_t \right)$$

$$\overset{\text{(iii)}}{\leq} \frac{8 c_1 \log T}{T} \exp\left( -\frac{C_1}{8} k \log T \right) \mathbb{E}^{1/2}\left[ \| Z_t \|_2^4 \right] \overset{\text{(iv)}}{\leq} \exp\left( -\frac{C_1}{16} k \log T \right).$$

Here step (i) follows from the definition of $Y_t^\star$ in (4.1) and of $\mu_t^\star$ in (4.7); step (ii) follows from the Cauchy-Schwarz inequality; step (iii) utilizes Lemma 8 and Lemma 1; while step (iv) follows from $Z_t \sim \mathcal{N}(0, I_d)$ and holds provided that $T$ is sufficiently large and $k \geq \log d$. Taking the above bounds collectively yields

$$|K_{t,2}| \leq \gamma_1 \gamma_2 \leq \sqrt{\gamma_{1,1} + \gamma_{1,2}} \gamma_2 \leq 2 \exp\left( -\frac{C_1}{32} k \log T \right) \mathbb{E}_{x_t \sim q_t}^{1/2} \left[ \| \varepsilon_t (x_t) \|_2^2 \right].$$

# B  Proof of Theorem 2

In view of the update rule (2.1), the variables $X_0, X_1, \ldots, X_T$ are jointly Gaussian, and we can check from (2.2) that

$$X_t = \sqrt{\overline{\alpha}_t} X_0 + \sqrt{1-\overline{\alpha}_t} \, \overline{W}_t \sim \mathcal{N}\left( 0, \overline{\alpha}_t I_k + (1-\overline{\alpha}_t) I_d \right), \tag{B.1}$$

hence the score functions

$$s_t^\star (x) = -\left( \overline{\alpha}_t I_k + (1-\overline{\alpha}_t) I_d \right)^{-1} x, \qquad \forall x \in \mathbb{R}^d. \tag{B.2}$$

We first derive the density of $X_{t-1}$ conditional on $X_t = x_t$. Since the joint distribution of $(X_{t-1}, X_t)$ is

$$
\begin{bmatrix} X_{t-1} \\ X_t \end{bmatrix} \sim \mathcal{N} \left( \begin{bmatrix} 0 \\ 0 \end{bmatrix}, \begin{bmatrix} \overline{\alpha}_{t-1} I_k + (1 - \overline{\alpha}_{t-1}) I_d & \sqrt{\alpha_t} \left( \overline{\alpha}_{t-1} I_k + (1 - \overline{\alpha}_{t-1}) I_d \right) \\ \sqrt{\alpha_t} \left( \overline{\alpha}_{t-1} I_k + (1 - \overline{\alpha}_{t-1}) I_d \right) & \overline{\alpha}_t I_k + (1 - \overline{\alpha}_t) I_d \end{bmatrix} \right),
$$

we can derive that

$$
X_{t-1} \mid X_t = x_t \sim \mathcal{N} \left( \sqrt{\alpha_t} \left( I_k + \frac{1 - \overline{\alpha}_{t-1}}{1 - \overline{\alpha}_t} \left( I_d - I_k \right) \right) x_t, (1 - \alpha_t) \left( I_k + \frac{1 - \overline{\alpha}_{t-1}}{1 - \overline{\alpha}_t} \left( I_d - I_k \right) \right) \right).
$$

In addition, with perfect score estimation, we can use (2.3) and (B.2) to achieve

$$
Y_{t-1} = \frac{Y_t + \eta_t s_t^\star (Y_t) + \sigma_t Z_t}{\sqrt{\alpha_t}} = \frac{Y_t - \eta_t \left( \overline{\alpha}_t I_k + (1 - \overline{\alpha}_t) I_d \right)^{-1} Y_t + \sigma_t Z_t}{\sqrt{\alpha_t}},
$$

which indicates that

$$
Y_{t-1} \mid Y_t = x_t \sim \mathcal{N} \left( \frac{1}{\sqrt{\alpha_t}} \left( (1 - \eta_t) I_k + \left( 1 - \frac{\eta_t}{1 - \overline{\alpha}_t} \right) (I_d - I_k) \right) x_t, \frac{\sigma_t^2}{\alpha_t} I_d \right).
$$

Then we can check that for any $x_t \in \mathbb{R}^d$,

$$
\mathsf{KL} \left( p_{X_{t-1} \mid X_t} \left( \cdot \mid x_t \right) \, \| \, p_{Y_{t-1} \mid Y_t} \left( \cdot \mid x_t \right) \right)
$$

$$
= \frac{(1 - \alpha_t - \eta_t)^2}{2 \sigma_t^2} \| I_k x_t \|_2^2 + \frac{k}{2} \left( \frac{\alpha_t (1 - \alpha_t)}{\sigma_t^2} - \log \frac{\alpha_t (1 - \alpha_t)}{\sigma_t^2} - 1 \right)
$$

$$
+ \frac{(1 - \alpha_t - \eta_t)^2}{2 (1 - \overline{\alpha}_t)} \| (I_d - I_k) x_t \|_2^2 + \frac{d - k}{2} \left( \frac{(1 - \alpha_t) (\alpha_t - \overline{\alpha}_t)}{\sigma_t^2 (1 - \alpha_t)} - \log \frac{(1 - \alpha_t) (\alpha_t - \overline{\alpha}_t)}{\sigma_t^2 (1 - \alpha_t)} - 1 \right).
$$

One can check that

$$
z - \log z - 1 \geq 0.1 \min \left\{ 1, (z - 1)^2 \right\}, \qquad \forall z > 0.
$$

We combine the above two relations as well as the assumption that $k \leq d/2$ to achieve

$$
\mathsf{KL} \left( p_{X_{t-1} \mid X_t} \left( \cdot \mid x_t \right) \, \| \, p_{Y_{t-1} \mid Y_t} \left( \cdot \mid x_t \right) \right) \geq \frac{(1 - \alpha_t - \eta_t)^2}{2 (1 - \overline{\alpha}_t)} \| (I_d - I_k) x_t \|_2^2 + \frac{d}{40} \left( \frac{(1 - \alpha_t) (\alpha_t - \overline{\alpha}_t)}{\sigma_t^2 (1 - \alpha_t)} - 1 \right)^2.
$$

By taking expectation w.r.t. $x_t$, we have

$$
\mathbb{E}_{x_t \sim q_t} \left[ \mathsf{KL} \left( p_{X_{t-1} \mid X_t} \left( \cdot \mid x_t \right) \, \| \, p_{Y_{t-1} \mid Y_t} \left( \cdot \mid x_t \right) \right) \right] \geq \frac{d}{4} (1 - \alpha_t - \eta_t)^2 + \frac{d}{40} \left( \frac{(1 - \alpha_t) (\alpha_t - \overline{\alpha}_t)}{\sigma_t^2 (1 - \alpha_t)} - 1 \right)^2,
$$

where we use the fact that

$$
\mathbb{E}_{x_t \sim q_t} \left[ \| (I_d - I_k) x_t \|_2^2 \right] = (d - k) (1 - \overline{\alpha}_t) \geq \frac{d}{2} (1 - \overline{\alpha}_t).
$$

## C  Technical lemmas

This section collects a few useful technical tools that are useful in the analysis.

**Lemma 8.** *When $T$ is sufficiently large, for $1 \leq t \leq T$, we have*

$$
\alpha_t \geq 1 - \frac{c_1 \log T}{T} \geq \frac{1}{2}.
$$

*In addition, for $2 \leq t \leq T$, we have*

$$
\frac{1 - \alpha_t}{1 - \overline{\alpha}_t} \leq \frac{1 - \alpha_t}{\alpha_t - \overline{\alpha}_t} \leq \frac{8 c_1 \log T}{T}.
$$

*Proof.* See Appendix A.2 in [14]. $\qquad \square$

**Lemma 9.** *For $Z \sim \mathcal{N}(0,1)$ and any $t \geq 1$, we know thatProposition 2.1.2*

$$\mathbb{P}\left(|Z| \geq t\right) \leq e^{-t^2/2}, \qquad \forall t \geq 1.$$

*In addition, for a chi-square random variable $Y \sim \chi^2(d)$, we have*

$$\mathbb{P}(\sqrt{Y} \geq \sqrt{d} + t) \leq e^{-t^2/2}, \qquad \forall t \geq 1.$$

*Proof.* See Proposition 2.1.2 in [27] and Section 4.1 in [12]. □

**Lemma 10.** *Suppose that $T$ is sufficiently large. Then we have*

$$\mathsf{KL}\left(p_{X_T} \| p_{Y_T}\right) \leq T^{-100}.$$

*Proof.* See Lemma 3 in [14]. □

