# OpenReview forum: "Adapting to Unknown Low-Dimensional Structures in Score-Based Diffusion Models"
_NeurIPS.cc/2024/Conference — NeurIPS 2024 poster_

### Official Review · Reviewer_kJLP · 2024-07-07

**Soundness:** 3
**Presentation:** 3
**Contribution:** 2
**Rating:** 5
**Confidence:** 3

**Summary:**

The paper considers the problem of convergence of probability distribution learned by diffusion models to target data distribution when the data distribution lies has some low-dimensional structure. Previous work either provides a convergence rate bound in Wasserstein distance for low-dimensional distribution or provides a bound in total variation distance with polynomial dependence on the ambient dimension. This paper improves the convergence rate to logarithmically depend on the ambient dimension and polynomially on the intrinsic dimension.

**Strengths:**

- It is interesting that the authors do not only consider the case of data on a k-dimensional manifold but consider a general version of a low-dimensional structure provided by the covering number of the support.
- The paper is overall well-written and easy to follow.

**Weaknesses:**

- The result in the paper is proved under the assumption of the bounded support of the data distribution (i.e. $|| x || \leq R$ where $R = \textrm{poly}(T)$). The results in the previous works are under the assumption that the data has a bounded second moment therefore, the bounded support assumption is more restrictive than the assumptions considered in the previous work.
- The argument about the uniqueness of coefficients that leads to dimension-independent discretization error in Section 3.2 is not convincing because it proves a lower bound on the error incurred in one denoising step which is an upper bound on the total variation distance (the authors also mention it).

**Questions:**

- I understand that showing the lower bound on the total variation error in section 3.2 might be difficult but can you experimentally show that the claimed coefficient in eq.(2.4) leads to a better convergence compared to other coefficients (e.g., coefficients used in practice) on some synthetic low-dimensional distribution?
- The final bound on T in theorem 1 seems to depend on polylogarithmically on the dimension of the distribution (instead of polynomially in previous works). Is the polylogarithmical dependence of dimension necessary?

Minor comment: it might be worth including a discussion in the paper on why earlier approaches get the dimension-dependent error.

**Limitations:**

Yes, the authors discuss the limitations.

---

> ### Author Rebuttal · Authors · 2024-08-07
>
> Thank you for your review. We address your comments below.
>
> **Bounded assumption on the target distribution.**
> Thank you for raising this point!
> - We agree that the bounded support assumption is stronger than, for example, the bounded second moment assumption, and it excludes Gaussian distributions. However, we argue that this condition is satisfied in most practical scenarios. Arguably, the most important applications of diffusion models are to generate new images from a target distribution, where image data is naturally bounded. For instance, the CIFAR and ImageNet datasets consist of images with pixel values ranging from 0 to 255 or $[0, 1]$, and it is common to normalize them to the range $[-1, 1]$. Then the $\ell_2$ norm of an image from, e.g., the CIFAR dataset, is typically below $60$. Moreover, for DDPM in image generative tasks, $T$ is typically around $100$. Hence, we believe it is reasonable to assume in theoretical analysis that the radius of the data distribution is bounded by $T^{c_R}$, where $c_R$ is any given fixed constant that can be very large.
> - We acknowledge that it is unfortunate not to cover common distributions like the Gaussian. Although we believe the bounded support assumption is not essential, it greatly facilitates our analysis, especially since we use a novel set of analytical tools that characterize the dynamics of DDPM in a more deterministic manner to establish convergence guarantees in the presence of low-dimensional structure. While we can relax this assumption in some specific cases (e.g., for Gaussian distribution, as illustrated in the next paragraph), we find it challenging to achieve a clean and concise result that holds generally for any distribution with a finite second moment. We leave this to future investigation.
>
> We will incorporate these discussions in our next revision.
>
> **Regarding the gap between our lower bound and the TV error.**
> Thank you for your suggestion! We conduct an experiment to examine whether our coefficient design (3.2) is indeed the unique coefficient design that leads to dimension-independent TV error $\mathsf{TV}(q_1,p_1)$, described as follows.
>
> Using the degenerated Gaussian distribution considered in Theorem 2 as a tractable example, we ran the DDPM sampler with exact score functions (so that the error only comes from discretization) and plotted the error curves of $\mathsf{TV}(q_1,p_1)$ versus the ambient dimension $d$ under several different choices of $T$. We implemented both our coefficient design (2.4) and another widely-used design $\eta_t = \sigma_t = 1-\alpha_t$. The results demonstrate that the TV error $\mathsf{TV}(q_1,p_1)$ is independent of the ambient dimension $d$ under our coefficient design (2.4), while it grows with $d$ when using the other design. **Please refer to the figures in the attached PDF in the response to all reviewers.** This supports our key message that (3.2) represents a unique coefficient design for DDPM in achieving dimension-independent error.
>
> **Dependency on $\log d$.**
> Thank you for raising this point! The polylogarithmic dependency w.r.t.~the ambient dimension $d$ arises from the analysis tools that allows us to characterize the algorithmic dynamics in a more deterministic manner. This analysis framework is crucial for tackling the convergence of DDPM in the presence of low-dimensional structure.
>
> In the proof, we identified a "typical" high-probability set where we are able to characterize the evolution of conditional density precisely (see our discussion at the beginning of Section 4). However, the algorithmic dynamic outside this set is difficult to track, and we have to bound some quantities related to the dynamics outside the typical set at the order of $d$. As a result, we have to set the radius of the "typical" high-probability set to be large enough (larger than some logarithmic factor of $d$), such that the exceptional probability is small (e.g., smaller than $O(1/\mathsf{poly}(d))$ in order to offset the impact of these terms. We think it might be possible to improve the polylogarithmic dependency on $d$ via more refined analysis, and we leave it to future investigation.
>
> **Discussion on coefficient design.**
> Thank you for the suggestion! We have extended Theorem 2 to a general lower bound that works for arbitrary low-dimensional data distributions. **Please see our response to all reviewers for this general lower bound, as well as an outline of the proof.** We believe that this general result can help us understand the role of our coefficient design (2.4), as well as why earlier designs  get the dimension-dependent error. To facilitate discussion, we copy the lower bound here:
> $$\\mathbb{E}_{x\_{t}\sim q\_{t}}[\\mathsf{KL}(p\_{X\_{t-1}|X\_{t}}(\\cdot| x\_{t})\\parallel p\_{Y\_{t-1}|Y\_{t}}(\\cdot| x\_{t}))] \\geq \\bigg(\\frac{\\sigma\_{t}^{\\star2}}{\\sigma\_t^2} + 2\\log\\frac{\\sigma\_t}{\\sigma\_t^\\star}- 1\\bigg)\\frac{d}{2} + \\frac{c\_0(\\eta\_{t}-\\eta\_{t}^{\\star})^2d}{2\\sigma\_{t}^{2}(1-\\overline{\\alpha}\_t)}  - C\_{5}^2\\frac{k^{4}\\log^{6}T}{T^2}\\bigg(3+\\frac{\\sigma\_{t}^{\\star2}}{\\sigma_t^2}\\bigg) - C\_{5}\\frac{k^{2}\\log^{3}T}{T}\\bigg|\\frac{\\sigma\_{t}^{\\star2}}{\\sigma\_t^2} - 1\\bigg|\\sqrt{d} - \\exp(-c\_1k\\log T)$$
> This bound is achieved by a novel set of analysis tools that characterize the algorithmic dynamics in a more deterministic manner, which together with our analysis in Theorem 1 provide a sharp characterization of the impact of coefficient design (i.e., $\eta_t$ and $\sigma_t$) in determining the error induced in each denoising step. It can be seen that, unless we take $\eta_t=\eta_t^\star$  and $\sigma_t=\sigma_t^\star$ as suggested in (2.4), there will be inevitable error that scales at least linear in $d$ incurred in each denoising step.

---

> ### Comment · Reviewer_kJLP · 2024-08-11
>
> I thank the authors for providing the detailed rebuttal and also providing the additional experiment for the uniqueness of the coefficient.
>
> - I understand some of the explanations provided by the authors but as authors mentioned, each pixel is normalized between 0 to 1 or -1 to 1. If we think of each pixel as a coordinate in the input, then the norm of the input is typically $O(\sqrt{d})$ but the result is proven under the assumption that norm of the input is $poly(T) = poly(k, log(d))$.
> - The experiments on the synthetic data are interesting. Can you provide more details on the baseline choices for $\eta_t$ and $\sigma_t$? The original DDPM paper [1] uses different $\alpha_t$ and $\sigma_t^2 = \sqrt{\alpha}_t \beta_t$ (note that the definition of $\sigma_t$ differs in this work and the original work). The follow-up work [2] also seems to have different choices than $\eta_t$ and $\sigma_t$ than the one used in the above experiment so can you provide some of the references that use the choices ($\eta_t = \sigma_t = 1-\alpha_t$)?
>
>
>
> [1] Denoising Diffusion Probabilistic Models
>
> [2] Improved Denoising Diffusion Probabilistic Models

---

> ### Author Response · Authors · 2024-08-12
>
> Thank you for your reply!
>
> **On the $d$-dependence.** Thanks for the insightful comment. We agree that the norm of the input image $X_0$ is typically $\sqrt{d}$. However this won't affect our main message, and the reason is as follows:
> * Our paper requires that, for some fixed (arbitrary) constant $c_R$,  the norm of $X_0$ is bounded by $T^{c_R}$. This means that we need $T \geq d^{1/(2c_R)}$. Since $c_R$ can be chosen arbitrarily, this is weaker than any polynomial dependence on $d$.
> * In fact, we can see that the $d$-dependence in this requirement can be removed at the price of an additional $\log d$ factor in the final error bound. Although we treat $c_R$ as a universal constant in the current paper and hide its dependence in Theorem 1, the final upper bound in Theorem 1 actually depends polynomially on $c_R$ as follows:
> $$\\mathsf{TV}\\left(q\_{1},p\_{1}\\right)\\lesssim c\_R^2\\frac{\\left(k+\\log d\\right)^{2}\\log^{3}T}{\\sqrt{T}}+\\varepsilon\_{\\mathsf{score}}\\log T.$$
> We will make this dependence explicit in the next revision. If we take $c_R \asymp \log d$, then our error bound becomes larger by a factor of $\log^2 d$, which is still nearly as good as the current one. However, notice that $d^{1/\Omega(\log d)} = O(1)$, the requirement $T \geq d^{1/(2c_R)} = O(1)$ is independent of $d$.
>
> **On the choices of $\eta_t$ and $\sigma_t$.** Thanks for raising this point. We use the same $\eta_t$ as [1] and [2], and we agree that the choices of $\sigma_t$ in [1] and [2] are different with our choices in the experiment.
> Here, [1] chose $\sigma_t^2 = \alpha_t\beta_t$, while our choice is $\sigma_t^2 = \beta_t$.
> We use this type of $\sigma_t^2$ as our baseline, since it is commonly adopted in the theoretical literature (e.g., [3]) and has been shown to achieve $\mathsf{poly}(d)/T$ convergence.
> In addition, we have also implemented the choice $\sigma_t^2 = \alpha_t\beta_t$ in [1] in the numerical experiments, and the performance is very similar to using $\sigma_t^2 = \beta_t$. We will include this result in our next revision.
> Our lower bound based on the degenerated Gaussian distribution also show that this choice $\sigma_t^2 = \alpha_t\beta_t$ will incur some error that is linear in $d$.
>
> Note that $\sigma_t^2$ in [2] is learned from the dataset in the hope of achieving optimal performance, which takes a different approach from our current work, where we intend to identify the coefficients based on the learning rates of the forward process (i.e., $\alpha_t$'s) in a non-data-driven way. will discuss this point more clearly in the next revision.
>
> [3] Towards Non-Asymptotic Convergence for Diffusion-Based Generative Models, G. Li, Y. Wei, Y. Chen and Y. Chi, ICLR 2024.

---

> > ### Author Response · Authors · 2024-08-13
> >
> > Thanks again for your efforts in reviewing our paper and for your helpful comments! We have carefully considered your questions and addressed them in our response. The discussion phase is due to conclude in less than 20 hours, and we would like to know whether our response has appropriately addressed your questions and concerns about our paper. If we have addressed your concerns, we would appreciate it if you consider increasing your score for our paper. Please let us know if you have further comments or concerns about our paper. Thank you!

---

### Official Review · Reviewer_SgF5 · 2024-07-13

**Soundness:** 3
**Presentation:** 3
**Contribution:** 4
**Rating:** 7
**Confidence:** 3

**Summary:**

In DDPM-style diffusion models, we need to discretize a continuous
process.  Benton et al [3] showed that ~ d/eps^2 steps suffice over d
dimensions.  But what if (as is typical) the data lie in a space of
lower intrinsic dimension (e.g. k-sparse or a k-dimensional manifold)?
This paper shows how to replace the d in the iteration complexity by
k^4.

**Strengths:**

This paper shows that DDPM can be nearly dimension independent, up to
log factors only depending on the intrinsic dimension.  This holds for
a very general definition of intrinsic dimension---just the log
covering number.  I was pretty surprised to learn that such a result
is possible.

Doing so requires particular choices of step size, which they show is
necessary, at least for the typical Girsanov-style proof approach.

**Weaknesses:**

Obviously it would be nice to see k rather than k^4.  We don't
typically expect intrinsic dimension << d^{1/4}, so this isn't giving
a real quantitative improvement.

There's very little intuition for what's going on.  Why are these the
right step sizes?

**Questions:**

How does the schedule (2.4) compare to the suggestion from [3]?

What happens if X isn't exactly low dimensional, but is close?  Say,
there is eps TV mass outside the cover.

**Limitations:**

None.

---

> ### Author Rebuttal · Authors · 2024-08-07
>
> Thank you for your review. We address your comments below.
>
> **Intuition for our coefficient design.**
> Thanks for raising this point. We have extended Theorem 2 to a general lower bound that works for arbitrary low-dimensional data distributions. **Please see our response to all reviewers for this general lower bound, as well as an outline of the proof.** We believe that this general result can help us understand the role of our coefficient design (2.4). To facilitate discussion, we copy the lower bound here:
> $$\\mathbb{E}_{x\_{t}\sim q\_{t}}[\\mathsf{KL}(p\_{X\_{t-1}|X\_{t}}(\\cdot| x\_{t})\\parallel p\_{Y\_{t-1}|Y\_{t}}(\\cdot| x\_{t}))] \\geq \\bigg(\\frac{\\sigma\_{t}^{\\star2}}{\\sigma\_t^2} + 2\\log\\frac{\\sigma\_t}{\\sigma\_t^\\star}- 1\\bigg)\\frac{d}{2} + \\frac{c\_0(\\eta\_{t}-\\eta\_{t}^{\\star})^2d}{2\\sigma\_{t}^{2}(1-\\overline{\\alpha}\_t)}  - C\_{5}^2\\frac{k^{4}\\log^{6}T}{T^2}\\bigg(3+\\frac{\\sigma\_{t}^{\\star2}}{\\sigma_t^2}\\bigg) - C\_{5}\\frac{k^{2}\\log^{3}T}{T}\\bigg|\\frac{\\sigma\_{t}^{\\star2}}{\\sigma\_t^2} - 1\\bigg|\\sqrt{d} - \\exp(-c\_1k\\log T)$$
> This bound is achieved by a novel set of analysis tools that characterize the algorithmic dynamics in a more deterministic manner, which together with our analysis in Theorem 1 provide a sharp characterization of the impact of coefficient design (i.e., $\eta_t$ and $\sigma_t$) in determining the error induced in each denoising step. It can be seen that, unless we take $\eta_t=\eta_t^\star$  and $\sigma_t=\sigma_t^\star$ as suggested in (2.4), there will be inevitable error that scales at least linear in $d$ incurred in each denoising step.
>
> **Comparison with other coefficient designs.**
> Thank you for raising this point. In the prior work [3], the marginal distribution of the forward process is $X_{t_k} \sim e^{-t_k}X_0 + \sqrt{1-e^{-2t_k}}\,\overline{W}$ where $\overline{W}\sim\mathcal{N}(0,I_d)$.
> This means that $e^{-2t_k}$ plays the role of $\\overline{\\alpha}\_t$ in our paper.
> By examining their discretization rule (see Eq.~(4) therein), we find that they use the following update rule:
> $$
> Y_{t-1} = (1+\Delta_t)Y_t + 2\Delta_ts_t(Y_t) + \sqrt{\Delta_t}Z_t,
> $$
> where $\Delta_t = \frac{1}{2}\log\frac{1}{\alpha_t}$.
> By applying a similar calculation as in Theorem 2, we can show that:
> $$
> \\mathbb{E}\_{x\_{t}\\sim q\_{t}}\left[\mathsf{KL}\left(p_{X_{t-1}|X_{t}}\left(\cdot| x_{t}\right)\,\Vert\,p_{Y_{t-1}|Y_{t}}\left(\cdot| x_{t}\right)\right)\right]
> \gtrsim \frac{d}{T^2}.
> $$
> Hence, the coefficient design in [3] will also incur dimension-dependent error in each denoising step. We will incorporate these discussions in our next revision.
>
>
> **Approximately low-dimensional structure.**
> Thank you for raising this point. We first would like to clarify that we are not assuming that the data distribution is exactly low dimensional. Our characterization of low-dimensionality is based on the covering number of the support, as defined in Section 2, which accommodates approximately low-dimensional structure. For example, our setup includes the case that $\mathsf{supp}(p_\mathsf{data}) \in \cup_{i \le N_{\varepsilon}} \mathbb{B}(c_i, \varepsilon)$, the balls centered at $c_i$ with $\varepsilon$ radius, whose dimension is $d$ instead of $k$. Therefore even if the support of the data distribution is not exactly low-dimenaional, our results can still be applied. (We apologies if we misunderstood your question.)
>
> It is an interesting question whether our result is stable against adding an $\varepsilon$-mass outside the cover. This setting is beyond our current result, since this mass can be spread out in the full space (thus having large covering number that depends on $d$). We conjecture that our result will be stable to this perturbation, namely it is possible to show an error bound similar to Theorem 1, but has an additional term proportional to $\varepsilon$ (or $\varepsilon d$). Currently, we don't know how to prove this result, and we leave this for future investigation.
>
> **Improving the quartic dependency on $k$.**
> Thank you for raising this point. Currently, we don't know how to improve this dependency, which we believe requires new analysis frameworks and tools. We leave this for future investigation.

---

> > ### Comment · Reviewer_SgF5 · 2024-08-09
> >
> > Thank you for your response.

---

### Official Review · Reviewer_1aoU · 2024-07-20

**Soundness:** 3
**Presentation:** 3
**Contribution:** 3
**Rating:** 6
**Confidence:** 4

**Summary:**

This paper investigates score-based diffusion models when the underlying distribution is near low-dimensional manifolds in a higher-dimensional space. It addresses the gap in theoretical understanding of diffusion models, which are suboptimal in the presence of low-dimensional structures. For the DDPM, the error dependency on the ambient dimension $d$ is generally unavoidable during each denoising step in the existing literature. However, the authors identify a unique set of coefficients that yields a convergence rate of $O(k^2/\sqrt{T})$, where $k$ is the intrinsic dimension and $T$ is the number of steps. The analysis employs novel tools that characterize the algorithmic dynamics in a deterministic manner. Additionally, the paper establishes that the DDPM sampler's error, influenced by time discretization and score estimation, is nearly dimension-free, with the ambient dimension $d$ appearing only in logarithmic terms.

**Strengths:**

**1. Theoretical Advancement in Understanding Diffusion Models.** The paper makes a good contribution to the theoretical understanding of score-based diffusion models in the context of low-dimensional manifolds. By identifying a unique set of coefficients that yield a convergence rate dependent on the intrinsic dimension $k$ rather than the ambient dimension $d$, the authors provide a refined theoretical framework that addresses previously suboptimal theoretical support.

**2. New Analytical Tools and Methodology.** The paper introduces a new set of analytical tools to characterize the algorithmic dynamics of the DDPM sampler in a deterministic manner. This innovative approach allows for a more precise analysis of the error sources—time discretization and score estimation errors.

**3.** In general, the paper is well-written and joyful to read.

**Weaknesses:**

**1. Strict Assumption on the Data Distribution** In Line 126, the authors assume that the support set of the data distribution is bounded. This assumption is overly restrictive and may not hold for many real-world data distributions, such as Gaussian distributions, which have unbounded support. The authors should consider relaxing this assumption or providing justification for its necessity, as well as discussing the implications of this restriction on the generalizability and robustness of their findings.

**2.  Lack of Empirical Demonstration.** Although this paper presents a solid and sharp theoretical analysis of the convergence rate of DDPM, the authors don't provide any experimental results to support their theory. Without experimental evidence, it is difficult to assess the real-world performance and robustness of the proposed coefficient design and convergence rate improvements.

**Questions:**

**Q1.** Is it widely used in the literature to employ $\epsilon$-net and cover number to measure the intrinsic dimension of a data distribution? The authors should provide a more thorough discussion on this point.

**Q2.** In Line 126, the authors assume that the support of the data distribution is bounded. However, the support of the Gaussian distribution in Theorem 2 is not bounded. It is believed that this is not a good example to demonstrate the uniqueness of coefficient design.

---

> ### Author Rebuttal · Authors · 2024-08-07
>
> Thank you for your review. We address your comments below.
>
> **Bounded assumption on the target distribution.**
> - We agree that the bounded support assumption is stronger than, for example, the bounded second moment assumption, and it excludes Gaussian distributions. However, we argue that this condition is satisfied in most practical scenarios. Arguably, the most important applications of diffusion models are to generate new images from a target distribution, where image data is naturally bounded. For instance, the CIFAR and ImageNet datasets consist of images with pixel values ranging from 0 to 255 or $[0, 1]$, and it is common to normalize them to the range $[-1, 1]$. Then the $\ell_2$ norm of an image from, e.g., the CIFAR dataset, is typically below $60$. Moreover, for DDPM in image generative tasks, $T$ is typically around $100$. Hence, we believe it is reasonable to assume in theoretical analysis that the radius of the data distribution is bounded by $T^{c_R}$, where $c_R$ is any given fixed constant that can be very large.
> - We acknowledge that it is unfortunate not to cover common distributions like the Gaussian. Although we believe the bounded support assumption is not essential, it greatly facilitates our analysis, especially since we use a novel set of analytical tools that characterize the dynamics of DDPM in a more deterministic manner to establish convergence guarantees in the presence of low-dimensional structure. While we can relax this assumption in some specific cases (e.g., for Gaussian distribution, as illustrated in the next paragraph), we find it challenging to achieve a clean and concise result that holds generally for any distribution with a finite second moment. We leave this to future investigation.
>
> We will incorporate these discussions in our next revision.
>
> **Truncating Gaussian distribution in the lower bound.**
> We agree that the degenerated Gaussian distribution considered in Theorem 2 is not a good example because it is unbounded. As we mentioned in footnote 2 on page 5, this lower bound can be extended to the truncated Gauss distribution. We have changed the result to truncated Gaussian in the following.
>
> Before presenting the details, we also want to highlight that we have extended Theorem 2 to a general lower bound that works for arbitrary low-dimensional data distributions, which we believe also helps addressing this comment. **Please see our response to all reviewers for this general lower bound, as well as an outline of the proof.** Now we continue presenting results for the truncated Gaussian.
>
> Let $X_0 \sim \mathcal{N}(0,I_k)$, and define its truncated counterpart $\\widetilde{X}\_0$  as
> $$
> p\_{\\widetilde{X}\_0} (x_0) \\propto p_{X_0} (x_0) 1(\\|x_0\\|_{\infty} \le T).
> $$
> Define $\\widetilde{X}\_t = \\sqrt{\\alpha\_t}\\widetilde{X}\_{t-1} + \\sqrt{1-\\alpha\_t}Z\_t$, and construct the reverse process $\\widetilde{Y}\_t$ with score estimation of $\\widetilde{X}\_t$. Then we can establish exactly the same lower bounds for $\\widetilde{X}\_t$ and $\\widetilde{Y}\_t$.
> Notice that $\\widetilde{X}\_t$ and $\\widetilde{Y}\_t$ have independent entries, hence it suffices to studying each entry separately (we use $\\widetilde{X}\_{t, i}$ and $\\widetilde{Y}\_{t, i}$ to denote their $i$-th entry), i.e.,
> $$
> \\mathsf{KL}(p\_{\\widetilde{X}\_{t-1}|\\widetilde{X}\_t}(\\cdot | x\_t) \\parallel p\_{\\widetilde{Y}\_{t-1}|\\widetilde{Y}\_t}(\\cdot | x\_t)) \\ge \\sum\_{i > k} \\mathsf{KL}(p\_{\\widetilde{X}\_{t-1, i}|\\widetilde{X}\_{t, i}}(\\cdot | x\_{t, i}) \\parallel p\_{\\widetilde{Y}\_{t-1, i}|\\widetilde{Y}\_{t, i}}(\\cdot | x\_{t, i})).
> $$
> The right hand side of the above inequality obeys
> $$
> \\text{RHS}= \\sum\_{i > k} \\mathsf{KL}(p\_{X\_{t-1, i}|X\_{t, i}}(\\cdot | x\_{t, i}) \\parallel p\_{Y\_{t-1, i}|Y\_{t, i}}(\\cdot | x\_{t, i}))\\ge \\frac{d}{4}\\left(\\eta\_{t}-\\eta\_{t}^{\\star}\\right)^{2}+\\frac{d}{40}\\left(\\frac{\\sigma\_{t}^{\\star2}}{\\sigma\_{t}^{2}}-1\\right)^{2},
> $$
> where the first relation holds since the truncation does not affect the entries with zero variance, and the second relation is proved in Line 550-552 in Appendix B.
>
> **Empirical demonstration.**
> Thank you for raising this point.
> We have conducted a numerical experiment using the data distribution considered in Theorem 2. The results demonstrate that the TV error and KL divergence are independent of the ambient dimension $d$ under our coefficient design (2.4), while they grow with $d$ under other widely-used designs. **Please refer to the figures in the attached PDF in the response to all reviewers**, where we plot the error curves of $\mathsf{KL}(q_1\parallel p_1)$ and $\mathsf{TV}(q_1,p_1)$ versus $d$, with all other setups fixed. This supports the conjecture that (2.4) represents a unique coefficient design for achieving dimension-independent error.
>
> While the lack of GPUs prevents us from examining this design in more complex, large-scale tasks, we believe that our theoretical findings and empirical observations are already meaningful and interesting.
>
> **Our use of $\varepsilon$-covering.**
> Thanks for raising this point! While we believe that it's a good idea to use covering number for characterizing low-dimensional structure in our problem, we are not aware of other existing literature that does the same. We choose to use $\varepsilon$-net and covering number to define approximate low-dimensional structure because we believe that this is less stringent and more general than assuming an exact low-dimensional structure (e.g., by assuming that the support of the data distribution lives in a low-dimensional subspace). As a sanity check, we showed in Section 2 that when the support of the data distribution lives in an $r$-dimensional
> subspace, our intrinsic dimension $k$ defined through covering number is of order $r$, confirming that our definition is indeed more general. We will incorporate these discussions in our next revision.

---

> > ### Comment · Reviewer_1aoU · 2024-08-13
> >
> > Thanks for the reviewers' rebuttal. I have two further comments: (i) For the bounded assumption, can this proof be applied to a uniform distribution oversphere? (ii) Can you provide some literature on using $\epsilon$-net to characterize intrinsic dimension?

---

> > > ### Author Response · Authors · 2024-08-13
> > >
> > > Thank you for your response!
> > >
> > > **Uniform distribution over sphere.** Yes, our proof and result can be applied to a uniform distribution over sphere, since the only assumption we imposed on the data distribution $p_{\mathsf{data}}$ is boundedness. However, the intrinsic dimension $k$ of, e.g., the unit sphere $\mathbb{S}^{d-1}$ in $\mathbb{R}^d$ is $d-1$, which is not a typical low-dimensional structure. Although our theory holds for general $k$, the most interesting regime is $k\ll d$, where our results significantly improve the convergence rate that has polynomial dependence on $d$.
> > >
> > > **Using covering number to characterize intrinsic dimension.** Thank you for asking this. Here we provide some related literature and discussion on this issue.
> > > * In fact, our definition of the intrinsic dimension $k$ is actually the metric entropy of $\mathcal{X}$, the support of $p_{\mathsf{data}}$. Metric entropy is defined using covering number, and is widely used in statistics and learning theory to characterize the complexity of a set/class in a metric space, which is useful in proving sample complexity and generalization bounds for algorithms; see e.g., Sections 5 and 14 in [1] for the reference. The low-dimensionality is also a concept of complexity, therefore we believe it is very natural to use covering number, or metric entropy to characterize the intrinsic dimension.
> > > * Prior literature [2], which studied diffusion model on low-dimensional data, assumes that the data is supported on a low-dimensional linear subspace. More generally, another work [3] assumes that the distribution is supported on a union of low-dimensional linear subspace. As we discussed in Section 2 and in the rebuttal, our intrinsic dimension $k$ defined through covering number is of order $k$ for a $k$-dimensional linear subspace, and we can also easily seen that $\sum_{i=1}^{m} k_i$ for the union of $m$ linear subspace (each with dimension $k_i$). Therefore using covering number to characterize intrinsic dimension actually admits the setup in these prior literature as special examples, and is more general and robust.
> > >
> > > The discussion phase is due to conclude in 20 hours, and we would like to know whether our response has appropriately addressed your questions and concerns about our paper. If we have addressed your concerns, we would appreciate it if you consider increasing your score for our paper. Please let us know if you have further comments or concerns about our paper. Thank you!
> > >
> > > [1] High-Dimensional Statistics: A Non-Asymptotics Viewpoint, M. J. Wainwright, Cambridge University Press, 2019.
> > >
> > > [2] Score Approximation, Estimation and Distribution Recovery of Diffusion Models on Low-Dimensional Data, M. Chen, K. Huang, T. Zhao, M. Wang, ICML 2023.
> > >
> > > [3] Robust Subspace Clustering, M. Soltanolkotabi, E. Elhamifar, E. J. Candes, Annals of Statistics, 2014.

---

> > > > ### Comment · Reviewer_1aoU · 2024-08-13
> > > >
> > > > Thanks for the authors' response, which addresses most of my concerns. I have increased my score.

---

### Official Review · Reviewer_NxHA · 2024-07-26

**Soundness:** 3
**Presentation:** 3
**Contribution:** 2
**Rating:** 4
**Confidence:** 3

**Summary:**

This paper discusses the DDPM sampler's capability to adapt to unknown low-dimensional structures in the target distribution, and studies how this informs the coefficient design.

**Strengths:**

The paper is written clearly and has a nice structure in general. It contributes to an important topic in diffusion models about adapting to lower dimensional structure. In the first part of the paper, the authors show that, with a particular coefficient design (2.4), the TV error of the DDPM sampler has an upper bound that depends on the intrinsic dimension. In the second part of the paper, the authors exemplify the unique choice of the coefficients.

**Weaknesses:**

The main weakness of the paper is the generality. The adaptivity to the lower dimensional structure is based on a particular coefficient design, however, this design is not shown to be unique for the TV error, or for general target data. It is this reviewer's opinion that it is better to write the contribution section more precisely in terms of the setup and the scope of the results. The concerns are specified in the questions section.

**Questions:**

1. How does this paper's result compare with [1], which does not require knowing/estimating the manifold?
Can the result in the paper lead to a tighter bound when a subspace is given a priori? For example, how does it compare to [2]?

[1] Rong Tang and Yun Yang. Adaptivity of diffusion models to manifold structures. In International Conference on Artificial Intelligence and Statistics, 2024.
[2] Kazusato Oko, Shunta Akiyama, and Taiji Suzuki. Diffusion models are minimax optimal distribution estimators. arXiv preprint arXiv:2303.01861, 2023.

2. Section 3.2 about unique coefficient design is very interesting, however, the results lack generality.
    - Theorem 2's result is derived in the case that the target data distribution is a standard Gaussian distribution. What can you get when considering a general data distribution?
    - As the authors noted at the end of the section, the uniqueness shown is with respect to the upper bound of the TV error. How does one make use of the uniqueness with respect to the upper bound in practice? What are the possible ways to address the coefficient design for the actual TV error, or tighten the gap?

**Limitations:**

The authors have disccussed the limitations.

---

> ### Author Rebuttal · Authors · 2024-08-07
>
> Thank you for your review. We address your comments below.
>
> **Comparison with prior works.**
> Thank you for the reference. These two works and the current paper approach the diffusion model from different perspectives, making direct comparison challenging. Specifically, the two prior works establish error bounds for estimating density under the assumption that the true target density meets certain smoothness conditions. Their results rely on a score-matching procedure designed based on the level of smoothness, situating their findings within the context of nonparametric statistics.
>
> In contrast, our paper approaches the problem from an applied mathematics perspective, decoupling the error of DDPM into discretization error and score matching error, and characterizing them separately, akin to prior works [3, 4, 6, 13]. Our results assume minimal conditions on the target distribution, avoiding smoothness assumptions, and accommodating arbitrary score-matching procedures. Therefore, our results cannot be directly compared with these prior works.
>
> Our paper does not require knowledge or estimation of the manifold, and our error bound is expressed as the sum of discretization error and score matching error:
> $$
> 	\\mathsf{TV}(q_{1},p_{1})\leq \\underbrace{C\frac{\left(k+\log d\right)^{2}\log^{3}T}{\sqrt{T}}}\_{ \text{discretization error}}
> 	+\\underbrace{C\varepsilon_{\mathsf{score}}\log T}\_{\text{score matching error}}.
> $$
> When a subspace is given a priori, we believe it will not help in improving the discretization error bound, as it already adapts to the low-dimensional structure automatically. While there might still be room for improvement (e.g., reducing the polynomial dependency on $k$), we believe this requires new analytical tools rather than assuming access to the low-dimensional structure. However, knowing the subspace can aid in improving the score-matching error, as it is possible to exploit the subspace information to design an efficient score-matching procedure that achieves a smaller $\varepsilon_{\mathsf{score}}$. This, however, is not the main focus of our paper, as our result accommodates any score-matching procedure, which we believe is more general. We leave this for future investigation.
>
> We will incorporate and discuss these references in our next revision.
>
> **Extending the lower bound to arbitrary data distribution.**
> Thank you for raising this point. As far as we know, it is quite general to use the worst-case error (or risk) to characterize the inherent difficulty of a problem in areas like statistics (e.g., minimax lower bound) and optimization (e.g., algorithmic lower bound). We believe that establishing a lower bound for the (degenerated) standard Gaussian distribution effectively illustrates our point: if the algorithm cannot perform well without using the proposed coefficient design in probably the simplest case, we can hardly expect it to work well in more complicated examples. However, the good news is that for this problem, we can actually show a similar lower bound for arbitrary low-dimensional data distributions. **Please see our response to all reviewers for this general lower bound, as well as an outline of the proof.** We will include this result in our next revision.
>
>
> **Regarding the gap between our lower bound and the TV error.**
> Thank you for raising this point.
> First, we would like to point out that we actually have an upper bound for the KL divergence between $q_1$ and $p_1$.
> In the analysis, we control $\mathsf{TV}(q_1, p_1)$ by bounding $\mathsf{KL}(q_1\parallel p_1)$ (see Eq.~(3.2)).
> Then the established KL lower bound in Theorem 2 is meaningful.
> We will added the KL divergence bound in our main result.
>
> In addition, the lower bound for KL divergence can also be extended to TV distance.
> According to the calculation in Appendix B (see Line 545 and 547), we know $p_{X_{t-1}|X_t}$ and $p_{Y_{t-1}|Y_t}$ are two gaussian distributions.
> Then with basic calculations, it can also be shown that
> $$
> \\mathsf{TV}(p_{X_{t-1}|X_{t}}, p_{Y_{t-1}|Y_{t}}) \\gtrsim \\min\\bigg\\{1, d\\left(\eta_{t}-\eta_{t}^{\star}\\right)^{2}+d\\left(\frac{\sigma_{t}^{\star2}}{\sigma_{t}^{2}}-1\\right)^{2}\\bigg\\}.
> $$
> We will also include this result in the next revision.
>
> Finally, due to the lack of GPUs, we conducted a numerical experiment using the data distribution considered in Theorem 2. The results demonstrate that the TV error and KL divergence are independent of the ambient dimension $d$ under our coefficient design (2.4), while they grow with $d$ under other widely-used designs. **Please refer to the figures in the attached PDF in the response to all reviewers**, where we plot the error curves of $\mathsf{KL}(q_1\parallel p_1)$ and $\mathsf{TV}(q_1,p_1)$ versus $d$, with all other setups fixed. This supports the conjecture that (2.4) represents a unique coefficient design for achieving dimension-independent error.

---

> > ### Comment · Reviewer_NxHA · 2024-08-12
> >
> > I thank the authors for the extensive rebuttal addressing the comments, and for the additional proof and numerical experiments. This comment again asks about the requirement on the target data, and the generality of the results. The generalization to a more general target data is not trivial as the probability measure of the target data is not absolute continuous with respect to the measure of the prior distribution. So, depending on the target data, the error in the denoising steps could explode. The choice of the time schedule would also be relevant here.

---

> ### Author Response · Authors · 2024-08-12
>
> Thank you for the reply! We are not sure whether we understand the "prior distribution" in your comment correctly. Please let us know if we misunderstood anything.
>
> - First, we would like to clarify that the only assumption we imposed on the target distribution $p_{\mathsf{data}}$  is that it has bounded support. We do not need any further assumption, e.g., absolute continuousness, in order to establish our results. In what follows, we will explain why we don't need to worry about the explosion or error, both in the upper bound (Theorem 1) and lower bound (Theorem 2).
> - Regarding the upper bound in Theorem 1, our error metric is the TV distance between the distribution of $X_1$ and $Y_1$ (i.e., $q_1$ and $p_1$), which are both absolutely continuous w.r.t.~the Lebesgue measure (i.e., they both have densities). This circumvents any potential issue when the data distribution $p_{\mathsf{data}}$ of $X_0$ is not continuous. Since $X_0$ and $X_1$ are exceedingly close (since $\beta_1=T^{-c_0}$ is vanishingly small), this error metric also reflects the closeness of the generator distribution and the target distribution.
> - The generalized lower bound stated in the rebuttal is established for the error incurred in each denoising step, which is defined as the expected KL divergence between two conditional distributions $p_{X_{t-1} | X_t}$ and $p_{Y_{t-1} | Y_t}$, for $2\leq t \leq T$ (again, it does not concern $t=1$ to circumvent the case when the data distribution is not absolutely continuous). These two distributions are both absolutely continuous w.r.t.~the Lebesgue measure (i.e., they both have densities), regardless of whether the target distribution $p_{\mathsf{data}}$ of $X_0$ is absolutely continuous or not.
> - To further support our claim above, we would like to mention that prior works [3,4,6,13] showed that even without using our coefficient design (2.4), other reasonable coefficient designs also lead to convergence rates $\mathsf{poly}(d)/T^2$ for the error incurred in each of the denoising steps (and they sum up to an overall error $\mathsf{poly}(d)/T$). This also suggests that the error in the denoising steps will not explode even if $p_{\mathsf{data}}$ is not continuous -- they just suffer from dependence on the ambient dimension $d$.
> - While our upper bound is established under the specific time schedule (i.e., $\beta_t$'s) in Section 2, the lower bound, including the one for degenerated Gaussian as well as the generalized one, holds for a reasonably large class of time schedules (only with the exception of some corner cases). We will specify this in our next revision.
>
> We are happy to discuss more details with you in case you have concern on the generalized lower bound.
> Nevertheless, as we discussed in the rebuttal, we believe that our original lower bound established for a simple example (degenerated Gaussian) already effectively illustrates our point. We present the generalized lower bound in the rebuttal just because we think it is a beautiful, concise, yet powerful result that can make the paper better.

---

> > ### Author Response · Authors · 2024-08-13
> >
> > Thanks again for your efforts in reviewing our paper and for your helpful comments! We have carefully considered your questions and addressed them in our response. The discussion phase is due to conclude in less than 20 hours, and we would like to know whether our response has appropriately addressed your questions and concerns about our paper. If we have addressed your concerns, we would appreciate it if you consider increasing your score for our paper. Please let us know if you have further comments or concerns about our paper. Thank you!

---

### Author Rebuttal · Authors · 2024-08-07

We thank all reviewers for their feedback. Here we address some common comments and questions.

**Extending the lower bound to arbitrary data distribution.** We agree with several reviewer's comments that the lower bound in Theorem 2 only covers Gaussian distribution, which can be restrictive. Here we generalize Theorem 2 to establish a similar lower bound for general low-dimensional distribution:

**Theorem.** Consider arbitrary data distribution $p\_{\mathsf{data}}$ satisfying the assumptions in Section 2. For the DDPM sampler (2.3) with perfect score estimation and arbitrary coefficients $\eta_t$ and $\sigma_t$, we have
$$\\mathbb{E}_{x\_{t}\sim q\_{t}}[\\mathsf{KL}(p\_{X\_{t-1}|X\_{t}}(\\cdot| x\_{t})\\parallel p\_{Y\_{t-1}|Y\_{t}}(\\cdot| x\_{t}))] \\geq \\bigg(\\frac{\\sigma\_{t}^{\\star2}}{\\sigma\_t^2} + 2\\log\\frac{\\sigma\_t}{\\sigma\_t^\\star}- 1\\bigg)\\frac{d}{2} + \\frac{c\_0(\\eta\_{t}-\\eta\_{t}^{\\star})^2d}{2\\sigma\_{t}^{2}(1-\\overline{\\alpha}\_t)}  - C\_{5}^2\\frac{k^{4}\\log^{6}T}{T^2}\\bigg(3+\\frac{\\sigma\_{t}^{\\star2}}{\\sigma_t^2}\\bigg) - C\_{5}\\frac{k^{2}\\log^{3}T}{T}\\bigg|\\frac{\\sigma\_{t}^{\\star2}}{\\sigma\_t^2} - 1\\bigg|\\sqrt{d} - \\exp(-c\_1k\\log T)$$
for each $2\leq t\leq T$, where $c_0,c_1,C_5>0$ are some universal constants.

Notice the fact that $x^2 - 2\log x -1 \geq 0$ for any $x>0$, and the equality holds only if $x=1$. Therefore the above results suggests that, unless $\eta_t=\eta_t^\star$ and $\sigma_t=\sigma_t^\star$, the corresponding denoising step will incur an error that is linear in $d$, when $d$ is sufficiently large. We will include this result in our next revision.

**Proof sketch.** Following the equation around Line 245 in Section 4.4, we start with the following bound:
$$\\mathbb{E}_{x\_{t}\sim q\_{t}}[\\mathsf{KL}(p\_{X\_{t-1}|X\_{t}}(\\cdot| x\_{t})\\parallel p\_{Y\_{t-1}|Y\_{t}}(\\cdot| x\_{t}))]\\ge
 \\int\_{x\_{t-1}, x\_{t}} p\_{X\_{t-1}|X\_{t}}\\left(x\_{t-1} |  x\_{t}\\right)\\log\\left(\\frac{p\_{Y\_{t-1}^{\\star}|Y\_{t}}\\left(x\_{t-1} |  x\_{t}\\right)}{p\_{Y\_{t-1}|Y\_{t}}\\left(x\_{t-1} |  x\_{t}\\right)}\\right)p\_{X\_{t}}\\left(x\_{t}\\right)\\mathrm{d}x\_{t-1}\\mathrm{d}x\_{t}
 =:\\mathcal{I}\_{t},$$
where we recall the definition of $Y\_{t-1}^{\\star}$ in (4.1). It boils down to understanding $\\mathcal{I}\_t$:
$$\\mathcal{I}\_{t}
 = d\\log \\frac{\\sigma\_{t}}{\\sigma\_{t}^{\\star}} + \\int\_{x\_{t}, x\_{t-1}} p\_{X\_{t}}\\left(x\_{t}\\right)p\_{X\_{t-1}|X\_{t}}\\left(x\_{t-1} |  x\_{t}\\right) \\bigg(\\frac{\\Vert\\sqrt{\\alpha\_{t}}z\_t-(\\eta\_{t}-\\eta\_{t}^{\\star})s\_{t}^{\\star}\\left(x\_{t}\\right)\\Vert\_{2}^{2}}{2\\sigma\_{t}^{2}} - \\frac{\\Vert\\sqrt{\\alpha\_{t}}z\_t\\Vert\_{2}^{2}}{2\\sigma\_{t}^{\\star2}}\\bigg)\\mathrm{d}x\_{t-1}\\mathrm{d}x\_{t},$$
where $\\sqrt{\\alpha\_{t}}z\_t := \\sqrt{\\alpha\_{t}}x\_{t-1}-x\_{t}-\\eta\_{t}^{\\star}s\_{t}^{\\star}(x\_{t}).$
Using similar analysis as in Lemma 6, the above integral on $\\mathcal{A}\_{t}^{\\mathrm{c}}$ (outside the typical set) can be controlled to the order of $\\exp(-\\Omega(k\\log T))$. According to Lemma 3, for $(x\_t,x\_{t-1})\\in \\mathcal{A}\_{t}$ we have
$$\\left|\\frac{p\_{X\_{t-1}|X\_{t}}\\left(x\_{t-1} |  x\_{t}\\right)}{p\_{Y\_{t-1}^{\\star}|Y\_{t}}\\left(x\_{t-1} |  x\_{t}\\right)}-1\\right|\\leq C\_{5}\\frac{k^{2}\\log^{3}T}{T}.$$
Therefore,
$$\\mathcal{I}\_{t} \\ge d\\log \\frac{\\sigma\_{t}}{\\sigma\_{t}^{\\star}} + \\int\_{x\_t}\\bigg(\\frac{\\Vert(\\eta\_{t}-\\eta\_{t}^{\\star})s\_{t}^{\\star}\\left(x\_{t}\\right)\\Vert\_{2}^{2}}{2\\sigma\_{t}^{2}} - C\_{5}\\frac{k^{2}\\log^{3}T}{T}\\frac{\\sigma\_{t}^{\\star}\\Vert(\\eta\_{t}-\\eta\_{t}^{\\star})s\_{t}^{\\star}(x\_{t})\\Vert\_{2}}{\\sigma\_{t}^2}\\bigg)p\_{X\_{t}}(x\_{t})\\mathrm{d}x\_{t}  + \\bigg(\\frac{\\sigma\_{t}^{\\star2}}{\\sigma\_t^2} - 1\\bigg)\\frac{d}{2} - C\_{5}\\frac{k^{2}\\log^{3}T}{T}\\bigg|\\frac{\\sigma\_{t}^{\\star2}}{\\sigma\_t^2} - 1\\bigg|\\sqrt{d} - \\exp(-c\_1k\\log T).$$
Here, we make use of the observation that for $0 < \delta < 1$,
$$ \\mathbb{P}\\bigg(\\bigg|\\frac{\\Vert\\sqrt{\\alpha\_{t}}z\_t\\Vert\_{2}^{2}}{\\sigma\_{t}^{\\star2}} - d\\bigg|^2 \\ge 2d\\log \\frac{1}{\delta}\\bigg) \\le \delta.$$
Then the desired lower bound follows from the following facts:
$$\\int\_{x\_t}C\_{5}\\frac{k^{2}\\log^{3}T}{T}\\frac{\\sigma\_{t}^{\\star}\\Vert(\\eta\_{t}-\\eta\_{t}^{\\star})s\_{t}^{\\star}\\left(x\_{t}\\right)\\Vert\_{2}}{\\sigma\_{t}^2}p\_{X\_{t}}\\left(x\_{t}\\right)\\mathrm{d}x\_{t}
 \\le \\int\_{x\_t}\\frac{\\Vert(\\eta\_{t}-\\eta\_{t}^{\\star})s\_{t}^{\\star}\\left(x\_{t}\\right)\\Vert\_{2}^{2}}{4\\sigma\_{t}^{2}}p\_{X\_{t}}\\left(x\_{t}\\right)\\mathrm{d}x\_{t} + C\_{5}^2\\frac{k^{4}\\log^{6}T}{T^2}\\frac{\\sigma\_{t}^{\\star2}}{\\sigma\_t^2},$$
as well as
$$ \\int\_{x\_t} \\|s\_{t}^{\\star}(x\_{t})\\|\_{2}^{2}p\_{X\_{t}}(x\_{t})\\mathrm{d}x\_{t} \\asymp \\frac{d}{1-\\overline{\\alpha}\_t}.$$
We will add the complete proof to our next revision.

**Setup of the numerical experiment.** We conduct some experiments to examine whether (2.4) is indeed the unique coefficient design that leads to dimension-independent error. We provide the setup for this experiment.
We use the degenerated Gaussian distribution $p_\mathsf{data}=\mathcal{N}(0,I_k)$ in Theorem 2 as a tractable example, and run the DDPM sampler with exact score functions (so that the error only comes from discretization). We fix the low intrinsic dimension $k=8$, and let the ambient dimension $d$ grow from $10$ to $10^3$. We implement the experiment for different number of steps $T \in\\{100,200,500,1000\\}$. We implemented both our coefficient design (2.4) and another widely-used design $\eta_t = \sigma_t = 1-\alpha_t$. The error curve of $\mathsf{KL}(q_1\parallel p_1)$ and $\mathsf{TV}(q_1,p_1)$ versus $d$ can be found in the attached PDF. This supports our key message that (2.4) represents a unique coefficient design for DDPM in achieving dimension-independent error.

---

### Comment · Area_Chair_78AZ · 2024-08-08

Dear authors, dear reviewers,

the discussion period has begun as the authors have provided their rebuttals.
I encourage the reviewers to read all the reviews and the corresponding rebuttals: the current period might be an opportunity for further clarification on the paper results and in general to engage in an open and constructive exchange.

Many thanks for your work.
The AC

---

### Decision · Program_Chairs · 2024-09-25

**Decision:**

Accept (poster)

**Comment:**

The goal of the paper is the analysis of denoising diffusion probabilistic models under the assumption that the underlying target distribution is concentrated on a low-dimensional manifold with respect to the ambient dimension $d$. The Authors characterize the convergence of the model to the target data distribution, whose rate is shown to depend on the dimensionality $k$ of the aforementioned low-dimensional manifold. The paper has been appreciated for its clarity and for its contribution to the study of the performance of diffusion models in the presence of an underlying low-dimensional data structure, also showing how a proper design of the coefficients in the model can allow to reach convergence rate depending on $k$ and not $d$. A limitation stressed by the Reviewers is related to the underlying hypothesis on the target distribution, which is assumed to have bounded support.

In conclusion, the evaluation of the submission is positive, and I recommend acceptance. As an additional note, numerical experiment has been performed during the rebuttal phase to further support the results. I recommend the authors to incorporate the results presented in the rebuttal phase in the manuscript and clearly discuss the limitations of the results due to the technical assumptions.